# Understanding the Role of Rehearsal Mechanism in Continual Learning under Varying Model Capacities

## Abstract

Continual learning, which aims to learn from dynamically changing data distributions, has garnered significant attention in recent years. However, most existing theoretical work focuses on regularization-based methods, while theoretical understanding of the rehearsal mechanism in continual learning remains limited. In this paper, we provide a closed-form analysis of adaptation, memory and generalization errors for rehearsal-based continual learning within a linear-Gaussian regression framework, covering both underparameterized and overparameterized regimes. We derive explicit formulae linking factors such as rehearsal size to each error component, and obtain several insightful findings. Firstly, more rehearsal does not always better for memorability, and there exists a decreasing floor for memory error when tasks are similar and noise levels are low. Secondly, rehearsal enhances adaptability under underparameterization, but can be provably detrimental under overparameterization. Moreover, enlarging the rehearsal size can raise peaks in generalization error when slightly overparameterized, and may further degrade generalization when tasks are dissimilar or noise is high. Finally, numerical simulations validate these theoretical insights and we further extend the analysis to neural networks on MNIST, CIFAR-10, CIFAR-100 and Tiny-ImageNet. The empirical curves closely follow with the predicted trends, indicating that our linear analysis captures phenomena that persist in modern deep continual learning models.

## 1 Introduction

Intelligent systems need to acquire, update, and accumulate knowledge throughout their lifecycle to adapt to the dynamically changing real world, a capability known as continual learning (Thrun & Mitchell, 1995; Schlimmer & Fisher, 1986). Typically, continual learning machines are challenged by catastrophic forgetting (McCloskey & Cohen, 1989; Goodfellow et al., 2013; Ramasesh et al., 2021), where performance on previous tasks degrades dramatically due to parameter updates when learning new tasks. As new knowledge replaces previous knowledge, the model's adaptation performance improves while memorability diminishes (Abraham & Robins, 2005; Lin et al., 2022; Kim et al., 2023). Earlier efforts have attempted to address this problem by preserving previously learned knowledge (Lopez-Paz & Ranzato, 2017; Yan et al., 2021; Sun et al., 2023). However, recent work has focused more on facilitating the adaptability of new knowledge and the generalizability of models (Raghavan & Balaprakash, 2021; Simon et al., 2022; Lin et al., 2023). These efforts have deepened the understanding of continual learning: an ideal continual learning learner should strike an effective balance between retaining previous learned knowledge and acquiring new knowledge, while also being sufficiently predictable to accommodate differences in unseen data distributions.

In biological systems, hippocampal replay (Davidson et al., 2009; Mallory et al., 2025) has been proposed as a system-level mechanism that consolidates memories and improves the generalization by reactivating previously experienced scenes. Although biological and artificial systems differ significantly, they exhibit intriguing parallels: both consolidate knowledge and accelerate learning from past experiences (Shin et al., 2017; Van de Ven et al., 2020; Shi et al., 2025). Similar to biological systems, the rehearsal mechanism performs better in continual learning to resist catastrophic forgetting (Castro et al., 2018; Tiwari et al., 2022; Gao & Liu, 2023; Bellitto et al., 2024; Van de Ven et al., 2020). Despite recent advances in the empirical performance of rehearsal-based continual learning, the

Figure 1: Continual learning addresses sequential tasks by progressively learning a unified model, and an ideal continual learning system should strike a delicate balance among the adaptation of newly acquired knowledge, the memorization of previously learned knowledge, and the generalization of unseen data distributions.

theoretical understanding of how the rehearsal mechanism impacts continual learning, even in simple models, is not yet fully understood: How is knowledge of previously learned tasks consolidated and learned through playback? When does a continual learning model benefit from rehearsal mechanism? Will replay samples potentially adversely affect generalization performance of the model?

In this paper, we attempt to theoretically understand rehearsal mechanisms in continual learning by answering the above questions. We establish a theoretical characterization of rehearsal-based continual learning within a linear regression framework, taking into account key factors such as rehearsal size, parameter size, and optimal parameter similarity. Furthermore, we provide a unified three-dimensional closed-form expression that captures the model's adaptability, memorability, and generalizability. Specifically, our main contributions can be summarized as follows.

- We derive explicit expressions for memory, adaptation, and generalization errors for rehearsal-based continual learning under both underparameterized and overparameterized regimes, providing insights into key factors and their impact on performance.

- We demonstrate that increasing rehearsal size does not always lead to better performance, as additional rehearsal samples can impair adaptation in the overparameterized regime, while there is a decreasing floor in memory error as rehearsal size increases.

- We conducted numerical simulations and deep neural network experiments to validate and extend our theoretical findings. The results on the effects of rehearsal size, parameter size, and optimal parameter similarity are also consistent with our analysis.

## 2 RELATED WORK

Prior research on catastrophic forgetting in continual learning has primarily focused on empirical studies. These methods can be broadly categorized into three main categories: rehearsal-based methods (Lopez-Paz & Ranzato, 2017; Van de Ven et al., 2020; Bellitto et al., 2024), where a portion of the previous task data is stored and replayed to mitigate forgetting while learning new tasks. Expansion-based methods (Rusu et al., 2016; Gao et al., 2022; Douillard et al., 2022; Wang et al., 2024b) allocate separate network parameters to learn new tasks without interfering with previously learned ones, and recent research leverages pre-trained models with lightweight adaptations such as prompting or adapters (Zhou et al., 2025; Zhang et al., 2023; Wang et al., 2022c; Zhou et al., 2024). Regularization-based methods (Kirkpatrick et al., 2017; Akyürek et al., 2021; Song et al., 2023) constrain the parameters crucial to previous tasks when learning new tasks.

Recent theoretical studies on continual learning mainly focus on elucidating its dynamic evolutionary mechanisms through diverse frameworks and establishing links to related domains (Wang et al., 2024a; Evron et al., 2023; Zhao et al., 2024b). Kim et al. (2022) reformulates continual learning as a combination of within-task prediction and task-id prediction, with links to out-of-distribution detection. Peng et al. (2023) proposes a general formulation of ideal continual learning, linking it to related areas and providing generalization bounds for replayed samples. Lin et al. (2023) focuses on regularization-based continual learning estimators and demonstrates the phenomenon of benign

overfitting in continual learning. Connections between continual learning and alternating projections or Kaczmarz methods have also been explored, leading to worst-case forgetting bounds (Evron et al., 2022). The trade-off between forgetting and generalization is also modeled as a two-player game solved via dynamic programming (Raghavan & Balaprakash, 2021). Li et al. (2023) studies continual ridge regression with non-random features, focusing on the role of regularization parameters.

The most relevant work is Banayeeanzade et al. (2024), as both examine rehearsal-based continual learning methods. However, there are two key differences. Firstly, their work focuses on network width under overparameterization, highlighting the benefits of model dimensionality for multi-task and continual learning. In contrast, we focus on achieving a balance among adaptability, memorability, and generalizability under different parameterization regimes, thereby uncovering several interesting phenomena. Secondly, they assume an infinitely expanding memory buffer, which is unrealistic under storage constraints. In contrast, we consider a fixed-capacity memory with proportional sampling, enabling explicit analysis of error dynamics and prediction across arbitrary task numbers.

Recent studies have explored model performance under different sampling or rehearsal strategies (Deng et al., 2025; Zheng et al., 2024) and the effects of step size and network width (Ding et al., 2024; Goldfarb & Hand, 2023). While some of these studies also use the linear Gaussian model, they focus on different aspects of continual learning. Recent work [Deng et al., 2025] studied replay strategies in continual learning, showing that sequential replay outperforms concurrent replay when tasks are dissimilar, and the Appendix E provides a more detailed comparison. In contrast, our work examines the dynamic equilibrium of rehearsal across three dimensions and its behavior under underparameterized and overparameterized regimes. We show that increased rehearsal size does not always improve memorability, as a lower bound exists for error reduction, and that rehearsal affects adaptability differently depending on the parameterization regime. To our knowledge, this theoretical insight remains unexplored in prior literature. We further validated these findings through numerical simulations and deep neural network experiments on multiple real-world datasets.

## 3 PRELIMINARIES

**Data.** We consider a standard continual learning problem where tasks are introduced sequentially, indexed by $t = 1, 2, ..., T$. Suppose that each task $t$ holds a dataset $\mathcal{D}_t = \{(x_{t,i}, y_{t,i}) \in \mathbb{R}^p \times \mathbb{R}\}_{i=1}^{n_t}$, where $n_t$ denotes its sample size. Here, $x_{t,i}$ denotes the feature vector and $y_{t,i}$ denotes the corresponding response variable. Assume that $\{(x_{t,i}, y_{t,i})\}_{i=1}^{n_t}$ are $i.i.d$ sampled from a linear regression model, meaning each pair $(x_{t,i}, y_{t,i})$ follows the linear model $y_t = \boldsymbol{x}_t^\top \boldsymbol{w}_t^* + \epsilon_t$, where $\epsilon_t$ is random noise and $\boldsymbol{w}_t^*$ represents the optimal parameter of the $t$-th task specific model. The equation above can be rewritten into a compact matrix equation for training samples:

$$\boldsymbol{y}_t = \boldsymbol{X}_t^\top \boldsymbol{w}_t^* + \boldsymbol{\epsilon}_t, \tag{1}$$

where $\boldsymbol{X}_t := [x_{t,1}, x_{t,2}, ..., x_{t,n_t}] \in \mathbb{R}^{p \times n_t}$, $\boldsymbol{y}_t := [y_{t,1}, y_{t,2}, ..., y_{t,n_t}]^\top \in \mathbb{R}^{n_t}$, and $\boldsymbol{\epsilon}_t := [\epsilon_{t,1}, \epsilon_{t,2}, ..., \epsilon_{t,n_t}]^\top \in \mathbb{R}^{n_t}$. For analytical tractability, we adopt Gaussian features and noise, as formally stated in the following assumption.

**Assumption 1.** *For all $t \in T$, each element of $\boldsymbol{X}_t$ follows i.i.d standard Gaussian $\mathcal{N}(0, 1)$. Similarly, the noise $\boldsymbol{\epsilon}_t$ is independently drawn from Gaussian $N(0, \sigma_t^2 I_{n_t})$, where $\sigma_t \geq 0$ denotes noise level.*

In Assumption 1, the Gaussian model is used to exploit the favorable properties of orthogonal projection matrices (Raventós et al., 2023; Li et al., 2025), and the work of Li et al. (2023) assumes fixed features. As we will demonstrate, in the random design setting, our analysis reveals interesting statistical properties of rehearsal-based continual learning methods that were previously unknown.

**Assumption 2.** *For all $t \in T$, the sample size satisfies $n_t = n$, and the noise level satisfies $\sigma_t = \sigma$.*

In Assumption 2, each task has the same number of training samples and the same noise level. This simplification facilitates our analysis, making the theoretical results more interpretable. Furthermore, our analysis can be extended to scenarios where Assumption 2 does not hold.

Note that we focus on rehearsal-based continual learning from the perspective of the linear Gaussian model, as analyzing this model provides a critical first step toward understanding deep neural networks, as shown in recent studies (Evron et al., 2022; Ji et al., 2023; Lin et al., 2023). And

these theoretical insights are further extended through deep neural network experiments on multiple real-world datasets, incorporating longer task sequences and deeper architectures in Section 5.

**Evaluation metrics.** Our goal is to estimate $\boldsymbol{w}_t^*$ in continual learning setting. For any estimator $\hat{\boldsymbol{w}}$, we denote its estimation error by $\mathcal{L}(\hat{\boldsymbol{w}}) = \|\hat{\boldsymbol{w}} - \boldsymbol{w}^*\|^2$. Based on $\mathcal{L}(\hat{\boldsymbol{w}})$, the adaptation error($\mathcal{A}$), generalization error($\mathcal{G}$) and memory error($\mathcal{M}$) can be defined respectively as

$$\mathcal{A}\left(\hat{\boldsymbol{w}}_t, t = 1, \ldots, T\right) := \|\hat{\boldsymbol{w}}_t - \boldsymbol{w}_t^*\|^2, \tag{2}$$

$$\mathcal{G}\left(\hat{\boldsymbol{w}}_t, t = 1, \ldots, T\right) := \frac{1}{t} \sum_{i=1}^{t} \|\hat{\boldsymbol{w}}_t - \boldsymbol{w}_i^*\|^2, \tag{3}$$

$$\mathcal{M}\left(\hat{\boldsymbol{w}}_t, t = 1, \ldots, T\right) := \frac{1}{t-1} \sum_{i=1}^{t-1} \left[\|\hat{\boldsymbol{w}}_t - \boldsymbol{w}_i^*\|^2 - \|\hat{\boldsymbol{w}}_i - \boldsymbol{w}_i^*\|^2\right], \tag{4}$$

for each $t \in T$, where $\hat{\boldsymbol{w}}_t$ denotes the parameters of the continual learning algorithm after task t has been learned. A continual learning algorithm obtains increasing performance on previous tasks if, for each $t \in T$, the forgetting measure satisfies $\mathcal{M}\left(\hat{\boldsymbol{w}}_t\right) < 0$. Consigering that parameter differences in linear models reflect functional differences, we use the distance between optimal parameters to measure task similarity, consistent with the metric in (Evron et al., 2022; Zhao et al., 2024b).

**Rehearsal-based Continual Learning Estimator.** The rehearsal-based continual estimator assumes tasks arrive sequentially and preserves knowledge of previous tasks by storing a subset of their samples (Parisi et al., 2019; De Lange et al., 2021; Rolnick et al., 2018; Wang et al., 2022a; Jeeveswaran et al., 2023). For each task $t = 2, ..., T$, assume that a total of $s$ samples are stored. Specifically, we assume that the feature vector matrix of the $i$-th previous task ( $i = 1, 2, ..., t - 1$ ) stored in the memory buffer is $\boldsymbol{Z}_i \in \mathbb{R}^{p \times \frac{s}{t-1}}$, with the corresponding response variable denoted as $\boldsymbol{g}_i \in \mathbb{R}^{\frac{s}{t-1}}$. The training process converges to the optimal solution by minimizing the training loss, formulated as the following optimization problem: $\widehat{\boldsymbol{w}}_t^{(\text{Reh})} := \arg\min_{\boldsymbol{w}} \|\boldsymbol{X}_t^\top \boldsymbol{w} - \boldsymbol{y}_t\|^2 + \sum_{i=1}^{t-1} \|\boldsymbol{Z}_i^\top \boldsymbol{w} - \boldsymbol{g}_i\|^2$. When $p > n + s$ (overparameterized), multiple solutions exist that achieve zero training loss. In this case, we select the solution with the minimum $\ell_2$-norm , i.e., the optimization problem: $\arg\min_{\boldsymbol{w}} \left\{\|\boldsymbol{w} - \boldsymbol{w}_{t-1}\|^2, \text{s.t.}(\boldsymbol{X}_t)^\top \boldsymbol{w} = \boldsymbol{y}_t, (\boldsymbol{Z}_i)^\top \boldsymbol{w} = \boldsymbol{g}_i, i = 1, \ldots, t - 1\right\}$. Among all overfitting solutions, we focus on the minimum $\ell_2$-norm solution and demonstrate that, in continual learning, it corresponds to the convergence point of stochastic gradient descent or gradient descent ( proven in Appendix A ). In Section 4, we provide theoretical results for rehearsal-based continual learner.

**Striking a balance among adaptability, memorability, and generalizability.** The adaptation error quantifies the model's fitting performance for the current task, while the memory error measures the extent to which the model's performance on previous tasks deteriorates after learning a new task. Complementary to these, the generalization error assesses the model's ability to generalize to new tasks. An ideal continual learning learner should strike a balance between adaptability, memorability, and generalizability. We further explore the connection between these three aspects in Section 4.

## 4 MAIN RESULTS FOR REHEARSAL-BASED CONTINUAL LEARNING

In this section, we present the main results. For rehearsal-based continual learning methods, we establish three theorems that characterize the adaptation error, memory error, and generalization error of the model under both overparameterized and underparameterized regimes.

**Theorem 1** (Adaptation error). *Suppose that Assumption 1 and Assumption 2 hold. Then the adaptation error of the rehearsal-based continual learning model is formally given by*

$$\mathbb{E}[\mathcal{A}(\widehat{\boldsymbol{w}}_T)] = \begin{cases} \lambda^T \|\boldsymbol{w}_T^*\|^2 + \underbrace{\sum_{k=1}^{T} \lambda^{T-k} \frac{n+s}{p} \|\boldsymbol{w}_k^* - \boldsymbol{w}_T^*\|^2}_{\text{Term } A1} + a_{noise}, & \text{for } p > n+s+1, \tag{5} \\ \frac{p\sigma^2}{n+s-p-1}, & \text{for } n+s > p+1. \tag{6} \end{cases}$$

*where $\lambda := \frac{p-n-s}{p}$ and $a_{noise} := \frac{(1-\lambda^T)p\sigma^2}{(p-n-s-1)}$, with larger $\lambda$ indicating greater overparameterization.*

The proof is provided in Appendix B. It describes the model's ability to fit the current task, forming the basis for analyzing how it learns new knowledge and retains previous knowledge.

**Increasing the rehearsal size enhances the model's adaptation ability under underparameterization, whereas it can be detrimental under overparameterization.** Specifically, when $n + s > p + 1$ in Equation (6), $\mathbb{E}[\mathcal{A}(\widehat{\boldsymbol{w}}_T)]$ decreases as $s$ increases, indicating that more playback samples contribute to better adaptation performance. When slightly overparameterized in Equation (5), we have $p \approx n + s$ and thus $\lambda \approx 0$. At this point, Term A1 and the denominator in Term $a_{noise}$ approach zero when tasks are similar, and thus $a_{noise}$ dominates and causes $\mathbb{E}[\mathcal{A}(\widehat{\boldsymbol{w}}_T)]$ to be increasing w.r.t. $s$. When heavily overparameterized in Equation (5), Term A1 is close to zero, and thus $\mathbb{E}[\mathcal{A}(\widehat{\boldsymbol{w}}_T)]$ decreases as $s$ increases when the $\sigma$ is low. Intuitively, when tasks are similar, the model can leverage replay samples more effectively, leading to improvements in performance on the current task.

The impact of rehearsal size was also verified through numerical simulations in Figure 2(b), where average adaptation error is plotted against rehearsal size for different model parameters. The red curve marked with "×", decreases in the underparameterized regime ($s > p - n$) but first decreases and then increases in the overparameterized regime ($s < p - n$), which validates our earlier insights.

**Remark 1 (Turning point.)** Based on the above analysis, it can be observed that under overparameterization, the adaptation error exhibits a non-monotonic trend with respect to the rehearsal size $s$. Furthermore, we compute the derivative to identify its turning point.

$$\frac{\partial \mathbb{E}[A(w_2)]}{\partial s} = -\frac{2\lambda}{p} \|w_2^*\|^2 + \frac{2\lambda - 1}{p} \|w_2^* - w_1^*\|^2 + \frac{2\lambda \cdot (p - n - s - 1) + p \cdot (\lambda^2 - 1)}{(p - n - s - 1)^2} \cdot \sigma^2$$

Setting the derivative to zero, we can calculate that the inflection point $s$ occurs near 2100 when $n = 1000$, $\sigma = 0.02$, and $p = 3000$. Furthermore, through computer simulation, we determine that the inflection point at $T = 8$ occurs near 1000, consistent with the curve shown in Figure 2(b).

Beyond analyzing the impact of rehearsal size, we also examined the effects of inter-task similarity and model parameters. We found that under overparameterization, models require higher inter-task similarity to better adapt to the current task, whereas this does not hold in underparameterized settings (Figure 2(c)). Moreover, the overparameterization helps mitigate the impact of task variability and noise effects on model adaptability, as illustrated in Figure 2(a) and Figure 2(c). Due to space constraints, a more detailed discussion of these factors is provided in Appendix G.

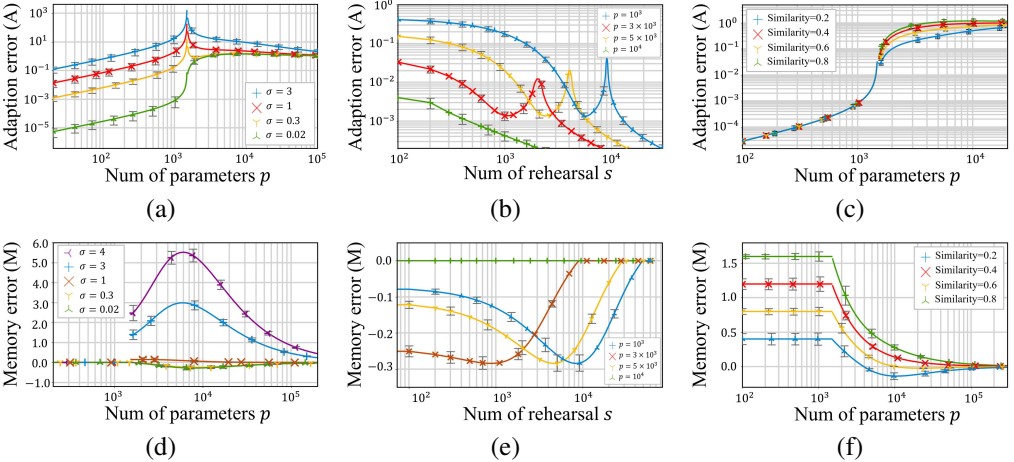

Figure 2: Adaptation performance and memory performance of rehearsal-based continual learning under different setups, where $T = 8$, $n = 1000$ and $\|\boldsymbol{w}_t^*\|^2 = 1$ for all $t \in T$. The discrete points indicated by markers are calculated by simulation and are the average of 100 random simulation runs. Additional settings for each subfigure are as follows: (a) and (d) : $s = 500$; (b) and (e) : $\sigma = 0.02$; (c) and (f) : $s = 500$, $\sigma = 0.02$.

**Theorem 2** (Memory error). *Under Assumption 1 and Assumption 2, the memory error of the rehearsal-based continual learning model is formally given by*

$$
\mathbb{E}[\mathcal{M}(\widehat{\boldsymbol{w}}_T)] =
\begin{cases}
\underbrace{\dfrac{1}{T-1}\sum_{k=1}^{T-1}\sum_{j>k}^{T}\dfrac{n+s}{p}u_{kj}\left\|\boldsymbol{w}_j^*-\boldsymbol{w}_k^*\right\|^2}_{Term\ M1} \\
\qquad\qquad\qquad\qquad\qquad\qquad\qquad\qquad\qquad \text{for } p>n+s+1,\ (7) \\
+\underbrace{\dfrac{1}{T-1}\sum_{i=1}^{T-1}\left(\lambda^T-\lambda^i\right)\left\|\boldsymbol{w}_i^*\right\|^2}_{Term\ M2}+m_{noise}, \\[2mm]
\dfrac{1}{T-1}\sum_{k=1}^{T-1}\left\|\boldsymbol{w}_T^*-\boldsymbol{w}_k^*\right\|^2, \qquad\qquad\qquad \text{for } n+s>p+1.\ (8)
\end{cases}
$$

*where $u_{kj} := \lambda^{T-k}-\lambda^{j-k}+\lambda^{T-j}$ and $m_{noise} := \frac{1}{T-1}\sum_{i=1}^{T-1}\frac{p\sigma^2}{p-n-s-1}\left(\lambda^i-\lambda^T\right)$. Specifically, when the number of tasks $T=2$, the Equation (7) can be reformulated as*

$$
\mathbb{E}[\mathcal{M}(\widehat{\boldsymbol{w}}_2)] = \frac{n+s}{p}\left\|\boldsymbol{w}_2^*-\boldsymbol{w}_1^*\right\|^2 - \frac{(n+s)(p-n-s)}{p^2}\left\|\boldsymbol{w}_1^*\right\|^2 + \frac{(n+s)(p-n-s)\sigma^2}{(p-n-s-1)p}, \quad (9)
$$

*Similarly, the corresponding case in Equation (8) can be reformulated as*

$$
\mathbb{E}[\mathcal{M}(\widehat{\boldsymbol{w}}_2)] = \left\|\boldsymbol{w}_2^*-\boldsymbol{w}_1^*\right\|^2. \tag{10}
$$

The detailed proof is provided in Appendix C. Based on Theorems 2, we further explore their analytical insights and examine the influence of factors such as rehearsal size, as well as the performance differences observed under both overparameterized and underparameterized regimes.

**Increasing the rehearsal size does not always lead to better memory performance in continual learning models in the overparameterized regime.** Specifically, we consider the case where $T=2$. For the overparameterized regime result in Equation (9), when tasks are similar and $\sigma$ is low, the second term dominates and causes $\mathbb{E}[\mathcal{M}(\widehat{\boldsymbol{w}}_T)]$ first decreases and then increases as $s$ increases, indicating the existence of decreasing floor. For the underparameterized regime result in Equation (10), rehearsal no longer contributes to memory performance. In this situation, the $\mathbb{E}[\mathcal{M}(\widehat{\boldsymbol{w}}_T)]$ depends solely on the inherent similarity between tasks (i.e. $\left\|\boldsymbol{w}_T^*-\boldsymbol{w}_k^*\right\|^2$ ), meaning that the memory error is fully determined by similarity between the final task and preceding ones.

In Figure 2(e), the yellow curve marked ″ Y ″ clearly illustrates how the average memory error varies with rehearsal size when $p=3\times10^4$. In the overparameterized regime ($s<p-n$), the memory error first decreases and then increases, indicating the existence of a decreasing performance floor. In contrast, in the underparameterized regime ($s>p-n$), the error remains unaffected by rehearsal size, and zero forgetting is achieved when the task-optimal parameters remain consistent.

**Remark 2 (Turning point.)** For the turning point of memory error with respect to rehearsal $s$ under overparameterization, we compute the derivative of memory error with respect to s for T=2 due to the ease of analysis.

$$
\frac{\partial\mathbb{E}[M(w_2)]}{\partial s} = \frac{1-2\lambda}{p}\cdot\|w_1^*\|^2 + \frac{1}{p}\cdot\|w_2^*-w_1^*\|^2 + \left(\frac{2\lambda-1}{p-n-s-1} + \frac{p\cdot(\lambda-\lambda^2)}{(p-n-s-1)^2}\right)\cdot\sigma^2
$$

Setting the derivative to zero yields that when $n=1000$, $\sigma=0.02$, and $p=3000$, the inflection point $s$ occurs near 500. Furthermore, through computer simulation, we determine that the inflection point at $T=8$ lies near 4600, consistent with the curve shown in Figure 2(e).

Apart from rehearsal size, we also examined additional factors. For example, increasing task similarity enhances memory performance under underparameterization but can have adverse effects under overparameterization. In the overparameterized regime, parameter size influences memory performance more strongly than noise (Figure 2(d)). Further analysis is provided in Appendix G.

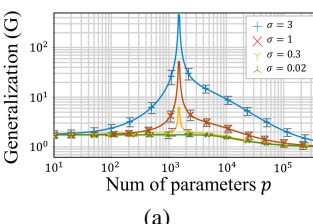 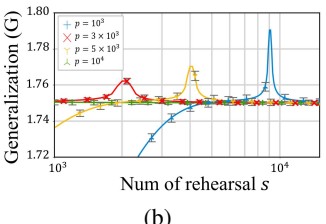 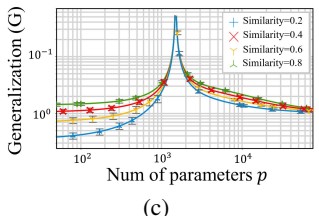

(a)  (b)  (c)

Figure 3: The trend of average generalization error w.r.t. the number of model parameters or rehearsal samples, with $T = 8$, $n = 1000$ and $\|\boldsymbol{w}_t^*\|^2 = 1$ for all $t \in T$. Discrete points denote averages over 100 random simulations for each setting. Subfigure settings: (a) : $s = 500$; (b) : $\sigma = 0.02$; (c) : $s = 500$, $\sigma = 0.02$.

**Theorem 3** (Generalization error). *Under Assumption 1 and Assumption 2, the generalization error of the rehearsal-based continual learning model is formally given by*

$$
\mathbb{E}[\mathcal{G}(\widehat{\boldsymbol{w}}_T)] =
\begin{cases}
\underbrace{\dfrac{1}{T}\sum_{k=1}^{T}\sum_{j=1}^{T}\dfrac{n+s}{p}\lambda^{T-k}\left\|\boldsymbol{w}_k^* - \boldsymbol{w}_j^*\right\|^2}_{\text{Term } G1} & \\[2mm]
\quad + \underbrace{\dfrac{1}{T}\sum_{k=1}^{T}\lambda^T\left\|\boldsymbol{w}_k^*\right\|^2}_{\text{Term } G2} + g_{noise}, & \text{for } p > n+s+1, \quad (11) \\[6mm]
\dfrac{1}{T}\sum_{k=1}^{T}\left\|\boldsymbol{w}_T^* - \boldsymbol{w}_k^*\right\|^2 + \dfrac{p\sigma^2}{n+s-p-1}, & \text{for } n+s > p+1, \quad (12)
\end{cases}
$$

*where $g_{noise}(\widehat{\boldsymbol{w}}_T) := \dfrac{p\sigma^2}{p-n-s-1}\left(1-\lambda^T\right)$. Specifically, when the number of tasks $T = 2$, we reformulate Equation (11) to provide a clearer interpretation of the error form , resulting in*

$$
\mathbb{E}[\mathcal{G}(\widehat{\boldsymbol{w}}_2)] = \frac{1}{2}(1-\lambda^2)\left\|\boldsymbol{w}_2^* - \boldsymbol{w}_1^*\right\|^2 + \frac{1}{2}\lambda^2(\left\|\boldsymbol{w}_1^*\right\|^2 + \left\|\boldsymbol{w}_2^*\right\|^2) + \frac{p\sigma^2(1-\lambda^2)}{p-n-s-1}, \quad (13)
$$

*Similarly, in the case when the number of tasks $T = 2$, we can reformulate Equation (12) as*

$$
\mathbb{E}[\mathcal{G}(\widehat{\boldsymbol{w}}_2)] = \frac{1}{2}\left\|\boldsymbol{w}_2^* - \boldsymbol{w}_1^*\right\|^2 + \frac{p\sigma^2}{n+s-p-1}. \quad (14)
$$

**Increasing rehearsal size can degrade generalization performance under overparameterization, especially when tasks are dissimilar.** Consider the case where $T = 2$. When slightly overparameterized in Equation (13), the second term approaches zero, the denominator $p - n - s - 1$ in the third term approaches zero, and thus this term dominates. In this situation, increasing the rehearsal size raises the peak generalization error. In contrast, when heavily overparameterized in Equation (13), the second term dominates and decreases as $s$ increases when tasks are similar and $\sigma$ is low. Moreover, the $\mathbb{E}[\mathcal{G}(\widehat{\boldsymbol{w}}_T)]$ decreases as $s$ increases in Equation (14), indicating that larger rehearsal size consistently enhances generalization performance under underparameterization.

In Figure 3(b), average generalization error varies with rehearsal size under different model parameters, with optimal parameters being orthogonal. The yellow curve with markers " Y " decreases when underparameterized ($s > p - n$), but increases with rehearsal size when overparameterized ($s < p - n$), confirming these insights. Additionally, we examined other factors under different parameterization regimes. As shown in Figure 3(c), increasing similarity enhances generalization in both underparameterized and overparameterized settings. However, enlarging the parameter size reduces the influence of rehearsal and inter-task similarity on generalization (Figures 3(a), 3(c)).

Noted that in Figures 2–3, the logarithmic axes are used to capture variations in error under different parameterization regimes. The horizontal axis illustrates the transition from underparameterization to overparameterization, allowing an intuitive comparison, while the vertical axis highlights the differing

impact of noise levels. It is worth noting that logarithmic scaling provides a wider coordinate range while preserving the original trends. Similar configurations have been used in (Evron et al., 2022; Li et al., 2023; Zhao et al., 2024b) to reveal subtle variations in model error.

In conclusion, we derive expressions for adaptation, memory, and generalization errors, and analyze key factors. The following proposition reveals the connection among adaptability, memorability and generalizability for $T = 2$, and presents conditions for effective generalization performance.

**Proposition 1.** *Assuming that Assumptions 1 and 2 hold. For $T = 2$, the generalization error $\mathbb{E}[\mathcal{G}(\widehat{\boldsymbol{w}}_T)]$ increases with the error on initial task when $\|\boldsymbol{w}_2^* - \boldsymbol{w}_1^*\|$ is low. Minimizing error requires small memory and adaptation errors while maintaining performance on the initial task.*

The detailed proof is provided in Appendix D. As indicated by Proposition 1, better generalization performance requires excelling at the current task while retaining knowledge from previous tasks. In addition, performance on the initial task is also crucial, consistent with the empirical analyses by Shi et al. (2022); Wang et al. (2024a). From the perspective of model's memorability, failing to learn the initial task well can lead to error accumulation if knowledge retention is overemphasized.

## 5 EMPIRICAL VALIDATION ON DEEP NEURAL NETWORKS

Thus far, we have explored different aspects influencing the performance of rehearsal-based continual learning. To validate whether our theoretical insights from linear models under overparameterization extend to deep neural networks, we conduct experiments on real datasets. After training each task, adaptation, memory, and generalization errors were evaluated. The experiments were conducted on MNIST (LeCun et al., 1989) , CIFAR-10 (Krizhevsky et al., 2009) , CIFAR-100 (Krizhevsky et al., 2009), and Tiny-ImageNet (Le & Yang, 2015). All experiments were repeated at least three times, and average results are reported, with additional experimental details provided in Appendix H.

**More rehearsal is not always better for memory performance**. The impact of rehearsal size on memory error is illustrated in Figures 4(e)–(f) and Figure 5(a). The MNIST, CIFAR-10, CIFAR-100, and Tiny-ImageNet datasets were partitioned into 2, 2, 10, and 20 tasks, respectively, with each task containing 5, 5, 10, and 20 categories. As shown in the figures, the memory error initially decreases but then increases as the rehearsal size grows, with this effect being more pronounced when class overlap is two. These observations suggest that larger rehearsal sizes do not always lead to better performance, and that further gains become marginal once rehearsal reaches a certain level.

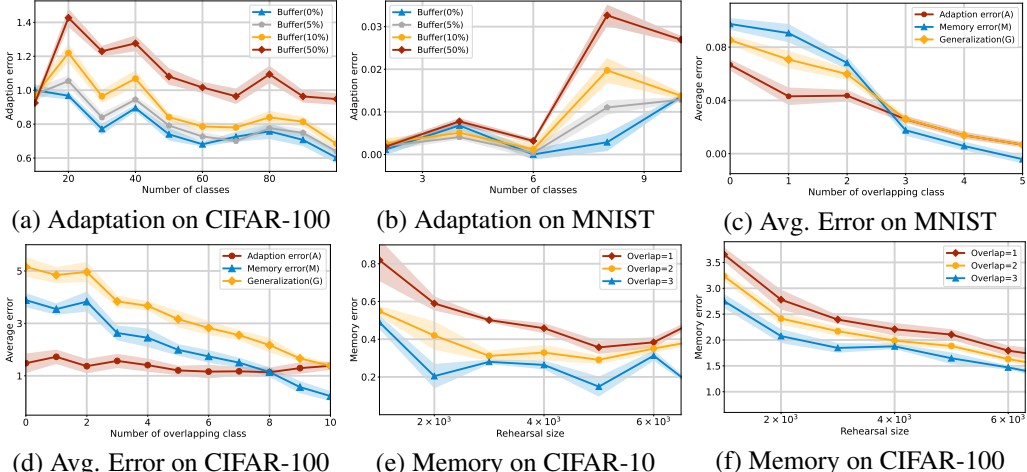

Figure 4: Impact of rehearsal size and task similarity on adaptation, memory, and generalization errors in deep neural networks trained on MNIST, CIFAR-10 and CIFAR-100. Subfigures (a)-(b) show how accumulated classes affect adaptation error under different buffer sizes; (c)-(d) illustrate how overlapping classes affect all three errors; (e)-(f) illustrate how rehearsal size affects memory error for varying similarity levels.

**Different sampling strategies**. To evaluate sampling strategies, we compared model errors under Random, Herding, and Reservoir methods (Table 1). Herding achieved the lowest error by selecting samples near class centers, offering better representation of previous tasks.

**Various network architectures**. We further analyzed performance across different architectures (Table 2). Deeper networks yielded lower average errors, especially in memory and generalization, suggesting that larger models capture richer features and better mitigate forgetting.

**Longer training sequences**. The analysis was further extended to the Tiny-ImageNet dataset with more training tasks shown in Table 5. The results indicate that adaptation error increases with larger rehearsal size, highlighting the adverse effect of the rehearsal mechanism.

Table 1: The adaptation error across increasing buffer sizes under various sampling strategies on CIFAR-10.

| Methods | 0% | 5% | 10% | 50% |
|---|---|---|---|---|
| Random | 0.196 | 0.212 | 0.257 | 0.382 |
| Reservoir | 0.201 | 0.222 | 0.289 | 0.613 |
| Herding | 0.196 | 0.203 | 0.207 | 0.218 |
| GSS | 0.193 | 0.201 | 0.215 | 0.483 |
| CSReL | 0.198 | 0.203 | 0.205 | 0.213 |

Table 2: Average adaptation, memory, and generalization errors across different architectures on CIFAR-10.

| Arch. | Adapt. | Memo. | Gener. |
|---|---|---|---|
| CNN | 0.21 | 3.60 | 2.28 |
| ResNet18 | 0.18 | 2.40 | 1.48 |
| ResNet50 | 0.17 | 2.17 | 1.44 |

| Methods | Forg. 1000 | Diff. 1000 | Forg. 2000 | Diff. 2000 | Forg. 3000 | Diff. 3000 |
|---|---|---|---|---|---|---|
| EWC(+Rehearsal) | 61.744 | - | 49.513 | 12.232 | 42.356 | 7.157 |
| LwF(+Rehearsal) | 40.256 | - | 34.567 | 5.689 | 31.533 | 3.033 |
| iCaRL | 59.744 | - | 46.356 | 13.389 | 40.844 | 5.511 |
| DER | 23.467 | - | 16.656 | 6.811 | 12.522 | 4.133 |
| FOSTER | 18.589 | - | 27.156 | -8.567 | 30.267 | -3.111 |
| MEMO | 25.344 | - | 19.111 | 6.233 | 14.956 | 4.156 |
| CSReL | 56.733 | - | 44.689 | 12.044 | 31.411 | 13.278 |

| Methods | Forg. 4000 | Diff. 4000 | Forg. 5000 | Diff. 5000 | Forg. 6000 | Diff. 6000 |
|---|---|---|---|---|---|---|
| EWC(+Rehearsal) | 37.633 | 4.722 | 33.322 | 4.311 | 30.689 | 2.633 |
| LwF(+Rehearsal) | 30.622 | 0.911 | 28.422 | 2.200 | 26.367 | 2.056 |
| iCaRL | 35.811 | 5.033 | 31.856 | 3.956 | 31.013 | 0.843 |
| DER | 10.322 | 2.200 | 8.489 | 1.833 | 7.456 | 1.033 |
| FOSTER | 31.456 | -1.189 | 31.090 | 0.366 | 30.811 | 0.279 |
| MEMO | 12.700 | 2.256 | 10.200 | 2.500 | 8.744 | 1.456 |
| CSReL | 27.822 | 3.589 | 27.833 | -0.011 | 27.439 | 0.394 |

Table 3: Comparison of forgetting rates between continual learning baselines on CIFAR100.

As shown in the table 3, the forgetting rate decreases for most algorithms as the replay size increases, but the rate of decrease gradually slows as the replay scale grows larger. When the number of replays reaches around 6000, the gain in forgetting resistance is only about 0.3% for some algorithms. Given the negative impact of rehearsal on current task performance, more effective techniques are needed to address this diminishing return and performance degradation.

Table 4: Comparison of model errors and traditional metrics with increasing training tasks on CIFAR-10.

| Training Tasks | T1 | T2 | T3 | T4 | T5 |
|---|---|---|---|---|---|
| Adaptation Error | 0.13±0.00 | 0.40±0.00 | 0.28±0.00 | 0.11±0.00 | 0.16±0.00 |
| Memory Error | – | 1.54±0.03 | 3.90±0.05 | 4.24±0.16 | 4.66±0.07 |
| Generalization Error | 0.13±0.00 | 1.04±0.01 | 2.87±0.04 | 3.41±0.12 | 3.94±0.05 |
| Forgetting Ratio | – | 7.30±0.63 | 22.35±0.54 | 27.23±0.43 | 38.38±0.14 |
| Average Accuracy | 95.05±0.26 | 65.78±0.34 | 39.20±0.41 | 35.78±0.18 | 32.06±0.33 |

In deep neural network experiments, model performance on continual learning is evaluated using adaptation, memory, and generalization errors rather than accuracy or forgetting rate, consistent with the theoretical analysis. Similar metrics are also used in research by Zhao et al. (2024b); Evron et al.

(2022). Additionally, we also report accuracy and forgetting rate (Table 4). As shown in the table, average accuracy and generalization error reflect overall generalization, with accuracy gradually decreasing as training classes increase. And forgetting rate and memory error indicate the ability to retain previous knowledge, both rising during training to demonstrate the phenomenon of forgetting.

Table 5: Adaptation error across varying buffer sizes on Tiny-ImageNet

| Tasks | T3 | T6 | T12 | T16 | T20 |
|---|---|---|---|---|---|
| 0% | $1.120 \pm 0.025$ | $0.959 \pm 0.027$ | $0.888 \pm 0.023$ | $0.991 \pm 0.022$ | $1.012 \pm 0.014$ |
| 5% | $1.180 \pm 0.019$ | $1.018 \pm 0.030$ | $0.909 \pm 0.024$ | $1.031 \pm 0.003$ | $1.050 \pm 0.019$ |
| 10% | $1.257 \pm 0.037$ | $1.055 \pm 0.022$ | $0.938 \pm 0.021$ | $1.103 \pm 0.017$ | $1.115 \pm 0.013$ |
| 50% | $1.454 \pm 0.033$ | $1.220 \pm 0.051$ | $1.085 \pm 0.069$ | $1.208 \pm 0.039$ | $1.271 \pm 0.023$ |

**Rehearsal mechanism may impair adaptation performance**. To examine the impact of the number of training classes under different buffer sizes, we divided the ten classes in MNIST and CIFAR-10 into five tasks, each containing two classes. The division scheme used in CIFAR-100 and Tiny-ImageNet follows the previous settings. In Figures 4(a)–(b) and Figure 5(b), the adaptation error varies with the number of training classes at different rehearsal sizes. As observed, the error increases with larger rehearsal sizes (e.g., the red curves in the figures). This observation indicates that rehearsal may impair the model's adaptation performance, consistent with our analysis of linear models.

**The impact of task similarity on memory performance is most pronounced**. We regulate task similarity by adjusting the number of overlapping classes between tasks. Beyond class similarity metric, we also evaluate different similarity metrics (Appendix H). As shown in Figures 4(c)–(d), adaptation, memory, and generalization errors all decrease as similarity level increases, with memory error dropping most sharply, reflecting that stronger task similarity enhances knowledge retention.

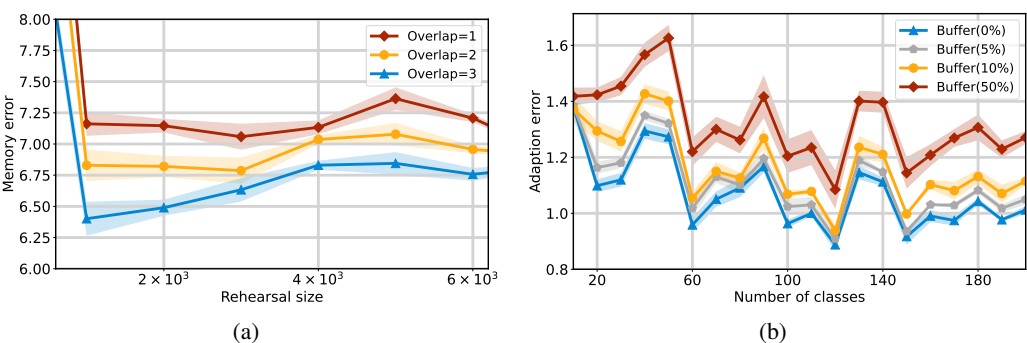

(a)                                                          (b)

Figure 5: Impact of rehearsal size on adaptation error and memory error in deep neural networks trained on Tiny-ImageNet. Subfigures (a) illustrate how rehearsal size affects memory error for varying similarity levels. Subfigures (b) show how accumulated classes affect adaptation error under different buffer sizes.

## 6 CONCLUSION

In this work, we investigate rehearsal-based continual learning under both underparameterized and overparameterized regimes, formulating each task as a linear regression problem. We derive explicit expressions for memory, adaptation, and generalization errors, providing a foundational understanding of rehearsal-based continual learning. In contrast to common views, more rehearsal is not always beneficial; even for mitigating forgetting, there exists a lower bound on error reduction. Moreover, the rehearsal mechanism affects model adaptability differently in underparameterized and overparameterized scenarios. These findings provide valuable insights into the role and limitations of rehearsal mechanism. Furthermore, we validate these theoretical understanding through numerical simulations and further extend the analysis from linear models to deep neural networks. Experiments on MNIST, CIFAR-10, CIFAR-100, and Tiny-ImageNet further explore longer task sequences, deeper architectures, and various sampling strategies, supporting and extending these theoretical insights.

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

# A    ADDITIONAL ANALYSIS AND SUPPORT LEMMAS

## A.1    EXTENDED ANALYSIS ON CIFAR100 BY STRONG CONTINUAL LEARNING BASELINES

| Methods | 2000 | 4000 | 6000 | 8000 | 10000 |
|---------|------|------|------|------|-------|
| EWC | 86.20 | 84.60 | 83.17 | 81.91 | 80.11 |
| LwF | 73.35 | 73.86 | 72.92 | 73.90 | 74.76 |
| iCaRL | 84.59 | 82.98 | 81.59 | 80.89 | 80.19 |
| DER | 73.87 | 73.57 | 72.40 | 72.86 | 72.40 |
| FOSTER | 79.48 | 84.71 | 84.35 | 83.95 | 83.10 |
| MEMO | 68.10 | 66.78 | 66.01 | 65.56 | 65.16 |
| CSReL | 73.29 | 68.20 | 68.13 | 67.96 | 60.59 |

As shown in the table, as the replay scale increases, the average accuracy on the current task decreases for most algorithms, and CSReL exhibits a more pronounced decline, while DER and MEMO decrease more slowly. This may be due to CSReL's use of an additional network to constrain model loss, which can introduce extra interference to current task learning. We then analyze how forgetting rates vary across these methods with increasing rehearsal size. The experimental results are as follows.

## A.2    EXTENDED ANALYSIS ON IMAGENET-R BY STRONG CONTINUAL LEARNING BASELINES

| Methods | 2000 | 4000 | 6000 |
|---------|------|------|------|
| EWC(+Rehearsal) | 53.053 | 51.476 | 49.690 |
| iCaRL | 53.567 | 50.893 | 49.992 |
| DER | 46.211 | 45.839 | 44.209 |
| FOSTER | 29.473 | 40.846 | 44.229 |
| MEMO | 19.256 | 17.608 | 16.218 |

As shown in the table, the model's accuracy on the current task gradually declines as the replay scale increases, demonstrating the negative impact on the model's adaptability.

## A.3    EXTENSION EXPERIMENTS ON THE IMAGENET-R DATASET

| Buffer Size | High similarity | Medium similarity | Low similarity |
|-------------|-----------------|-------------------|----------------|
| 0 | 7.721 | 9.192 | 11.725 |
| 200 | 5.673 | 6.639 | 6.908 |
| 600 | 6.687 | 6.367 | 7.150 |
| 800 | 6.795 | 6.707 | 7.320 |
| 1200 | 7.148 | 6.964 | 7.903 |
| 1600 | 6.917 | 7.105 | 7.580 |
| 2000 | 7.381 | 6.995 | 7.751 |
| Average | 6.903 | 7.138 | 8.048 |

As shown in the table, introducing replay initially improves performance. However, as the number of replays increases, the benefit for memory performance diminishes, and memory errors can even increase. The experimental results comparing the continual learning baselines are as follows.

## A.4    EXTENSION EXPERIMENTS ON THE CUB200 DATASET

| Tasks | T3 | T6 | T12 | T16 | T20 | Avg. |
|-------|-----|-----|-----|-----|-----|------|
| 0% | 2.146 | 2.153 | 2.329 | 2.437 | 2.270 | 2.267 |
| 5% | 2.125 | 1.735 | 2.310 | 2.522 | 2.739 | 2.286 |
| 10% | 2.309 | 2.054 | 4.513 | 5.270 | 5.267 | 3.882 |
| 50% | 2.296 | 2.009 | 2.373 | 2.333 | 2.427 | 2.287 |

As shown in the table, adaptation error on the current task increases as the replay buffer grows, with the largest difference observed when replaying 10% of the samples.

| Buffer Size | High similarity | Medium similarity | Low similarity |
|---|---|---|---|
| 0 | 4.518 | 5.721 | 5.895 |
| 200 | 4.085 | 4.748 | 5.106 |
| 400 | 5.049 | 5.683 | 5.495 |
| 600 | 5.933 | 5.097 | 6.420 |
| 800 | 6.127 | 5.811 | 6.421 |
| 1000 | 6.363 | 5.897 | 6.400 |
| Average | 5.346 | 5.493 | 5.956 |

As shown in the table, memory error exhibited a noticeable decrease when replay samples were initially introduced. However, further increasing the number of replay samples leads to a rise in memory error. Additionally, for high similarity tasks, the model achieves better memory performance.

## A.5    VERIFICATION OF CONVERGENCE OF GRADIENT DESCENT FOR NORMS

| Epoch | Train Loss | Solution Norm | Difference | Relative Error |
|---|---|---|---|---|
| 50 | 4.2465 | 7.3235 | 2.4340 | 29.53% |
| 100 | 0.9057 | 7.8090 | 1.3898 | 16.86% |
| 250 | 0.0706 | 8.1575 | 0.5051 | 6.13% |
| 300 | 0.0390 | 8.1872 | 0.3873 | 4.70% |
| 350 | 0.0226 | 8.2052 | 0.3008 | 3.65% |
| 500 | 0.0051 | 8.2293 | 0.1465 | 1.78% |
| 800 | 0.0003 | 8.2409 | 0.0369 | 0.45% |
| 850 | 0.0002 | 8.2416 | 0.0294 | 0.36% |
| 900 | 0.0001 | 8.2421 | 0.0235 | 0.28% |
| 950 | 0.0001 | 8.2424 | 0.0187 | 0.23% |
| 992(ES) | 0.0001 | 8.2427 | 0.0155 | 0.0019% |
| 1000 | 0.0001 | 8.2427 | 0.0155 | 0.0019% |

As shown by the experimental results, although the norm of the gradient descent solution does not exactly match the minimum-norm solution in continual learning scenario, the gap decreases progressively with more training epochs. The gradient-updated solution continues to converge toward the small-norm trend, and future work will extend this analysis to more complex scenarios.

## A.6    ADAPTIVE ERROR UNDER DIFFERENT SAMPLING STRATEGIES

| Methods | 0 | 5% | 10% | 50% |
|---|---|---|---|---|
| Random | 16.945 | 3.602 | 2.131 | 0.767 |
| Reservoir | 17.415 | 3.314 | 1.549 | 0.386 |
| Herding | 17.255 | 4.816 | 3.624 | 3.389 |
| GSS | 17.748 | 5.085 | 4.035 | 3.810 |
| CSReL | 18.049 | 11.287 | 11.080 | 8.245 |

As shown in the table, under different sampling strategies, increasing the buffer size reduces memory error to some extent, with more performance improvements in the initial stages. As replay size increases, the performance gains gradually diminish. Under the Herding sampling, replaying 50% of the samples reduces the error by only 0.235 compared to replaying 10% of the samples.

| Methods | 0 | 5% | 10% | 50% |
|---|---|---|---|---|
| Random | 11.819 | 2.820 | 1.849 | 0.954 |
| Reservoir | 12.121 | 2.609 | 1.422 | 0.799 |
| Herding | 12.014 | 3.049 | 2.838 | 2.762 |
| GSS | 9.942 | 3.083 | 2.542 | 2.700 |
| CSReL | 10.081 | 6.467 | 6.468 | 4.851 |

As shown in the table, under different sampling strategies, the model's generalization error decreases to varying degrees as the replay size increases. This effect is particularly pronounced for Herding sampling, Random sampling, and Reservoir sampling strategies.

A.7 ANALYSIS OF THEORETICAL ASSUMPTION LIMITATIONS

In continual learning, the Gaussian assumption leverages properties such as orthogonal projection and closed-form solutions to separate factors affecting performance and quantify results. However, it often struggles to capture the nonlinear nature of deep networks and the structural distribution of noise. The fixed design assumption treats the input data matrix as fixed or non-random; analyzing the geometric structure of data across tasks can reveal interference between tasks but neglects the inherent randomness in real data distributions. The fourth-moment assumption accommodates broader data distributions and enables analysis of noise and data perturbations, but it often fails to capture structural noise and does not yield explicit closed-form expressions. Therefore, each assumption has distinct strengths and limitations. Selecting the appropriate theoretical framework for research context is therefore essential to effectively address diverse research topics under continual learning.

A.8 THEORETICAL PROOF UNDER RELAXED ASSUMPTIONS

Under the relaxed Assumption 2, assuming the sample size and noise satisfying $n_1 \neq n_2 \neq ... \neq n_T$ and $\sigma_1 \neq \sigma_2 \neq ... \neq \sigma_T$, the optimization objective under the underparameterized regime is given by: $\widehat{\boldsymbol{w}}_t^{(\mathrm{Reh})} := \arg\min_{\boldsymbol{w}} \left\| \boldsymbol{X}_t^\top \boldsymbol{w} - \boldsymbol{y}_t \right\|^2 + \sum_{i=1}^{t-1} \left\| \boldsymbol{Z}_i^\top \boldsymbol{w} - \boldsymbol{g}_i \right\|^2$. Let $L(\boldsymbol{w}) = \left\| \boldsymbol{X}_t^\top \boldsymbol{w} - \boldsymbol{y}_t \right\|^2 + \sum_{i=1}^{t-1} \left\| \boldsymbol{Z}_i^\top \boldsymbol{w} - \boldsymbol{g}_i \right\|^2$, and we have

$$\left\| \boldsymbol{X}_t^\top \boldsymbol{w} - \boldsymbol{y}_t \right\|^2 + \sum_{i=1}^{t-1} \left\| \boldsymbol{Z}_i^\top \boldsymbol{w} - \boldsymbol{g}_i \right\|^2$$

$$= (\boldsymbol{X}_t^\top \boldsymbol{w} - \boldsymbol{y}_t)^\top (\boldsymbol{X}_t^\top \boldsymbol{w} - \boldsymbol{y}_t) + \sum_{i=1}^{t-1} (\boldsymbol{Z}_i^\top \boldsymbol{w} - \boldsymbol{g}_t)^\top (\boldsymbol{Z}_i^\top \boldsymbol{w} - \boldsymbol{g}_i)$$

$$= \boldsymbol{w}^\top \boldsymbol{X}_t \boldsymbol{X}_t^\top \boldsymbol{w} - \boldsymbol{y}_t^\top \boldsymbol{X}_t \boldsymbol{w} - \boldsymbol{w}^\top \boldsymbol{X}_t \boldsymbol{y}_t - \boldsymbol{y}_t^\top \boldsymbol{y}_t$$

$$+ \sum_{i=1}^{t-1} (\boldsymbol{w}^\top \boldsymbol{Z}_i \boldsymbol{Z}_i^\top \boldsymbol{w} - \boldsymbol{g}_i^\top \boldsymbol{Z}_i \boldsymbol{w} - \boldsymbol{w}^\top \boldsymbol{Z}_i \boldsymbol{g}_i - \boldsymbol{g}_i^\top \boldsymbol{g}_i)$$

By setting the derivative to zero, we can have

$$\boldsymbol{w}_t = (\boldsymbol{X}_t \boldsymbol{X}_t^\top + \sum_{i=1}^{t-1} \boldsymbol{Z}_i \boldsymbol{Z}_i^\top)^{-1} (\boldsymbol{X}_t \boldsymbol{y}_t + \sum_{i=1}^{t-1} \boldsymbol{Z}_i \boldsymbol{g}_i)$$

$$= (\boldsymbol{X}_t \boldsymbol{X}_t^\top + \sum_{i=1}^{t-1} \boldsymbol{Z}_i \boldsymbol{Z}_i^\top)^{-1} [\boldsymbol{X}_t (\boldsymbol{X}_t^\top \boldsymbol{w}^* + \boldsymbol{\varepsilon}_t) + \sum_{i=1}^{t-1} \boldsymbol{Z}_i (\boldsymbol{Z}_i^\top \boldsymbol{w}^* + \boldsymbol{\varepsilon}_i)]$$

$$= \boldsymbol{w}_t^* + (\boldsymbol{X}_t \boldsymbol{X}_t^\top + \sum_{i=1}^{t-1} \boldsymbol{Z}_i \boldsymbol{Z}_i^\top)^{-1} (\boldsymbol{X}_t \boldsymbol{\varepsilon}_t + \sum_{i=1}^{t-1} \boldsymbol{Z}_i \boldsymbol{\varepsilon}_i)$$

For notational convenience, let $\boldsymbol{U}_t = [\boldsymbol{X}_t \ \boldsymbol{Z}_1 \ \boldsymbol{Z}_2 \ ... \ \boldsymbol{Z}_{t-1}] \in \mathbb{R}^{p \times (n+s)}$ by concatenating the matrices along the second dimension and let $\boldsymbol{Y}_t = [\boldsymbol{y}_t \ \boldsymbol{g}_1 \ \boldsymbol{g}_2 \ ... \ \boldsymbol{g}_{t-1}]^\top \in \mathbb{R}^{n+s}$ denotes the

corresponding concatenated response vector. Then we can calculate the expected $\ell_2$-norm as

$$
\begin{aligned}
\mathbb{E}[L_i(\boldsymbol{w}_t)] =& \mathbb{E}\|\boldsymbol{w}_t - \boldsymbol{w}_i^*\|^2 \\
=& \mathbb{E}\|\boldsymbol{w}_t^* + (\boldsymbol{X}_t\boldsymbol{X}_t^\top + \sum_{i=1}^{t-1}\boldsymbol{Z}_i\boldsymbol{Z}_i^\top)^{-1}(\boldsymbol{X}_t\boldsymbol{\varepsilon}_t + \sum_{i=1}^{t-1}\boldsymbol{Z}_i\boldsymbol{\varepsilon}_i) - \boldsymbol{w}_i^*\|^2 \\
=& \mathbb{E}\|\boldsymbol{w}_t^* - \boldsymbol{w}_i^*\|^2 + \mathbb{E}\|(\boldsymbol{X}_t\boldsymbol{X}_t^\top + \sum_{i=1}^{t-1}\boldsymbol{Z}_i\boldsymbol{Z}_i^\top)^{-1}(\boldsymbol{X}_t\boldsymbol{\varepsilon}_t + \sum_{i=1}^{t-1}\boldsymbol{Z}_i\boldsymbol{\varepsilon}_i)\|^2 \\
=& \|\boldsymbol{w}_t^* - \boldsymbol{w}_i^*\|^2 + \mathbb{E}[\boldsymbol{z}_t^\top\boldsymbol{U}_t^\top(\boldsymbol{U}_t\boldsymbol{U}_t^\top)^{-1})(\boldsymbol{U}_t\boldsymbol{U}_t^\top)^{-1})\boldsymbol{U}_t\boldsymbol{z}_t] \\
=& \|\boldsymbol{w}_t^* - \boldsymbol{w}_i^*\|^2 + \mathbb{E}[tr\{\boldsymbol{U}_t^\top(\boldsymbol{U}_t\boldsymbol{U}_t^\top)^{-1})(\boldsymbol{U}_t\boldsymbol{U}_t^\top)^{-1})\boldsymbol{U}_t\boldsymbol{z}_t\boldsymbol{z}_t^\top\}] \\
=& \|\boldsymbol{w}_t^* - \boldsymbol{w}_i^*\|^2 + \sigma^2\mathbb{E}[tr\{(\boldsymbol{U}_t\boldsymbol{U}_t^\top)^{-1})\}] \\
=& \|\boldsymbol{w}_t^* - \boldsymbol{w}_i^*\|^2 + \frac{p\sigma_t^2}{n+s-p-1}
\end{aligned}
$$

Then we can obtain the adaptation error

$$
\begin{aligned}
\mathbb{E}[A(\boldsymbol{w}_t)] =& \mathbb{E}\|\boldsymbol{w}_t - \boldsymbol{w}_t^*\|^2 \\
=& \mathbb{E}\|\boldsymbol{w}_t^* + (\boldsymbol{X}_t\boldsymbol{X}_t^\top + \sum_{i=1}^{t-1}\boldsymbol{Z}_i\boldsymbol{Z}_i^\top)^{-1}(\boldsymbol{X}_t\boldsymbol{\varepsilon}_t + \sum_{i=1}^{t-1}\boldsymbol{Z}_i\boldsymbol{\varepsilon}_i) - \boldsymbol{w}_t^*\|^2 \\
=& \mathbb{E}\|(\boldsymbol{U}_t\boldsymbol{U}_t^\top)^{-1}\boldsymbol{U}_t\boldsymbol{z}_t\|^2 \\
=& \frac{p\sigma_t^2}{n+s-p-1}
\end{aligned}
\tag{5}
$$

Moreover, we can calculate the memory error

$$
\begin{aligned}
&\mathbb{E}[\mathcal{M}(\boldsymbol{w}_t)] \\
=& \mathbb{E}\left[\frac{1}{t-1}\sum_{k=1}^{t-1}\left(\|\boldsymbol{w}_t - \boldsymbol{w}_k^*\|^2 - \|\boldsymbol{w}_k - \boldsymbol{w}_k^*\|^2\right)\right] \\
=& \frac{1}{t-1}\sum_{k=1}^{t-1}\left[\|\boldsymbol{w}_t^* - \boldsymbol{w}_k^*\|^2 + \frac{p\sigma_t^2}{n_t+s-p-1} - \|\boldsymbol{w}_k^* - \boldsymbol{w}_k^*\|^2 - \frac{p\sigma_k^2}{n_k+s-p-1}\right] \\
=& \frac{1}{t-1}\sum_{k=1}^{t-1}\left[\|\boldsymbol{w}_t^* - \boldsymbol{w}_k^*\|^2 + \frac{p\sigma_t^2}{n_t+s-p-1} - \frac{p\sigma_k^2}{n_k+s-p-1}\right]. \\
=& \frac{1}{t-1}\sum_{k=1}^{t-1}\|\boldsymbol{w}_t^* - \boldsymbol{w}_k^*\|^2 - \frac{1}{t-1}\sum_{k=1}^{t-1}\frac{p\sigma_k^2}{n_k+s-p-1} + \frac{p\sigma_t^2}{n_t+s-p-1}
\end{aligned}
\tag{6}
$$

Finally, the generalization error can be calculated as

$$
\begin{aligned}
\mathbb{E}[\mathcal{G}(\boldsymbol{w}_t)] =& \mathbb{E}\left[\frac{1}{t}\sum_{k=1}^{t}\|\boldsymbol{w}_t - \boldsymbol{w}_k^*\|^2\right] \\
=& \frac{1}{t}\sum_{k=1}^{t}\mathbb{E}\|\boldsymbol{w}_t^* + (\boldsymbol{X}_t\boldsymbol{X}_t^\top + \sum_{i=1}^{t-1}\boldsymbol{Z}_i\boldsymbol{Z}_i^\top)^{-1}(\boldsymbol{X}_t\boldsymbol{\varepsilon}_t + \sum_{i=1}^{t-1}\boldsymbol{Z}_i\boldsymbol{\varepsilon}_i) - \boldsymbol{w}_k^*\|^2 \\
=& \frac{1}{t}\sum_{k=1}^{t}\left[\|\boldsymbol{w}_t^* - \boldsymbol{w}_k^*\|^2 + \frac{p\sigma_t^2}{n_t+s-p-1}\right]
\end{aligned}
\tag{7}
$$

**Lemma A.1.** *Consider continual linear regression models, where the data $\mathcal{D}_t := (\boldsymbol{X}_t, \boldsymbol{y}_t)$ of each task t consist of a feature matrix $\boldsymbol{X}_t \in \mathbb{R}^{p \times n}$ and a response vector $\boldsymbol{y}_t \in \mathbb{R}^n$. When GD/SGD converges to zero mean squared error, the convergence point corresponds to the minimum $\ell_2$-norm solution that lies nearest to the previous task's parameters in parameter space.*

*Proof.* Let $\boldsymbol{w}_t$ be the parameter for the current task, we can calculate the gradient for the $i$-th training sample as $\partial(y_i - \boldsymbol{x}_i^\top \boldsymbol{w})^2/\partial \boldsymbol{w} = -2(y_i - \boldsymbol{x}_i^\top \boldsymbol{w})\boldsymbol{x}_i$. Since the parameter change in each iteration is located in the column space of $\mathbf{X}$, we can always find $a \in \mathbb{R}^n$ such that $\boldsymbol{w} - \boldsymbol{w}_{t-1} = \boldsymbol{X}a$. For the convergence point that makes the training loss become zero, we have $\boldsymbol{X}^\top \boldsymbol{w} = \boldsymbol{y}$. Then we have

$$\boldsymbol{X}^\top \boldsymbol{w} = \boldsymbol{X}^\top(\boldsymbol{w}_{t-1} + \boldsymbol{X}a) = \boldsymbol{X}^\top \boldsymbol{w}_{t-1} + \boldsymbol{X}^\top \boldsymbol{X}a. \tag{15}$$

Then, we can calculate $a$ and subsequently derive a dynamic expression for $\boldsymbol{w}$:

$$\begin{aligned}
\boldsymbol{w} &= \boldsymbol{w}_{t-1} + \boldsymbol{X}a \\
&= \boldsymbol{w}_{t-1} + \boldsymbol{X}(\boldsymbol{X}^\top \boldsymbol{X})^{-1}(\boldsymbol{y} - \boldsymbol{X}^\top \boldsymbol{w}_{t-1}) \\
&= (\boldsymbol{I} - \boldsymbol{X}(\boldsymbol{X}^\top \boldsymbol{X})^{-1}\boldsymbol{X}^\top)\boldsymbol{w}_{t-1} + \boldsymbol{X}(\boldsymbol{X}^\top \boldsymbol{X})^{-1}\boldsymbol{y},
\end{aligned} \tag{16}$$

which is exactly the minimum $\ell_2$-norm solution of the continual linear regression:

$$\arg\min_{\boldsymbol{w}} \|\boldsymbol{w} - \boldsymbol{w}_{t-1}\|^2, \text{ st. } \mathbf{X}^\top \boldsymbol{w} = \boldsymbol{y}. \tag{17}$$

**Lemma A.2.** *Let each element of random matrix $\boldsymbol{U} \in \mathbb{R}^{p \times n}$ be drawn i.i.d. from a standard normal distribution $\mathcal{N}(0,1)$ and $p > n + 1$. There exists a fixed vector $\boldsymbol{w} \in \mathbb{R}^p$ and a normal distribution random vector $\boldsymbol{z} \sim \mathcal{N}(0, \sigma^2 \boldsymbol{I}_n)$ satisfying the following equation:*

$$\mathbb{E}\left[(\boldsymbol{U}^\top \boldsymbol{U})^{-1}\right] = \frac{\boldsymbol{I}_{n \times n}}{p - n - 1}, \tag{18}$$

$$\mathbb{E}\left\|\boldsymbol{U}\left(\boldsymbol{U}^\top \boldsymbol{U}\right)^{-1}\boldsymbol{U}^\top \boldsymbol{w}\right\|^2 = \frac{n}{p}\|\boldsymbol{w}\|^2, \tag{19}$$

$$\mathbb{E}\left\|\boldsymbol{U}(\boldsymbol{U}^\top \boldsymbol{U})^{-1}\boldsymbol{\epsilon}\right\|^2 = \frac{n\sigma^2}{p - n - 1}. \tag{20}$$

*Proof.* According to the definition of matrix $\boldsymbol{U}$, we have $(\boldsymbol{U}^\top \boldsymbol{U})^{-1}$ follows the inverse-Wishart distribution with identity scale matrix $\boldsymbol{I} \in \mathbb{R}^{n \times n}$ and $p$ degrees-of-freedom, Therefore, for $p > n+1$, we have $\mathbb{E}\left[(\boldsymbol{U}^\top \boldsymbol{U})^{-1}\right] = \frac{\boldsymbol{I}_{n \times n}}{p-n-1}$ and the first equation holds. Let $\boldsymbol{\Pi} := \boldsymbol{U}\left(\boldsymbol{U}^\top \boldsymbol{U}\right)^{-1}\boldsymbol{U}^\top$, it can be shown that $\boldsymbol{\Pi}$ is a orthogonal projection matrix that projects a $p$-dim vector to the column space of $\boldsymbol{U}$ and satisfies $\boldsymbol{\Pi}^2 = \boldsymbol{\Pi}$. Since each element of $\boldsymbol{U}$ is $i.i.d.$ following standard Gaussian, we have $\mathbb{E}[\|\boldsymbol{\Pi}\boldsymbol{w}\|^2] = \frac{n}{p}\|\boldsymbol{w}\|^2$ by rotational symmetry of the standard normal distribution. We can get

$$\begin{aligned}
\|\boldsymbol{U}(\boldsymbol{U}^\top \boldsymbol{U})^{-1}\boldsymbol{\epsilon}\|^2 &= \boldsymbol{\epsilon}^\top(\boldsymbol{U}^\top \boldsymbol{U})^{-1}(\boldsymbol{U}^\top \boldsymbol{U})(\boldsymbol{U}^\top \boldsymbol{U})^{-1}\boldsymbol{\epsilon} \\
&= tr((\boldsymbol{U}^\top \boldsymbol{U})^{-1}(\boldsymbol{U}^\top \boldsymbol{U})(\boldsymbol{U}^\top \boldsymbol{U})^{-1}\boldsymbol{\epsilon}\boldsymbol{\epsilon}^\top) \\
&= tr((\boldsymbol{U}^\top \boldsymbol{U})^{-1}\boldsymbol{\epsilon}\boldsymbol{\epsilon}^\top).
\end{aligned} \tag{21}$$

By computing the expectation of the above equation, we derive

$$\begin{aligned}
\mathbb{E}[\|\boldsymbol{U}(\boldsymbol{U}^\top \boldsymbol{U})^{-1}\boldsymbol{\epsilon}\|^2] &= \mathbb{E}\left[\mathrm{tr}\left((\boldsymbol{U}^\top \boldsymbol{U})^{-1}\boldsymbol{\epsilon}\boldsymbol{\epsilon}^\top\right)\right] \\
&= \mathrm{tr}\left(\mathbb{E}\left[(\boldsymbol{U}^\top \boldsymbol{U})^{-1}\boldsymbol{\epsilon}\boldsymbol{\epsilon}^\top\right]\right) \\
&= \mathrm{tr}\left(\mathbb{E}\left[(\boldsymbol{U}^\top \boldsymbol{U})^{-1}\right]\mathbb{E}\left[\boldsymbol{\epsilon}\boldsymbol{\epsilon}^\top\right]\right) \\
&= \sigma^2 \, \mathrm{tr}\left(\mathbb{E}\left[(\boldsymbol{U}^\top \boldsymbol{U})^{-1}\right]\right) \\
&= \frac{n\sigma^2}{p - n - 1}.
\end{aligned} \tag{22}$$

**Lemma A.3.** *Consider the oracle estimator described in Appendix E. We construct $\boldsymbol{X}_{1T} \in \mathbb{R}^{p \times nT}$ by concatenating the data matrices from $\boldsymbol{X}_1$ to $\boldsymbol{X}_T$ along the second dimension, i.e. $\boldsymbol{X}_{1T} = [\boldsymbol{X}_1 \ \boldsymbol{X}_2 \ ... \ \boldsymbol{X}_T]$, and $\boldsymbol{y}_{1T}$ denotes the corresponding concatenated response vector. The solution to*

*the corresponding optimization problem for the oracle estimator takes different forms depending on the parameterization regime. In the overparameterized regime, the solution is given by*

$$\widehat{\boldsymbol{w}}_T^{(Ora)} = \left( \boldsymbol{I} - \boldsymbol{X}_{1T} \left( \boldsymbol{X}_{1T}^\top \boldsymbol{X}_{1T} \right)^{-1} \boldsymbol{X}_{1T}^\top \right) \boldsymbol{w}_0 + \boldsymbol{X}_{1T} \left( \boldsymbol{X}_{1T}^\top \boldsymbol{X}_{1T} \right)^{-1} \boldsymbol{y}_{1T}. \tag{23}$$

*In the underparameterized regime, the solution is given by the following expression*

$$\widehat{\boldsymbol{w}}_T^{(Ora)} = \left( \boldsymbol{X}_{1T} \boldsymbol{X}_{1T}^\top \right)^{-1} \boldsymbol{X}_{1T} \boldsymbol{y}_{1T}. \tag{24}$$

*Proof.* When underparameterized, the optimization problem for the oracle estimator can be formulated as: $\arg\min_{\boldsymbol{w}} \left\| \boldsymbol{X}_{1T}^T \boldsymbol{w} - \boldsymbol{y}_{1T} \right\|^2$. Let $L(\boldsymbol{w}) = \left\| \boldsymbol{X}_{1T}^\top \boldsymbol{w} - \boldsymbol{y}_{1T} \right\|^2$ denotes a function of $\boldsymbol{w}$ and we have

$$L(\boldsymbol{w}) = \| \boldsymbol{X}_{1T}^\top \boldsymbol{w} - \boldsymbol{y}_{1T} \|^2 = (\boldsymbol{X}_{1T}^\top \boldsymbol{w} - \boldsymbol{y}_{1T})^\top (\boldsymbol{X}_{1T}^\top \boldsymbol{w} - \boldsymbol{y}_{1T}). \tag{25}$$

By computing the derivative of $L$ with respect to $\boldsymbol{w}$ and setting it to zero, we must have

$$\boldsymbol{X}_{1T} \boldsymbol{X}_{1T}^\top \widehat{\boldsymbol{w}} = \boldsymbol{X}_{1T} \boldsymbol{y}_{1T} \Rightarrow \widehat{\boldsymbol{w}} = \left( \boldsymbol{X}_{1T} \boldsymbol{X}_{1T}^\top \right)^{-1} \boldsymbol{X}_{1T} \boldsymbol{y}_{1T}. \tag{26}$$

When overparameterized, the optimization problem can be formalized as: $\arg\min_{\boldsymbol{w}} \| \boldsymbol{w} - \boldsymbol{w}_0 \|^2$, st. $\boldsymbol{X}_{1T}^\top \boldsymbol{w} = \boldsymbol{y}_{1T}$. Using the Lagrange multipliers, we can get the objective function

$$\arg\min_{\boldsymbol{w}, \boldsymbol{\lambda}} \frac{1}{2} \| \boldsymbol{w} - \boldsymbol{w}_0 \|_2^2 + \boldsymbol{\lambda}^\top (\boldsymbol{X}_{1T}^\top \boldsymbol{w} - \boldsymbol{y_{1T}}). \tag{27}$$

By setting the derivative with respect to $\boldsymbol{w}$ and $\lambda$ to 0, it directly follows that

$$\widehat{\boldsymbol{w}} - \boldsymbol{w}_0 + \boldsymbol{X}_{1T} \boldsymbol{\lambda} = 0 \Rightarrow \widehat{\boldsymbol{w}} = -\boldsymbol{X}_{1T} \boldsymbol{\lambda} + \boldsymbol{w}_0, \tag{28}$$

$$\boldsymbol{\lambda} = (\boldsymbol{X}_{1T}^\top \boldsymbol{X}_{1T})^{-1} \boldsymbol{X}_{1T}^\top \boldsymbol{w}_0 - (\boldsymbol{X}_{1T}^\top \boldsymbol{X}_{1T})^{-1} \boldsymbol{y}_{1T}. \tag{29}$$

By substituting Equation(27) into Equation(26), we obtain the following result

$$\widehat{\boldsymbol{w}} = \left( \boldsymbol{I} - \boldsymbol{X}_{1T} \left( \boldsymbol{X}_{1T}^\top \boldsymbol{X}_{1T} \right)^{-1} \boldsymbol{X}_{1T}^\top \right) \boldsymbol{w}_0 + \boldsymbol{X}_{1T} \left( \boldsymbol{X}_{1T}^\top \boldsymbol{X}_{1T} \right)^{-1} \boldsymbol{y}_{1T}. \tag{30}$$

## B  PROOF OF THEOREM 1

Define $\boldsymbol{U}_t = [\boldsymbol{X}_t \ \boldsymbol{Z}_1 \ \boldsymbol{Z}_2 \ ... \ \boldsymbol{Z}_{t-1}] \in \mathbb{R}^{p \times (n+s)}$, and define $\boldsymbol{Y}_t = [\boldsymbol{y}_t \ \boldsymbol{g}_1 \ \boldsymbol{g}_2 \ ... \ \boldsymbol{g}_{t-1}]^\top \in \mathbb{R}^{n+s}$. In the overparameterized regime (where $p > n + s$), the optimization problem can be represented by the following expression, describing the solution structure and constraints involved, as

$$\arg\min_{\boldsymbol{w}} \| \boldsymbol{w} - \boldsymbol{w}_{t-1} \|^2, \text{s.t.} (\boldsymbol{X}_t)^\top \boldsymbol{w} = \boldsymbol{y}_t, (\boldsymbol{Z}_i)^\top \boldsymbol{w} = \boldsymbol{g}_i (i = 1, 2, ..., t-1), \tag{31}$$

and we can reformulate it as: $\arg\min \| \boldsymbol{w} - \boldsymbol{w}_{t-1} \|^2, s.t. \boldsymbol{U}_t^\top \boldsymbol{w} = \boldsymbol{Y}_t$. We can define

$$L(\boldsymbol{w}, \lambda) = \frac{1}{2} \| \boldsymbol{w} - \boldsymbol{w}_{t-1} \|^2 + \lambda^T (\boldsymbol{U}_t^\top \boldsymbol{w} - \boldsymbol{Y}_t). \tag{32}$$

Using the Lagrange multipliers and set the derivative w.r.t. $\boldsymbol{w}$ and $\lambda$ to 0, we can get

$$\boldsymbol{w} - \boldsymbol{w}_{t-1} + \boldsymbol{U}_t \boldsymbol{\lambda} = 0 \Rightarrow \boldsymbol{w} = -\boldsymbol{U}_t \boldsymbol{\lambda} + \boldsymbol{w}_{t-1}, \tag{33}$$

$$\boldsymbol{\lambda} = (\boldsymbol{U}_t^\top \boldsymbol{U}_t)^{-1} \boldsymbol{U}_t^\top \boldsymbol{w}_{t-1} - (\boldsymbol{U}_t^\top \boldsymbol{U}_t)^{-1} \boldsymbol{Y}_t, \tag{34}$$

$$\boldsymbol{w} = (\boldsymbol{I} - \boldsymbol{U}_t (\boldsymbol{U}_t^\top \boldsymbol{U}_t)^{-1} \boldsymbol{U}_t^\top) \boldsymbol{w}_{t-1} + \boldsymbol{U}_t (\boldsymbol{U}_t^\top \boldsymbol{U}_t)^{-1} \boldsymbol{Y}_t. \tag{35}$$

Therefore, for the $t$-th task, the parameters to be optimized can be represented as

$$\boldsymbol{w_t} = (\boldsymbol{I} - \boldsymbol{U}_t (\boldsymbol{U}_t^\top \boldsymbol{U}_t)^{-1} \boldsymbol{U}_t^\top) \boldsymbol{w}_{t-1} + \boldsymbol{U}_t (\boldsymbol{U}_t^\top \boldsymbol{U}_t)^{-1} \boldsymbol{Y}_t \tag{36}$$

Then, we compute the expected $\ell_2$-norm between the optimized parameters of the $t+1$-th task and the optimal parameters of the $i$-th task, providing a measure of their difference in parameter space.

$$
\begin{aligned}
&\mathbb{E}[\|\boldsymbol{w}_{t+1} - \boldsymbol{w}_i^*\|^2]\\
=&\mathbb{E}[\|(\boldsymbol{I} - \boldsymbol{U}_{t+1}(\boldsymbol{U}_{t+1}^\top \boldsymbol{U}_{t+1})^{-1}\boldsymbol{U}_{t+1}^\top)\boldsymbol{w}_t + \boldsymbol{U}_{t+1}(\boldsymbol{U}_{t+1}^\top \boldsymbol{U}_{t+1})^{-1}\boldsymbol{Y}_{t+1} - \boldsymbol{w}_i^*\|^2]\\
=&\mathbb{E}[\|(\boldsymbol{I} - \boldsymbol{U}_{t+1}(\boldsymbol{U}_{t+1}^\top \boldsymbol{U}_{t+1})^{-1}\boldsymbol{U}_{t+1}^\top)\boldsymbol{w}_t + \boldsymbol{U}_{t+1}(\boldsymbol{U}_{t+1}^\top \boldsymbol{U}_{t+1})^{-1}(\boldsymbol{U}_{t+1}^\top \boldsymbol{w}_{t+1}^* + \boldsymbol{z}_{t+1}) - \boldsymbol{w}_i^*\|^2]\\
=&\mathbb{E}[\|(\boldsymbol{I} - \boldsymbol{U}_{t+1}(\boldsymbol{U}_{t+1}^\top \boldsymbol{U}_{t+1})^{-1}\boldsymbol{U}_{t+1}^\top)(\boldsymbol{w}_t - \boldsymbol{w}_i^*) + \boldsymbol{U}_{t+1}(\boldsymbol{U}_{t+1}^\top \boldsymbol{U}_{t+1})^{-1}\boldsymbol{U}_{t+1}^\top(\boldsymbol{w}_{t+1}^* - \boldsymbol{w}_i^*)\\
&+ \boldsymbol{U}_{t+1}(\boldsymbol{U}_{t+1}^\top \boldsymbol{U}_{t+1})^{-1}\boldsymbol{z}_{t+1}\|^2]\\
=&\mathbb{E}[\|(\boldsymbol{I} - \boldsymbol{U}_{t+1}(\boldsymbol{U}_{t+1}^\top \boldsymbol{U}_{t+1})^{-1}\boldsymbol{U}_{t+1}^\top)(\boldsymbol{w}_t - \boldsymbol{w}_i^*) + \boldsymbol{U}_{t+1}(\boldsymbol{U}_{t+1}^\top \boldsymbol{U}_{t+1})^{-1}\boldsymbol{U}_{t+1}^\top(\boldsymbol{w}_{t+1}^* - \boldsymbol{w}_i^*)\|^2]\\
&+ 2\mathbb{E}[\langle(\boldsymbol{I} - \boldsymbol{U}_{t+1}(\boldsymbol{U}_{t+1}^\top \boldsymbol{U}_{t+1})^{-1}\boldsymbol{U}_{t+1}^\top)(\boldsymbol{w}_t - \boldsymbol{w}_i^*) + \boldsymbol{U}_{t+1}(\boldsymbol{U}_{t+1}^\top \boldsymbol{U}_{t+1})^{-1}\boldsymbol{U}_{t+1}^\top(\boldsymbol{w}_{t+1}^* - \boldsymbol{w}_i^*),\\
&\boldsymbol{U}_{t+1}(\boldsymbol{U}_{t+1}^\top \boldsymbol{U}_{t+1})^{-1}\boldsymbol{z}_{t+1}\rangle] + \mathbb{E}[\|\boldsymbol{U}_{t+1}(\boldsymbol{U}_{t+1}^\top \boldsymbol{U}_{t+1})^{-1}\boldsymbol{z}_{t+1}\|^2]
\end{aligned}
$$
(37)

For notational convenience, we define $\boldsymbol{\Pi}_t := \boldsymbol{U}_t(\boldsymbol{U}_t^\top \boldsymbol{U}_t)^{-1}\boldsymbol{U}_t^\top$. It can be shown that $\boldsymbol{\Pi}$ is a orthogonal projection matrix that projects a $p$-dim vector to the column space of $\boldsymbol{U}$, and it satisfies $\boldsymbol{\Pi}_t^2 = \boldsymbol{\Pi}_t$. Therefore, the Equation (37) can be equivalently expressed as follows

$$
\begin{aligned}
&\mathbb{E}[\|\boldsymbol{w}_{t+1} - \boldsymbol{w}_i^*\|^2]\\
=&\mathbb{E}[\|(\boldsymbol{I} - \boldsymbol{\Pi}_{t+1})(\boldsymbol{w}_t - \boldsymbol{w}_i^*) + \boldsymbol{\Pi}_{t+1}(\boldsymbol{w}_{t+1}^* - \boldsymbol{w}_i^*)\|^2] + \mathbb{E}[\|\boldsymbol{U}_{t+1}(\boldsymbol{U}_{t+1}^\top \boldsymbol{U}_{t+1})^{-1}\boldsymbol{z}_{t+1}\|^2]\\
&+ 2\mathbb{E}[\langle(\boldsymbol{I} - \boldsymbol{\Pi}_{t+1})(\boldsymbol{w}_t - \boldsymbol{w}_i^*) + \boldsymbol{\Pi}_{t+1}(\boldsymbol{w}_{t+1}^* - \boldsymbol{w}_i^*),\ \boldsymbol{U}_{t+1}(\boldsymbol{U}_{t+1}^\top \boldsymbol{U}_{t+1})^{-1}\boldsymbol{z}_{t+1}\rangle]\\
=&(1 - \frac{n+s}{p})\mathbb{E}[\|\boldsymbol{w}_t - \boldsymbol{w}_i^*\|^2] + \frac{n+s}{p}\|\boldsymbol{w}_{t+1}^* - \boldsymbol{w}_i^*\|^2 + \frac{(n+s)\sigma^2}{p-n-s-1},
\end{aligned}
$$
(38)

where $\mathbb{E}[\|\boldsymbol{\Pi}_{t+1}(\boldsymbol{w}_{t+1}^* - \boldsymbol{w}_i^*)\|^2] = \frac{n+s}{p}\|\boldsymbol{w}_{t+1}^* - \boldsymbol{w}_i^*\|^2$ as established in Lemma A.2, and

$$
\begin{aligned}
&\mathbb{E}[\langle(\boldsymbol{I} - \boldsymbol{\Pi}_{t+1})(\boldsymbol{w}_t - \boldsymbol{w}_i^*) + \boldsymbol{\Pi}_{t+1}(\boldsymbol{w}_{t+1}^* - \boldsymbol{w}_i^*),\ \boldsymbol{U}_{t+1}(\boldsymbol{U}_{t+1}^\top \boldsymbol{U}_{t+1})^{-1}\boldsymbol{z}_{t+1}\rangle]\\
=&\mathbb{E}[\langle(\boldsymbol{I} - \boldsymbol{\Pi}_{t+1})(\boldsymbol{w}_t - \boldsymbol{w}_i^*),\ \boldsymbol{U}_{t+1}(\boldsymbol{U}_{t+1}^\top \boldsymbol{U}_{t+1})^{-1}\boldsymbol{z}_{t+1}\rangle]\\
&+ \mathbb{E}[\langle\boldsymbol{\Pi}_{t+1}(\boldsymbol{w}_{t+1}^* - \boldsymbol{w}_i^*),\ \boldsymbol{U}_{t+1}(\boldsymbol{U}_{t+1}^\top \boldsymbol{U}_{t+1})^{-1}\boldsymbol{z}_{t+1}\rangle]\\
=&\mathbb{E}[\langle(\boldsymbol{U}_{t+1}(\boldsymbol{U}_{t+1}^\top \boldsymbol{U}_{t+1})^{-1})^\top \boldsymbol{\Pi}_{t+1}(\boldsymbol{w}_{t+1}^* - \boldsymbol{w}_i^*), \boldsymbol{z}_{t+1}\rangle] = 0
\end{aligned}
$$
(39)

Regarding the calculation of the noise term, we have the following expression

$$
\begin{aligned}
&\mathbb{E}[\|\boldsymbol{U}_{t+1}(\boldsymbol{U}_{t+1}^\top \boldsymbol{U}_{t+1})^{-1}\boldsymbol{z}_{t+1}\|^2]\\
=&\mathbb{E}[tr((\boldsymbol{U}_{t+1}^\top \boldsymbol{U}_{t+1})^{-1}(\boldsymbol{U}_{t+1}^\top \boldsymbol{U}_{t+1})(\boldsymbol{U}_{t+1}^\top \boldsymbol{U}_{t+1})^{-1}\boldsymbol{z}_{t+1}\boldsymbol{z}_{t+1}^\top)]\\
=&\mathbb{E}[tr((\boldsymbol{U}_{t+1}^\top \boldsymbol{U}_{t+1})^{-1}\boldsymbol{z}_{t+1}\boldsymbol{z}_{t+1}^\top)] = tr[\mathbb{E}[(\boldsymbol{U}_{t+1}^\top \boldsymbol{U}_{t+1})^{-1}\boldsymbol{z}_{t+1}\boldsymbol{z}_{t+1}^\top]]\\
=&\sigma^2 tr(\mathbb{E}[(\boldsymbol{U}_{t+1}^\top \boldsymbol{U}_{t+1})^{-1}])\\
=&\frac{(n+s)\sigma^2}{p-n-s-1}
\end{aligned}
$$
(40)

Therefore, the expression for $\mathbb{E}[\|\boldsymbol{w}_t - \boldsymbol{w}_i^*\|^2]$ can be computed iteratively.(Let $\boldsymbol{w}_0 = \boldsymbol{0}$)

$$
\begin{aligned}
&\mathbb{E}[\|\boldsymbol{w}_t - \boldsymbol{w}_i^*\|^2]\\
=&(\frac{p-n-s}{p})\mathbb{E}[\|\boldsymbol{w}_{t-1} - \boldsymbol{w}_i^*\|^2] + \frac{n+s}{p}\|\boldsymbol{w}_t^* - \boldsymbol{w}_i^*\|^2 + \frac{(n+s)\sigma^2}{p-n-s-1}\\
=&\left(\frac{p-n-s}{p}\right)^t \|\boldsymbol{w}_i^*\|^2 + \sum_{k=1}^t \left(\frac{p-n-s}{p}\right)^{t-k} \frac{n+s}{p}\|\boldsymbol{w}_k^* - \boldsymbol{w}_i^*\|^2\\
&+ \frac{(n+s)\sigma^2}{p-n-s-1}\sum_{k=1}^t \left(\frac{p-n-s}{p}\right)^{t-k}
\end{aligned}
$$
(41)

Therefore, based on the previous derivations, we can calculate the adaptation error

$$\mathbb{E}[\mathcal{A}(\boldsymbol{w}_t)] = \mathbb{E}\left\|\boldsymbol{w}_t - \boldsymbol{w}_t^*\right\|^2$$

$$= \left(\frac{p-n-s}{p}\right)^t \|\boldsymbol{w}_t^*\|^2 + \sum_{k=1}^{t} \left(\frac{p-n-s}{p}\right)^{t-k} \frac{n+s}{p} \|\boldsymbol{w}_k^* - \boldsymbol{w}_t^*\|^2$$

$$+ \frac{(n+s)\sigma^2}{p-n-s-1} \sum_{k=1}^{t} \left(\frac{p-n-s}{p}\right)^{t-k} \tag{42}$$

$$= \left(\frac{p-n-s}{p}\right)^t \|\boldsymbol{w}_t^*\|^2 + \sum_{k=1}^{t} \left(\frac{p-n-s}{p}\right)^{t-k} \frac{n+s}{p} \|\boldsymbol{w}_k^* - \boldsymbol{w}_t^*\|^2$$

$$+ \frac{(1 - (\frac{p-n-s}{p})^t)p\sigma^2}{(p-n-s-1)}$$

In the underparameterized regime ($p < n + s$), the optimization problem can be reformulated as $\arg\min_{\boldsymbol{w}} \left\|\boldsymbol{U}_t^\top \boldsymbol{w} - \boldsymbol{Y}_t\right\|^2$. In this situation, the optimization problem admits a unique solution, the model parameters are independent of those from the previous tasks. Define objective function

$$L(\boldsymbol{w}) = \|\boldsymbol{U}_t^\top \boldsymbol{w} - \boldsymbol{Y}_t\|^2$$

$$= (\boldsymbol{U}_t^\top \boldsymbol{w} - \boldsymbol{Y}_t)^\top (\boldsymbol{U}_t^\top \boldsymbol{w} - \boldsymbol{Y}_t) \tag{43}$$

$$= \boldsymbol{w}^\top \boldsymbol{U}_t \boldsymbol{U}_t^\top \boldsymbol{w} - \boldsymbol{Y}_t^\top \boldsymbol{U}_t^\top \boldsymbol{w} - \boldsymbol{w}^\top \boldsymbol{U}_t \boldsymbol{Y}_t + \boldsymbol{Y}_t^\top \boldsymbol{Y}_t$$

By setting the derivative w.r.t. $\boldsymbol{w}$ to 0, it follows that

$$\boldsymbol{w} = (\boldsymbol{U}_t \boldsymbol{U}_t^\top)^{-1} \boldsymbol{U}_t \boldsymbol{Y}_t = (\boldsymbol{U}_t \boldsymbol{U}_t^\top)^{-1} \boldsymbol{U}_t (\boldsymbol{U}_t^\top \boldsymbol{w}^* + \boldsymbol{z}_t)$$

$$= \boldsymbol{w}^* + (\boldsymbol{U}_t \boldsymbol{U}_t^\top)^{-1} \boldsymbol{U}_t \boldsymbol{z}_t. \tag{44}$$

Then we can calculate the expected $\ell_2$-norm, which is given by

$$\mathbb{E}[L_i(\boldsymbol{w}_t)]$$

$$= \mathbb{E}\|\boldsymbol{w}_t - \boldsymbol{w}_i^*\|^2$$

$$= \mathbb{E}\|\boldsymbol{w}_t^* + (\boldsymbol{U}_t \boldsymbol{U}_t^\top)^{-1} \boldsymbol{U}_t \boldsymbol{z}_t - \boldsymbol{w}_i^*\|^2$$

$$= \mathbb{E}\|\boldsymbol{w}_t^* - \boldsymbol{w}_i^*\|^2 + \frac{p\sigma^2}{n+s-p-1} \tag{45}$$

$$= \|\boldsymbol{w}_t^* - \boldsymbol{w}_i^*\|^2 + \frac{p\sigma^2}{n+s-p-1}$$

Therefore, the adaptation error can be expressed as

$$\mathbb{E}[\mathcal{A}(\boldsymbol{w}_t)]$$

$$= \mathbb{E}\left\|\boldsymbol{w}_t^* + (\boldsymbol{U}_t \boldsymbol{U}_t^\top)^{-1} \boldsymbol{U}_t \boldsymbol{z}_t - \boldsymbol{w}_t^*\right\|^2$$

$$= \mathbb{E}\left\|(\boldsymbol{U}_t \boldsymbol{U}_t^\top)^{-1} \boldsymbol{U}_t \boldsymbol{z}_t\right\|^2 \tag{46}$$

$$= \frac{p\sigma^2}{n+s-p-1}$$

## C    PROOF OF THEOREM 2

In the overrparameterized regime($p > n + s$), according to Equation (41), we have

$$\mathbb{E}[\|\boldsymbol{w}_t - \boldsymbol{w}_i^*\|^2]$$

$$= \left(\frac{p-n-s}{p}\right)^t \|\boldsymbol{w}_i^*\|^2 + \sum_{k=1}^{t} \left(\frac{p-n-s}{p}\right)^{t-k} \frac{n+s}{p} \|\boldsymbol{w}_k^* - \boldsymbol{w}_i^*\|^2$$

$$+ \frac{(n+s)\sigma^2}{p-n-s-1} \sum_{k=1}^{t} \left(\frac{p-n-s}{p}\right)^{t-k}, \tag{47}$$

and by setting $t = i$, we have

$$
\begin{aligned}
&\mathbb{E}[\|\boldsymbol{w}_i - \boldsymbol{w}_i^*\|^2] \\
&= \left(\frac{p-n-s}{p}\right)^i \|\boldsymbol{w}_i^*\|^2 + \sum_{k=1}^{i} \left(\frac{p-n-s}{p}\right)^{i-k} \frac{n+s}{p} \|\boldsymbol{w}_k^* - \boldsymbol{w}_i^*\|^2 \\
&\quad + \frac{(n+s)\sigma^2}{p-n-s-1} \sum_{k=1}^{i} \left(\frac{p-n-s}{p}\right)^{i-k}
\end{aligned}
\tag{48}
$$

Thus, through calculation, we derive the expression for the memory error as

$$
\begin{aligned}
\mathbb{E}[\mathcal{M}(\mathbf{w}_t)] &= \frac{1}{t-1} \sum_{i=1}^{t-1} \mathbb{E}\left[\|\boldsymbol{w}_t - \boldsymbol{w}_i^*\|^2 - \|\boldsymbol{w}_i - \boldsymbol{w}_i^*\|^2\right] \\
&= \frac{1}{t-1} \sum_{i=1}^{t-1} \Bigg[ \left(\frac{p-n-s}{p}\right)^t \|\boldsymbol{w}_i^*\|^2 + \sum_{k=1}^{t} \left(\frac{p-n-s}{p}\right)^{t-k} \frac{n+s}{p} \|\boldsymbol{w}_k^* - \boldsymbol{w}_i^*\|^2 \\
&\quad + \frac{(n+s)\sigma^2}{p-n-s-1} \sum_{k=1}^{t} \left(\frac{p-n-s}{p}\right)^{t-k} \\
&\quad - \left(\frac{p-n-s}{p}\right)^i \|\boldsymbol{w}_i^*\|^2 - \sum_{k=1}^{i} \left(\frac{p-n-s}{p}\right)^{i-k} \frac{n+s}{p} \|\boldsymbol{w}_k^* - \boldsymbol{w}_i^*\|^2 \\
&\quad - \frac{(n+s)\sigma^2}{p-n-s-1} \sum_{k=1}^{i} \left(\frac{p-n-s}{p}\right)^{i-k} \Bigg] \\
&= \frac{1}{t-1} \sum_{i=1}^{t-1} \Bigg\{ \left[\left(\frac{p-n-s}{p}\right)^t - \left(\frac{p-n-s}{p}\right)^i\right] \|w_i^*\|^2 \\
&\quad + \sum_{k=i+1}^{t} \frac{n+s}{p} \left(\frac{p-n-s}{p}\right)^{t-k} \|\boldsymbol{w}_k^* - \boldsymbol{w}_i^*\|^2 + \frac{(n+s)\sigma^2}{p-n-s-1} \sum_{k=1}^{i} \left[\left(\frac{p-n-s}{p}\right)^{t-k} - \left(\frac{p-n-s}{p}\right)^{i-k}\right] \\
&\quad + \frac{(n+s)\sigma^2}{p-n-s-1} \sum_{k=i+1}^{t} \left(1 - \frac{n+s}{p}\right)^{t-k} + \sum_{k=1}^{i} \frac{n+s}{p} \left[\left(\frac{p-n-s}{p}\right)^{t-k} - \left(\frac{p-n-s}{p}\right)^{i-k}\right] \|\boldsymbol{w}_k^* - \boldsymbol{w}_i^*\|^2 \Bigg\} \\
&= \frac{1}{t-1} \sum_{i=1}^{t-1} \Bigg\{ \left[\left(\frac{p-n-s}{p}\right)^t - \left(\frac{p-n-s}{p}\right)^i\right] \|\boldsymbol{w}_i^*\|^2 + \frac{(n+s)\sigma^2}{p-n-s-1} \sum_{k=1}^{i} \left[\left(\frac{p-n-s}{p}\right)^{t-k} - \left(\frac{p-n-s}{p}\right)^{i-k}\right] \\
&\quad + \sum_{j>i}^{t} u_{kj} \|\boldsymbol{w}_i^* - \boldsymbol{w}_j^*\|^2 + \frac{(n+s)\sigma^2}{p-n-s-1} \sum_{k=i+1}^{t} \left(\frac{p-n-s}{p}\right)^{t-k} \Bigg\} \\
&= \frac{1}{t-1} \sum_{i=1}^{t-1} \Bigg\{ \left[\left(\frac{p-n-s}{p}\right)^t - \left(\frac{p-n-s}{p}\right)^i\right] \|w_i^*\|^2 + \sum_{j>i}^{t} u_{kj} \|\boldsymbol{w}_i^* - \boldsymbol{w}_j^*\|^2 \\
&\quad + \frac{(n+s)\sigma^2}{p-n-s-1} \left[\sum_{k=1}^{t} \left(\frac{p-n-s}{p}\right)^{t-k} - \sum_{k=1}^{i} \left(\frac{p-n-s}{p}\right)^{i-k}\right] \Bigg\} \\
&= \frac{1}{t-1} \sum_{k=1}^{t-1} \left(\lambda^t - \lambda^i\right) \|\boldsymbol{w}_i^*\|^2 + \frac{1}{t-1} \sum_{k=1}^{t-1}\sum_{j>k}^{t} \frac{n+s}{p} u_{kj} \|\boldsymbol{w}_j^* - \boldsymbol{w}_k^*\|^2 \\
&\quad + \frac{1}{t-1} \sum_{k=1}^{t-1} \frac{p\sigma^2}{p-n-s-1} \left(\lambda^i - \lambda^t\right)
\end{aligned}
\tag{49}
$$

In the underparameterized regime ($p < n + s$), according to Equation (45), we have

$$\mathbb{E}[L_i(\boldsymbol{w}_t)] = \|\boldsymbol{w}_t^* - \boldsymbol{w}_i^*\|^2 + \frac{p\sigma^2}{n+s-p-1} \tag{50}$$

Thus, we can derive the expression for the memory error as

$$\mathbb{E}[\mathcal{M}(\boldsymbol{w}_t)] = \mathbb{E}\left[\frac{1}{t-1}\sum_{k=1}^{t-1}\left(\|\boldsymbol{w}_t - \boldsymbol{w}_k^*\|^2 - \|\boldsymbol{w}_k - \boldsymbol{w}_k^*\|^2\right)\right]$$

$$= \frac{1}{t-1}\sum_{k=1}^{t-1}\left(\mathbb{E}\|\boldsymbol{w}_t - \boldsymbol{w}_k^*\|^2 - \mathbb{E}\|\boldsymbol{w}_k - \boldsymbol{w}_k^*\|^2\right)$$

$$= \frac{1}{t-1}\sum_{k=1}^{t-1}\left[\|\boldsymbol{w}_t^* - \boldsymbol{w}_k^*\|^2 + \frac{p\sigma^2}{n+s-p-1}\right. \tag{51}$$

$$\left. -\|\boldsymbol{w}_k^* - \boldsymbol{w}_k^*\|^2 - \frac{p\sigma^2}{n+s-p-1}\right]$$

$$= \frac{1}{t-1}\sum_{k=1}^{t-1}\|\boldsymbol{w}_t^* - \boldsymbol{w}_k^*\|^2.$$

## D  PROOF OF THEOREM 3

In the overparameterized regime ($p > n + s$), we have

$$\mathbb{E}[\|\boldsymbol{w}_t - \boldsymbol{w}_k^*\|^2]$$

$$= \left(\frac{p-n-s}{p}\right)^t\|\boldsymbol{w}_k^*\|^2 + \sum_{j=1}^{t}\left(\frac{p-n-s}{p}\right)^{t-j}\frac{n+s}{p}\|\boldsymbol{w}_k^* - \boldsymbol{w}_j^*\|^2 \tag{52}$$

$$+ \frac{(n+s)\sigma^2}{p-n-s-1}\sum_{i=1}^{t}\left(1 - \frac{n+s}{p}\right)^{t-i}$$

Thus, we can derive the expression for the forsight error as

$$\mathbb{E}[\mathcal{G}(\boldsymbol{w}_t)] = \mathbb{E}[\frac{1}{t}\sum_{k=1}^{t}\|\boldsymbol{w}_t - \boldsymbol{w}_k^*\|^2]$$

$$= \frac{1}{t}\sum_{k=1}^{t}\mathbb{E}\|\boldsymbol{w}_t - \boldsymbol{w}_k^*\|^2$$

$$= \frac{1}{t}\sum_{k=1}^{t}\left[\left(\frac{p-n-s}{p}\right)^t\|\boldsymbol{w}_k^*\|^2 + \sum_{j=1}^{t}\left(\frac{p-n-s}{p}\right)^{t-k}\frac{n+s}{p}\|\boldsymbol{w}_k^* - \boldsymbol{w}_j^*\|^2\right.$$

$$\left. + \frac{(n+s)\sigma^2}{p-n-s-1}\sum_{i=1}^{t}\left(1 - \frac{n+s}{p}\right)^{t-i}\right] \tag{53}$$

$$= \frac{1}{t}\sum_{k=1}^{t}\left(\frac{p-n-s}{p}\right)^t\|\boldsymbol{w}_k^*\|^2 + \frac{1}{t}\sum_{k=1}^{t}\sum_{j=1}^{t}\left(\frac{p-n-s}{p}\right)^{t-k}\frac{n+s}{p}\|\boldsymbol{w}_k^* - \boldsymbol{w}_j^*\|^2$$

$$+ \frac{p\sigma^2}{p-n-s-1}\left[1 - \left(\frac{p-n-s}{p}\right)^t\right]$$

$$= \frac{1}{t}\sum_{k=1}^{t}\lambda^t\|\boldsymbol{w}_k^*\|^2 + \frac{1}{t}\sum_{k=1}^{t}\sum_{j=1}^{t}\frac{n+s}{p}\lambda^{t-k}\|\boldsymbol{w}_k^* - \boldsymbol{w}_j^*\|^2$$

$$+ \frac{p\sigma^2}{p-n-s-1}\left(1 - \lambda^t\right)$$

In the underparameterized regime ($p < n + s$), we have

$$\mathbb{E}[L_i(\boldsymbol{w}_t)] = \|\boldsymbol{w}_t^* - \boldsymbol{w}_i^*\|^2 + \frac{p\sigma^2}{n + s - p - 1} \tag{54}$$

Thus, we can derive the expression for the forsight error as

$$\mathbb{E}[\mathcal{G}(\boldsymbol{w}_t)] = \mathbb{E}\left[\frac{1}{t}\sum_{k=1}^{t}\|\boldsymbol{w}_t - \boldsymbol{w}_k^*\|^2\right]$$

$$= \frac{1}{t}\sum_{k=1}^{t}\mathbb{E}\|\boldsymbol{w}_t - \boldsymbol{w}_k^*\|^2 \tag{55}$$

$$= \frac{1}{t}\sum_{k=1}^{t}\|\boldsymbol{w}_t^* - \boldsymbol{w}_k^*\|^2 + \frac{p\sigma^2}{n + s - p - 1}.$$

## D.1 PROOF OF PROPOSITION 1

Under overparameterization ($p > n + s$), the expression for the foresight error for $T = 2$ is given by

$$\mathbb{E}[\mathcal{G}(\widehat{\boldsymbol{w}}_2)] = \frac{1}{2}(1 - \lambda^2)\|\boldsymbol{w}_2^* - \boldsymbol{w}_1^*\|^2 + \frac{1}{2}\lambda^2(\|\boldsymbol{w}_1^*\|^2 + \|\boldsymbol{w}_2^*\|^2) + \frac{p\sigma^2(1 - \lambda^2)}{p - n - s - 1} \tag{56}$$

and then we have

$$2\mathbb{E}[\mathcal{G}(\widehat{\boldsymbol{w}}_2)] = (1 - \lambda^2)\|\boldsymbol{w}_2^* - \boldsymbol{w}_1^*\|^2 + \lambda^2(\|\boldsymbol{w}_1^*\|^2 + \|\boldsymbol{w}_2^*\|^2) + \frac{2p\sigma^2(1 - \lambda^2)}{p - n - s - 1}$$

$$= (1 - \lambda^2 + \lambda - \lambda)\|\boldsymbol{w}_2^* - \boldsymbol{w}_1^*\|^2 + \lambda^2\|\boldsymbol{w}_1^*\|^2 + \lambda^2\|\boldsymbol{w}_2^*\|^2 + \frac{2p\sigma^2(1 - \lambda^2)}{p - n - s - 1}$$

$$= (\lambda - \lambda^2)\|\boldsymbol{w}_2^* - \boldsymbol{w}_1^*\|^2 + (1 - \lambda)\|\boldsymbol{w}_2^* - \boldsymbol{w}_1^*\|^2 + \lambda^2\|\boldsymbol{w}_1^*\|^2$$

$$\quad + \lambda^2\|\boldsymbol{w}_2^*\|^2 + \frac{2p\sigma^2(1 - \lambda^2)}{p - n - s - 1}$$

$$= (1 - \lambda)\|\boldsymbol{w}_2^* - \boldsymbol{w}_1^*\|^2 + \lambda^2\|\boldsymbol{w}_1^*\|^2 + \mathbb{E}[\mathcal{A}(\widehat{\boldsymbol{w}}_2)] + \frac{p\sigma^2(1 - \lambda^2)}{p - n - s - 1} \tag{57}$$

$$= (1 - \lambda)\|\boldsymbol{w}_2^* - \boldsymbol{w}_1^*\|^2 + (\lambda^2 + \lambda - \lambda)\|\boldsymbol{w}_1^*\|^2 + \mathbb{E}[\mathcal{A}(\widehat{\boldsymbol{w}}_2)]$$

$$\quad + \frac{p\sigma^2(1 - \lambda^2 + \lambda - \lambda)}{p - n - s - 1}$$

$$= (1 - \lambda)\|\boldsymbol{w}_2^* - \boldsymbol{w}_1^*\|^2 + (\lambda^2 - \lambda)\|\boldsymbol{w}_1^*\|^2 + \lambda\|\boldsymbol{w}_1^*\|^2$$

$$\quad + \mathbb{E}[\mathcal{A}(\widehat{\boldsymbol{w}}_2)] + \frac{p\sigma^2(\lambda - \lambda^2)}{p - n - s - 1} + \frac{p\sigma^2(1 - \lambda)}{p - n - s - 1}$$

$$= \lambda\|\boldsymbol{w}_1^*\|^2 + \frac{p\sigma^2(1 - \lambda)}{p - n - s - 1} + \mathbb{E}[\mathcal{M}(\widehat{\boldsymbol{w}}_2)] + \mathbb{E}[\mathcal{A}(\widehat{\boldsymbol{w}}_2)]$$

Based on the parameter iteration formula in the overparameterized regime, we have

$$\boldsymbol{w_t} = (\boldsymbol{I} - \boldsymbol{U}_t(\boldsymbol{U}_t^\top\boldsymbol{U}_t)^{-1}\boldsymbol{U}_t^\top)\boldsymbol{w}_{t-1} + \boldsymbol{U}_t(\boldsymbol{U}_t^\top\boldsymbol{U}_t)^{-1}\boldsymbol{Y}_t$$

$$= (\boldsymbol{I} - \boldsymbol{U}_t(\boldsymbol{U}_t^\top\boldsymbol{U}_t)^{-1}\boldsymbol{U}_t^\top)\boldsymbol{w}_{t-1} + \boldsymbol{U}_t(\boldsymbol{U}_t^\top\boldsymbol{U}_t)^{-1}(\boldsymbol{U}_t^\top\boldsymbol{w}_t^* + \boldsymbol{z}_t) \tag{58}$$

And then we have

$$\mathbb{E}\|\boldsymbol{w}_1 - \boldsymbol{w}_1^*\|^2 = \mathbb{E}\left\|(\boldsymbol{I} - \boldsymbol{U}_1(\boldsymbol{U}_1^\top\boldsymbol{U}_1)^{-1}\boldsymbol{U}_1^\top)\boldsymbol{w}_0 + \boldsymbol{U}_1(\boldsymbol{U}_1^\top\boldsymbol{U}_1)^{-1}(\boldsymbol{U}_1^\top\boldsymbol{w}_1^* + \boldsymbol{z}_1) - \boldsymbol{w}_1^*\right\|^2$$

$$= \mathbb{E}\left\|(\boldsymbol{I} - \boldsymbol{U}_1(\boldsymbol{U}_1^\top\boldsymbol{U}_1)^{-1}\boldsymbol{U}_1^\top)\boldsymbol{w}_0 + \boldsymbol{U}_1(\boldsymbol{U}_1^\top\boldsymbol{U}_1)^{-1}\boldsymbol{U}_1^\top\boldsymbol{w}_1^*\right.$$

$$\quad \left. + \boldsymbol{U}_1(\boldsymbol{U}_1^\top\boldsymbol{U}_1)^{-1}\boldsymbol{z}_1 - \boldsymbol{w}_1^*\right\|^2 \tag{59}$$

$$= \lambda\|\boldsymbol{w}_1^*\|^2 + \frac{p\sigma^2(1 - \lambda)}{p - n - s - 1}$$

It follows from the combination of Equation (57) and (59) that the proposition holds.

# E ADDITIONAL RELATED WORKS

Recently, continual learning leveraging pre-trained models have achieved strong empirical performance (Wang et al. (2022b); McDonnell et al. (2023); Zhao et al. (2024a); Tran et al. (2025)). Unlike traditional approaches that train from randomly initialized weights, these methods exploit the representational power of pre-trained models via prompting or adapters for lightweight fine-tuning. For instance, L2P (Wang et al. (2022c)) uses a prompt pool to store prior knowledge; DualPrompt (Wang et al. (2022b)) separates public and task-specific knowledge; SLCA (Zhang et al. (2023)) adapts tasks by dynamically adjusting learning rates. Despite these advances, rehearsal remains widely used, yet its theoretical implications are still underexplored. A systematic understanding of how rehearsal influence continual learning, including their operation and constraints, is hence needed.

Moreover, we provide a detailed comparison and analysis of several rehearsal-based continual learning studies. Compared to Lin et al. (2023), both works adopt Gaussian linear regression, but they focuses on generalization and forgetting in regularization-based methods, emphasizing task ordering and benign overfitting in overparameterized models. Compared to Deng et al. (2025), they studied replay strategies, showing that sequential replay outperforms concurrent replay when tasks are dissimilar. Compared to Ding et al. (2024), they provide upper and lower bounds for the forgetting error from the perspective of stochastic gradient descent, investigating the impact of iteration step size and task ordering on forgetting performance. Compared to Zheng et al. (2024), they analyzed sampling strategies, finding that reservoir sampling requires larger models to reduce forgetting and that generalization improves with task similarity. It is worth noting that some of our theoretical findings regarding task similarity and model dimensions similar to their discoveries under overparameterization (e.g., "generalization capabilities may be weakened when tasks are dissimilar"). This is because we investigate a unified rehearsal-based method, rather than being confined to specific sampling strategies. Moreover, these findings are merely byproducts of our theoretical analysis. We primary focus on examining how rehearsal differs under parameterization mechanisms and the trade-offs between adaptability, memorability, and generalization. Furthermore, we observe that retrieval practice does not adhere to the conventional "more is better" principle; even for mitigating forgetting, there exists a lower bound on error reduction. The effectiveness of rehearsal in improving adaptability also varies significantly between underparameterized and overparameterized settings. Finally, influencing factors such as rehearsal size and task similarity were validated through simulation experiments, and the deep neural network experiments further extended our theoretical findings.

# F ORACLE AND REHEARSAL-BASED ESTIMATOR

The oracle estimator assumes that all data are available simultaneously, enabling global optimization over the entire dataset (Zhang & Yang, 2018; 2021; Bhattacharjee et al., 2022). Assume that $\bar{n} = \sum_{t=1}^{T} n_t$ represents the total number of training samples, the training process will converge to a solution $\widehat{\boldsymbol{w}}^{(\text{Ora})}$ that minimizes this training loss, i.e., $\widehat{\boldsymbol{w}}^{(\text{Ora})} := \arg\min_{\boldsymbol{w}} \sum_{t=1}^{T} \left\| \boldsymbol{X}_t^{\top} \boldsymbol{w} - \boldsymbol{y}_t \right\|^2$. When $p > \bar{n}$ (overparameterized), there exist multiple solutions that can make the training loss zero (with probability 1). In this situation, we choose the smallest $\ell_2$-norm which is defined as the solution of the following optimization problem: $\arg\min_{\boldsymbol{w}} \|\boldsymbol{w}\|^2$ subject to $(\boldsymbol{X}_{1T})^{\top} \boldsymbol{w} = \boldsymbol{y}_{1T}$, where $\boldsymbol{X}_{1T}$ and $\boldsymbol{y}_{1T}$ are obtained by stacking feature vectors and response variables from all tasks. Without sequence learning or memory constraints, the optimal parameter $\boldsymbol{w}^*$ can be estimated by directly solving the offline optimization problem (Zhao et al., 2024b; Shi et al., 2022). We further analyze the connection between the oracle and the rehearsal-based estimator in AppendixB. This serves as an ideal baseline for evaluating continual learning algorithms, which are considered optimal if they achieve performance comparable to the oracle estimator (Chavan et al., 2024; Bhat et al., 2024).

Although rehearsal-based continual learning methods have been proven effective in many practical applications, their theoretical understanding is still limited. Further, the introduction of additional memory complicates the analysis of continual learning. Our work provides explicit expressions of adaptation error, memory error, and foresight error for rehearsal-based continual learning. We explain the effectiveness of the rehearsal mechanism using linear regression models under both underparameterized and overparameterized regimes, which allows us to gain a comprehensive understanding of the factors that influence continual learning performance.

---

**Scheme 1** Oracle Estimator

---

**Initialization:** $\widehat{w}_0^{(\mathrm{ora})} = 0$
**Find the optimization problem for all tasks:**

  **if** $p < \bar{n}$ (Underparameterized) **then**
    Solving the optimization problem results in a unique solution:

$$\widehat{w}_T^{(\mathrm{Ora})} := \arg\min_{w} \left\{ \sum_{t=1}^{T} \left\| X_t^T w - y_t \right\|^2 \right\}$$

  **else if** $p > \bar{n}$ (Overparameterized) **then**
    Select the smallest $\ell_2$-norm solution among all overfitted solutions:

$$\widehat{w}_T^{(\mathrm{Ora})} := \arg\min_{w} \left\| w \right\|^2,$$

$$\mathrm{s.t.}(X_{1T})^\top w = y_{1T}$$

  **end if**
  **Return** $\widehat{w}_T$

---

**Scheme 2** Rehearsal-based Continual learning Estimator

---

**Initialization:** $\widehat{w}_0^{(\mathrm{reh})} = 0$
**Iterative update for each task** $t \in T$**:**
  **if** $p < n + s$ (Underparameterized) **then**
    Solving the optimization problem results in a unique solution:

$$\arg\min_{w} \left\| X_t^T w - y_t \right\|^2 + \sum_{i=1}^{t-1} \left\| Z_i^T w - g_i \right\|^2$$

  **else if** $p > n + s$ (Overparameterized) **then**
    Select the smallest $\ell_2$-norm solution among all overfitted solutions:

$$\widehat{w}_t^{(\mathrm{Reh})} := \arg\min_{w} \left\| w - w_{t-1} \right\|^2,$$

$$\mathrm{s.t.}(X_t)^\top w = y_t, (Z_i)^\top w = g_i, i = 1, ..., t-1$$

  **end if**
  **Return** $\widehat{w}_t$

---

# G ADDITIONAL THEORETICAL ANALYSIS

**Better adaptation of the model to the current task under the overparameterized regime requires higher inter-task similarity, which is not the case under the underparameterized regime.** Specifically, under overparameterization, the coefficient of Term A1 is positive in Equation (5). At this point, when tasks are dissimilar to each other(i.e., $\left\| w_k^* - w_T^* \right\|^2$ is large), the corresponding $\mathbb{E}[\mathcal{A}(\widehat{w}_T)]$ increases, making it detrimental to learn the current task and impairing the model's plasticity. In contrast, under the underparameterized regime in Equation (6), the optimization problem has a unique solution (see Section 3), and the task is learned independently of previous tasks, meaning that inter-task similarity no longer influences the learning of the current task. In Figure 2(c), the average adaptation error varies with model parameters for different cosine similarities between task-optimal parameters. The error is unaffected by similarity under underparameterization, while higher task similarity (e.g., red and blue curves with markers "$\times$" and inverted " Y ", respectively) reduces adaptation error under overparameterization.

**The overparameterized regime help mitigate the effects of dissimilarity between tasks and noise effects on model's adaptation performance.** Under underparameterization, $\mathbb{E}[\mathcal{A}(\widehat{w}_T)]$ increases as $\sigma$ increases and the value is at least $\frac{\sigma^2}{n+s}$, indicating that larger noise is detrimental to learning the current task. However, under overparameterization, when $p$ increase to $\infty$, Term A1 and Term $a_{noise}$ will decrease to zero. At this point, the negative effect of task dissimilarity and noise level is eliminated through more parameters even when tasks are not similar, i.e., $\left\| w_k^* - w_T^* \right\|^2$ is large. The blue curve in Figure 2(a) depicts how the adaptation error varies with model parameters when $\sigma = 3$. when underparameterized, larger $\sigma$ raises the adaptation error, while it gradually converges as $p$ increases when overparameterized, indicating reduced sensitivity to noise. A similar trend is observed in Figure 2(c), showing that the effects of task dissimilarity diminish under overparameterization.

**More parameters exerts stronger influence than rehearsal in the overparameterized regime, further enhancing the model's memory performance.** For the overparameterized regime result in Equation (7), When $p \to \infty$ and thus $\lambda^t - \lambda^i$ is close to zero, causing both Term M1, Term M2 and $m_{noise}$ will approache zero. This indicates that under the overparameterized regime, the model exhibits minimal forgetting as $p$ increases to $\infty$. Intuitively, models with more parameters are better at capturing data features and handling diverse inputs, allowing the model to retain and integrate more task-relevant information, thus exhibiting stronger memory capacity. Figure 2(d) shows the average memory error versus parameter size under different noise levels. Larger $\sigma$ results in higher

memory error in the overparameterized regime. As $p$ increases, the error for all $\sigma$ values gradually approaches zero, highlighting the positive effect of model parameters.

**Increased task similarity improves memory performance in the underparameterized regime, while it may be detrimental in the overparameterized regime.** Specifically, For the underparameterized regime result in Equation (8), the $\mathbb{E}[\mathcal{M}(\widehat{\boldsymbol{w}}_T)]$ becomes smaller when $\sum_{k=1}^{T-1} \|\boldsymbol{w}_T^* - \boldsymbol{w}_k^*\|^2$ is smaller. In the special case where $\boldsymbol{w}_1^* = \boldsymbol{w}_2^* = \cdots = \boldsymbol{w}_T^*$, $\mathbb{E}[\mathcal{M}(\widehat{\boldsymbol{w}}_T)]$ approaches zero. Intuitively, smaller differences between tasks require fewer parameter adjustments, facilitating the retention of prior knowledge. In contrast, for the overparameterized regime result in Equation (7), let $k = 1$, $j = 2$, $\lambda = \frac{1}{3}$ and $T = 4$, we have $u_{kj} = \lambda^3 - \lambda + \lambda^2 < 0$, leading to a negative coefficient for Term M1. In this situation, increasing task similarity results in larger memory error. In Figure 2(f), the average memory error varies with model parameters for different cosine similarities between task optimal parameters. Under underparameterization, higher task similarity leads to smaller memory error. While under overparameterization, high similarity can worsen memory performance when changes in task order cause negative coefficients in Equation (7), as detailed in Appendix H.

**Increased task similarity enhances model's generalization performance under both underparameterized and overparameterized regimes.** For the overparameterized result in Equation (11), the coefficient of Term G1 is always positive, leading to a decrease in $\mathbb{E}[\mathcal{G}(\widehat{\boldsymbol{w}}_T)]$ as the similarity between tasks increases. For the underparameterized result in Equation (12), when the difference between tasks is small, the $\mathbb{E}[\mathcal{G}(\widehat{\boldsymbol{w}}_T)]$ decreases accordingly. The green curve with markers "+" in Figure 3(c) depicts how the average generalization error varies with model parameters when tasks are partially similar. Higher task similarity (e.g., the blue curves with inverted " Y " markers) is associated with lower generalization error in both regimes, highlighting its positive impact on generalization performance.

**Increased parameters under overparameterization weaken the impact of rehearsal and task similarity on generalization performance.** For the underparameterized result in Equation (12), both rehearsal size and task similarity have a stronger impact on generalization performance, i.e., the $\mathbb{E}[\mathcal{G}(\widehat{\boldsymbol{w}}_T)]$ decreases with increasing task similarity and larger rehearsal size. In contrast, for the overparameterized result in Equation (11), when $p \to \infty$, we have $\lambda^T \to 1$, causing both Term G1 and $g_{noise}$ approach zero, meaning that the influence of rehearsal size and task similarity is significantly diminished. Figure 3(a) and Figure 3(c) show how the generalization error varies with model parameters $p$ when changing the noise $\sigma$ or task similarity. As $p$ increases, the effects of noise and task dissimilarity diminish in the overparameterized regime, validating the earlier insights.

# H  ADDITIONAL EXPERIMENTAL RESULTS

**Experimental details.**  For experiments on deep neural networks, we use a four-layer network with two convolutional layers and two fully-connected layers. Relu is applied to the first three layers and log-softmax to the output. The first convolutional layer is followed by a 2D maximum pooling operation. Adaptation, memory, and foresight errors are computed using Equation (2)-(4), where $\mathcal{L}(\widehat{\boldsymbol{w}})$ is defined as Log-Likelihood Loss. We use stochastic gradient descent to learn each task.

**Computational resources.**  All experiments were conducted on a GPU server running Ubuntu 20.04.6 LTS. The server was equipped with an Intel Core i9-14900k processor and utilized a single NVIDIA GeForce RTX 4090 GPU. The implementation was built using the PyTorch 2.7.0.

Table 6: Explanation of Notations

| Notation used in theorems | Meaning of the notation |
| :---: | :---: |
| $T$ | Number of tasks |
| $p$ | Number of model parameters |
| $n$ | Number of training samples for the current task |
| $s$ | Number of training samples in the replay buffer |
| $\bar{n}$ | The total number of training samples |
| $\boldsymbol{w}_t^*$ | Optimal parameters for the $t$-th task |
| $\sigma^2$ | Variance of the noise |

Figures 6 and 7 illustrate adaptation error and memory error when tasks are highly dissimilar (i.e., optimal parameters are orthogonal) and the noise level is large. In contrast, Figures 2 and 3 illustrate scenarios with more similar tasks and lower noise levels, unless otherwise specified. When the tasks are dissimilar, rehearsal mechanism has less effect on adaptation error under overparameterization, while the negative impact on memory error becomes more pronounced. Moreover, with larger noise level, increasing the number of model parameters helps further reduce adaptation error.

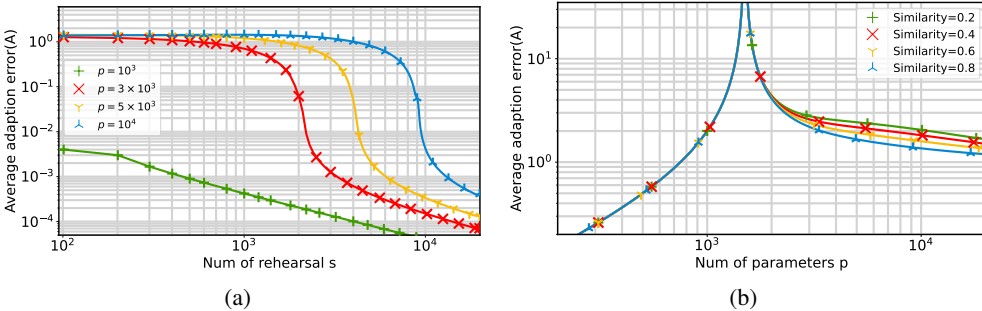

(a)  (b)

Figure 6: The trend of average adaptation error w.r.t. the number of rehearsal samples or model parameters, with $T = 8$, $n = 1000$ and $\|\boldsymbol{w}_t^*\|^2 = 1$ for all $t \in T$. Subfigure settings: (a): tasks are orthogonal; (b): $\sigma = 1$.

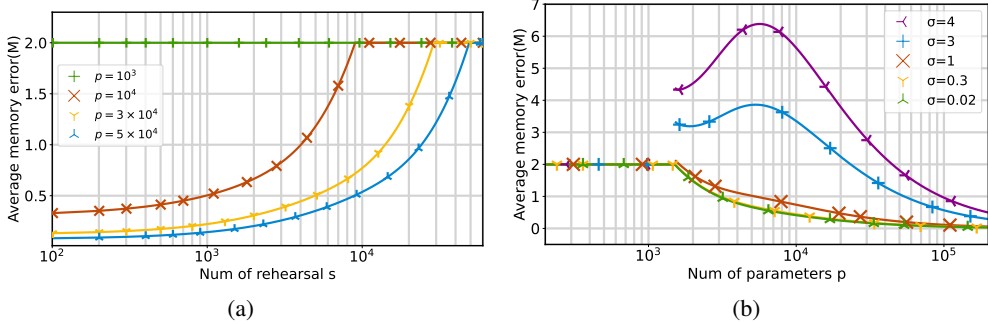

(a)  (b)

Figure 7: The trend of average memory error w.r.t. the number of rehearsal samples or model parameters when task-optimal parameters are orthogonal, with $T = 8$, $n = 1000$ and $\|\boldsymbol{w}_t^*\|^2 = 1$ for all $t \in T$.

Noted that our theoretical analysis shows that higher task similarity under overparameterization does not necessarily reduce forgetting. While such effects are uncommon in deep neural networks, recent empirical studies (Doan et al., 2021; Ramasesh et al., 2021) support this observation, showing that maximum forgetting occurs at moderate task similarity. To investigate this, we conducted experiments controlling the class overlap between the first and second tasks across four continual training tasks. In Table 7, results reveal that forgetting error initially increases and then decreases with task similarity, peaking at moderate similarity levels, consistent with empirical findings in (Evron et al., 2022).

Table 7: The forgetting error as task similarity increases under different task orderings on CIFAR-10

| Task Sequences | Order 1 | Order 2 | Order 3 |
| --- | --- | --- | --- |
| Low Similarity | 3.651 ± 0.074 | 4.099 ± 0.025 | 3.429 ± 0.029 |
| Medium Similarity | 4.274 ± 0.066 | 3.227 ± 0.033 | 3.805 ± 0.044 |
| High Similarity | 4.096 ± 0.052 | 2.917 ± 0.046 | 3.738 ± 0.010 |

Additionally, we investigated CNNs of varying depths to assess the impact of model dimensionality under overparameterization, shown in Table 8. Both memory error and generalization error decrease

as network depth increases. Within the same architecture, these errors gradually rise as the number of learned classes grows, though the rate of increase slows over time. These findings confirm the beneficial effect of greater network depth on rehearsal-based continual learning.

Table 8: Memory error and generalization error at different depths of CNN networks

| Architecture | | T1 | T2 | T3 | T4 | T5 |
|---|---|---|---|---|---|---|
| Conv.2 | Memory Error | / | 1.58 ± 0.03 | 3.82 ± 0.05 | 4.44 ± 0.16 | 4.56 ± 0.07 |
| | Generalization | 0.12 ± 0.004 | 1.05 ± 0.01 | 2.81 ± 0.04 | 3.56 ± 0.12 | 3.86 ± 0.05 |
| Conv.3 | Memory Error | / | 0.94 ± 0.03 | 2.66 ± 0.20 | 3.56 ± 0.18 | 3.40 ± 0.12 |
| | Generalization | 0.12 ± 0.002 | 0.74 ± 0.02 | 2.05 ± 0.13 | 2.90 ± 0.13 | 2.94 ± 0.10 |
| Conv.4 | Memory Error | / | 0.67 ± 0.05 | 2.01 ± 0.03 | 3.34 ± 0.12 | 3.51 ± 0.15 |
| | Generalization | 0.11 ± 0.002 | 0.67 ± 0.02 | 1.67 ± 0.03 | 2.79 ± 0.08 | 3.07 ± 0.13 |

Apart from class similarity, we also introduce semantic and distribution similarity metrics. Semantic similarity partitions the dataset into tasks with distinct semantic classes, while distribution similarity is adjusted through different levels of color perturbation. In Table 9, higher task similarity generally improves memorability. Interestingly, under both semantic and distribution similarities, negative memorization errors occur, reflecting positive knowledge transfer induced by high task similarity.

Table 9: Memory errors under different similarity metrics as the buffer size increases on CIFAR10

| Similarity Metrics | | 2000 | 4000 | 6000 | 8000 |
|---|---|---|---|---|---|
| Class Similarity | Low | 1.06 ± 0.04 | 0.61 ± 0.02 | 0.53 ± 0.03 | 0.55 ± 0.02 |
| | Medium | 0.92 ± 0.07 | 0.54 ± 0.03 | 0.51 ± 0.00 | 0.52 ± 0.02 |
| | High | 0.79 ± 0.09 | 0.37 ± 0.02 | 0.34 ± 0.01 | 0.32 ± 0.01 |
| Distribution Similarity | Low | -0.05 ± 0.00 | -0.04 ± 0.00 | -0.03 ± 0.00 | -0.02 ± 0.01 |
| | Medium | -0.07 ± 0.01 | -0.04 ± 0.01 | -0.05 ± 0.01 | -0.05 ± 0.02 |
| | High | -0.05 ± 0.01 | -0.05 ± 0.01 | -0.05 ± 0.01 | -0.05 ± 0.00 |
| Semantic Similarity | Low | 0.39 ± 0.02 | 0.32 ± 0.01 | 0.31 ± 0.02 | 0.32 ± 0.01 |
| | Medium | 0.48 ± 0.04 | 0.27 ± 0.01 | 0.24 ± 0.02 | 0.25 ± 0.01 |
| | High | -0.05 ± 0.00 | -0.04 ± 0.01 | -0.04 ± 0.00 | -0.03 ± 0.01 |

## I  USE OF LARGE LANGUAGE MODELS

The Large language models (LLMs) were used only as general-purpose writing and editing assistants to improve clarity and readability. The scientific ideas, results, and analyses were developed entirely by the authors. The LLM did not contribute to research design or result interpretation.

## J  ETHICS STATEMENT

The authors have read and adhered to the code of ethics. This work does not involve human subjects or sensitive data. All datasets used are publicly available and were processed in accordance with standard ethical guidelines. The methods and experiments introduce no harmful applications, conflicts of interest, and comply with all relevant legal and research standards throughout this research.

## K  REPRODUCIBILITY STATEMENT

Every effort has been made to ensure the reproducibility of the results. The model architectures, experimental settings, and the proofs of the theoretical results are provided in the appendix, while data preprocessing and evaluation protocols are documented in supplementary materials.

