# OpenReview forum: "Understanding the Role of Rehearsal in Continual Learning under Varying Model Capacities"
_ICLR.cc/2026/Conference — Submitted to ICLR 2026_

### Official Review · Reviewer_juEZ · 2025-10-28

**Soundness:** 3
**Presentation:** 2
**Contribution:** 2
**Rating:** 4
**Confidence:** 4

**Summary:**

This paper presents theoretical analysis of rehearsal-based methods for continual learning in terms of memory, adaptation, and generalization errors, and shows the effectiveness of the formulation with experiments on the four benchmark datasets. The analysis implies some interesting observations that more rehearsal is not always beneficial and the rehearsal affects model adaptability differently in underparameterized and overparameterized cases.

**Strengths:**

- Overall, a sound theoretical formulation is provided in a logical way.

- The experiments as well as a thorough derivation at the appendix support the main findings.

**Weaknesses:**

- The assumption of the theoretical formulation is too restricted to draw any interesting conclusion for practical algorithms based on rehearsal mechanism: the Gaussian assumption and linear regression problem for each task.

- The experiments are limited to the relative easy datasets, and more updated methods for rehearsal should be used for the comparative study.

**Questions:**

- How much effective is the proposed analysis for the practical rehearsal-based continual learning?

- Are the assumptions for the formulation reasonable for general rehearsal-based continual learning?

- How do you expect the results for the larger, practical datasets with recent advanced rehearsal-based methods?

---

> ### Author Response · Authors · 2025-11-28
> **Response to Reviewer juEZ ( Part I )**
>
> **We sincerely appreciate the reviewers’careful reading and valuable comments. Below are our responses to each point.**
>
> **Q1: The assumption of the theoretical formulation is too restricted to draw any interesting conclusion for practical algorithms based on rehearsal mechanism.**
> **A1**：Thanks for your valuable comments. We understand your concerns regarding theoretical hypothetical scenarios. **In practice,** task distributions do not necessarily follow the linear Gaussian assumption, and dynamic variations in data can lead to differences that are often uncontrollable. **Therefore,** we conduct theoretical analysis in this theoretical scenario, aiming to establish a controllable, stable, and interpretable framework that can clearly characterize the interactions among various factors. **Moreover,** analyzing linear models is an important step toward understanding deep neural networks, as demonstrated by recent studies [1][2][3]. **Within this framework,** we uncover several intriguing findings. Specifically, we discovered that increased replay does not always mitigate forgetting. And the replay mechanism can impair the model’s adaptability to the current task. **Importantly,** although the analysis is conducted within this framework, these theoretical findings are validated through experiments with deep neural networks on real datasets across longer task sequences, deeper network architectures, and diverse sampling strategies. Thank you again for your valuable comments.
>
> [1]Evron et al. “How catastrophic can catastrophic forgetting be in linear regression?”  COLT 2022.
> [2]Ju et al. On the generalization power of the overfitted three-layer neural tangent kernel model. NeurIPS 2022.
> [3]Ji et al. The power of contrast for feature learning: Atheoretical analysis. JMLR 2023.
>
>
> **Q2: The experiments are limited to the relative easy datasets.**
> **A2**：Thank you for your suggestion. The deep neural network experiments in this study use CIFAR-100, CIFAR-10, Tiny-ImageNet, and MNIST, covering most benchmark datasets commonly used for continual learning [4][5]. And according to your suggestion, we also extended experiments to the more challenging CUB-200. The experimental results are as follows.
>
> |Buffer Size|High similarity|Medium similarity|Low similarity|
> |-----------------|---------------|----------------|--------------|
> |0|4.518|5.721|5.895|
> |200|4.085|4.748|5.106|
> |400|5.049|5.683|5.495|
> |600|5.933|5.097|6.420|
> |800|6.127|5.811|6.421|
> |1000|6.363|5.897|6.400|
> |Average|5.346|5.493|5.956|
>
> As shown in the table, memory error exhibited a noticeable decrease when replay samples were initially introduced. However, further increasing the number of replay samples leads to a rise in memory error.
>
> |Tasks|T3|T6|T12|T16|T20|Avg.|
> |-------------|---|---|---|---|---|---|
> |0%|2.146|2.153|2.329|2.437|2.270|2.267|
> |5%|2.125|1.735|2.310|2.522|2.739|2.286|
> |10%|2.309|2.054|4.513|5.270|5.267|3.882|
> |50%|2.296|2.009|2.373|2.333|2.427|2.287|
>
> As shown in the table, adaptation error on the current task increases as the replay buffer grows, with the largest difference observed when replaying 10% of samples. Moreover, according to your suggestion, we also compared existing algorithms-**EWC [6], LwF [7], iCaRL [8], DER [9], FOSTER [10], MEMO [11], and the latest rehearsal method CSReL [12]**. Since EWC and LwF lack replay mechanisms, we added replay-enabled modules to evaluate performance under replay scenarios. Baseline settings and evaluation followed [4] to ensure fair comparisons. The results of average current accuracy are as follows.
>
> |Methods|2000|4000|6000|8000|10000|
> |-------|----|----|----|----|-----|
> |EWC|86.20|84.60|83.17|81.91|80.11|
> |LwF|73.35|73.86|72.92|73.90|74.76|
> |iCaRL|84.59|82.98|81.59|80.89|80.19|
> |DER|73.87|73.57|72.40|72.86|72.40|
> |FOSTER|79.48|84.71|84.35|83.95|83.10|
> |MEMO|68.10|66.78|66.01|65.56|65.16|
> |CSReL|73.29|68.20|68.13|67.96|60.59|
>
> As shown in the table, the average current accuracy of most algorithms decreases with increasing number of replays. CSRel exhibits a more pronounced decline, while DER and MEMO show a gradual decrease. The table below reports the experimental results on forgetting rate variation.
>
> |Methods|Forg. 1000| Diff. 1000|Forg. 2000 |Diff. 2000 |Forg. 3000|Diff. 3000|Forg. 4000|Diff. 4000|Forg. 5000|Diff. 5000|Forg. 6000|Diff. 6000|
> |--|--|--|--|--|--|--|--|--|--|--|--|--|
> |EWC|61.744|\\|49.513|12.232|42.356|7.157|37.633|4.722|33.322|4.311|30.689|2.633|
> |LwF|40.256|\\|34.567|5.689|31.533|3.033|30.622|0.911|28.422|2.200|26.367|2.056|
> |iCaRL|59.744|\\|46.356|13.389|40.844|5.511|35.811|5.033|31.856|3.956|31.013|0.843|
> |DER|23.467|\\|16.656|6.811|12.522|4.133|10.322|2.200|8.489|1.833|7.456|1.033|
> |FOSTER|18.589|\\|27.156|-8.567|30.267|-3.111|31.456|-1.189|31.090|0.366|30.811|0.279|
> |MEMO|25.344|\\|19.111|6.233|14.956|4.156|12.700|2.256|10.200|2.500|8.744|1.456|
> |CSReL|56.733|\\|44.689|12.044|31.411|13.278|27.822|3.589|27.833|-0.011|27.439|0.394|

---

> > ### Author Response · Authors · 2025-11-28
> > **Response to Reviewer juEZ ( Part II )**
> >
> > As shown in the table, when the replay is introduced in the initial phase, the forgetting rate decreases for most algorithms. **However,** as the replay scale gradually increases, the rate of decrease diminishes. When the replay size reaches around 6000 samples, the additional forgetting reduction becomes marginal-about 0.3% for some methods. **Furthermore,** FOSTER shows strong forgetting resistance only at a buffer size of 1000, after which its forgetting rate increases as replay grows. **This further demonstrates that** larger replay scales are not necessarily better: they not only exhibit diminishing returns in mitigating forgetting but can also impair learning on the current task. Thank you again for your valuable comments.
> >
> > [4]Zhou et al. Class-incremental learning: A survey. TPAMI 2024.
> > [5]Wickramasinghe et al. Continual learning: A review of techniques, challenges, and future directions[J]. TAI 2023.
> > [6]Kirkpatrick et al. Overcoming catastrophic forgetting in neural networks. PNAS 2017.
> > [7]Li et al. Learning without forgetting. TPAMI 2017.
> > [8]Rebuffi et al. icarl: Incremental classifier and representation learning. CVPR 2017.
> > [9]Yan et al. Der: Dynamically expandable representation for class incremental learning. CVPR 2021.
> > [10]Wang et al. Foster: Feature boosting and compression for class-incremental learning. ECCV 2022.
> > [11]Zhou et al. A Model or 603 Exemplars: Towards Memory-Efficient Class-Incremental Learning. ICLR 2023.
> > [12]Tong et al. Coreset selection via reducible loss in continual learning. ICLR 2025.
> >
> >
> > **Q3: How much effective is the proposed analysis for the practical rehearsal-based continual learning?**
> > **A3**： We sincerely appreciate the reviewers' valuable comments. However, we must clarify the contributions and significance of this work. **Firstly,** although replay mechanisms are widely used in empirical studies, their theoretical effects on continual learning remain insufficiently understood. **For instance,** when continual learning can benefit from replay, and whether replay poses potential risks, remain to be explored. **Secondly,** contrary to common intuition, our analysis shows that increasing replay scale does not always improve performance. Larger replay buffers may impair adaptability to the current task, and even forgetting reduction has a lower bound. The proposed proposition shows that strong generalization requires not only good adaptability and memorability but also strong initial-task performance, aligning with recent empirical findings [13][14][15]. **These theoretical findings** offer a clear foundation for understanding both the role and the limitations of rehearsal mechanisms in continual learning. **Finally,** these theoretical insights are validated through numerical simulations and extensive deep neural network experiments on real-world datasets. Thank you again for your valuable comments.
> >
> > [13]Wang et al. A comprehensive survey of continual learning: Theory, method and application. TPAMI 2024.
> > [14]Shi et al. Mimicking the oracle: An initial phase decorrelation approach for class incremental learning. CVPR 2022.
> > [15]Zhou et al. Forward compatible few-shot class-incremental learning. CVPR 2022.
> >
> >
> > **Q4: Are the assumptions for the formulation reasonable for general rehearsal-based continual learning?**
> > **A4**：Thank you for your suggestion. In theoretical analysis, the Gaussian assumption is adopted to obtain clear and concise closed-form results, allowing for a more precise characterization of how different factors interact and evolve. **It is noteworthy that** despite this assumption, these theoretical findings were experimentally extended to deep neural networks across multiple real-world datasets, encompassing analyses under **different real-world datasets, diverse network architectures, varied sampling strategies, and varying numbers of tasks**. This demonstrates that even when real data do not strictly follow the assumed distribution, the core structure revealed by this theory remains robust. **Moreover,** we further relaxed assumption by allowing varying sample sizes and heterogeneous noise levels across tasks, improving the applicability of this analysis. The main theoretical results are as follows.
> >
> >
> > Under the relaxed assumption, the sample size and noise satisfying $n\_1\ne n\_2 \ne ...\ne n\_T$ and $\sigma\_1 \ne \sigma\_2 \ne ... \ne \sigma\_T$, and the optimization objective under the underparameterized regime is given by: $\widehat{\boldsymbol{w}}\_{t}^{(\text {Reh})} := \underset{\boldsymbol{w}}{\arg \min } \left||\boldsymbol{X}\_{t}^{\top} \boldsymbol{w}-\boldsymbol{y}\_{t}\right||^{2}  + \sum\_{i=1}^{t-1} \left||\boldsymbol{Z}\_{i}^{\top} \boldsymbol{w} -\boldsymbol{g}\_{i}\right||^{2}$. In this situation, we can further calculate the adaptation, memory, and generalization errors, and the remaining analysis is provided in Part III.

---

> > > ### Author Response · Authors · 2025-11-28
> > > **Response to Reviewer juEZ ( Part III )**
> > >
> > > According to the analysis above, under relaxed assumptions, the adaptation error can be calculated as follows.
> > > $$
> > > \begin{aligned}
> > > &\mathbb{E}[A(\boldsymbol{w}\_{t})]= \mathbb{E}||\boldsymbol{w}\_{t} - \boldsymbol{w}\_{t}^{\*}||^{2} \\\\
> > > =&\mathbb{E}||\boldsymbol{w}\_t^\* +   (\boldsymbol{X}\_{t}\boldsymbol{X}\_{t}^{\top}+\sum\_{i=1}^{t-1}\boldsymbol{Z}\_{i}\boldsymbol{Z}\_{i}^{\top})^{-1} (\boldsymbol{X}\_{t}\boldsymbol \varepsilon \_t+\sum\_{i=1}^{t-1}\boldsymbol{Z}\_{i}\boldsymbol \varepsilon \_i)    - \boldsymbol{w}\_{t}^{\*}||^{2} \\\\
> > > =& ||\boldsymbol{w}\_t^\* -\boldsymbol{w}\_{t}^{\*}||^{2}   +  \mathbb{E}||  (\boldsymbol{X}\_{t}\boldsymbol{X}\_{t}^{\top}+\sum\_{i=1}^{t-1}\boldsymbol{Z}\_{i}\boldsymbol{Z}\_{i}^{\top})^{-1} (\boldsymbol{X}\_{t}\boldsymbol \varepsilon \_t+\sum\_{i=1}^{t-1}\boldsymbol{Z}\_{i}\boldsymbol \varepsilon \_i) ||^{2} \\\\
> > > =&  \frac{p\sigma^{2}\_t}{n + s - p - 1}\\\\
> > > \end{aligned}
> > > $$
> > > The memory error can be calculated as
> > > $$
> > > \begin{aligned}
> > > & \mathbb{E}[\mathcal{M}(\boldsymbol{w}\_t)] \\\\
> > > =& \mathbb{E}\left[\frac{1}{t-1} \sum\_{k=1}^{t-1} \left( ||\boldsymbol{w}\_{t} - \boldsymbol{w}\_{k}^{\*}||^{2} - ||\boldsymbol{w}\_{k} - \boldsymbol{w}\_{k}^{\*}||^{2} \right) \right] \\\\
> > > =& \frac{1}{t-1} \sum\_{k=1}^{t-1} \left[
> > > 	||\boldsymbol{w}\_{t}^{\*} - \boldsymbol{w}\_{k}^{\*}||^{2}
> > > 	+ \frac{p\sigma^{2}\_t}{n\_t + s - p - 1}
> > > 	- ||\boldsymbol{w}\_{k}^{\*} - \boldsymbol{w}\_{k}^{\*}||^{2}
> > > 	- \frac{p\sigma^{2}\_k}{n\_k + s - p - 1}
> > > 	\right] \\\\
> > > =& \frac{1}{t-1} \sum\_{k=1}^{t-1} ||\boldsymbol{w}\_{t}^{\*} - \boldsymbol{w}\_{k}^{\*}||^{2} - \frac{1}{t-1} \sum\_{k=1}^{t-1}\frac{p\sigma^{2}\_k}{n\_k + s - p - 1} +\frac{p\sigma^{2}\_t}{n\_t + s - p - 1}\\\\
> > > \end{aligned}
> > > $$
> > > Then, we have the generalization error
> > > $$
> > > \begin{aligned}
> > > 	\mathbb{E}[\mathcal{G}\left(\boldsymbol{w}\_t\right)]
> > > 	&= \mathbb{E}\left[\frac{1}{t}\sum\_{k=1}^{t}||\boldsymbol{w}\_{t} - \boldsymbol{w}\_{k}^{\*}||^{2} \right] \\\\
> > > 	&= \frac{1}{t} \sum\_{k=1}^{t} \mathbb{E}||\boldsymbol{w}\_t^\* +   (\boldsymbol{X}\_{t}\boldsymbol{X}\_{t}^{\top}+\sum\_{i=1}^{t-1}\boldsymbol{Z}\_{i}\boldsymbol{Z}\_{i}^{\top})^{-1} (\boldsymbol{X}\_{t}\boldsymbol \varepsilon \_t+\sum\_{i=1}^{t-1}\boldsymbol{Z}\_{i}\boldsymbol \varepsilon \_i)    - \boldsymbol{w}\_{k}^{\*}||^{2}  \\\\
> > > 	&= \frac{1}{t}\sum\_{k=1}^{t} \left[ ||\boldsymbol{w}\_{t}^{\*} - \boldsymbol{w}\_{k}^{\*}||^{2} + \frac{p\sigma^{2}\_t}{n\_t + s - p - 1}\right]
> > > \end{aligned}
> > > $$
> > >
> > >  After relaxing the assumptions, the theoretical results show that the adaptation error in the underparameterized regime is no longer constant but varies with sample sizes and noise levels. Nonetheless, its overall trend-decreasing with larger replay scales-remains consistent with the primary theoretical analysis, further supporting the robustness of these theoretical analyses.   Thank you again for your valuable comments.
> > >
> > >
> > >
> > >
> > > **Q5: How do you expect the results for the larger, practical datasets with recent advanced rehearsal-based methods?**
> > > **A5**：Thank you for your suggestion. In A2, we expanded experiments to the fine-grained CUB-200 dataset. We also conducted comparisons on classic continual learning algorithms including EWC[6], LwF[7], iCaRL[8], Der[9], Foster[10], MEMO[11], and the latest replay method CSReL[12]. Moreover, we extended analysis to the more challenging ImageNet-R dataset. The experimental results are as follows.
> > >
> > > |Methods|2000|4000|6000|
> > > |-------|----|----|----|
> > > |iCaRL|53.567|50.893|49.992|
> > > |EWC|53.053|51.476|49.690|
> > > |DER|46.211|45.839|44.209|
> > > |FOSTER|29.473|40.846|44.229|
> > > |MEMO|19.256|17.608|16.218|
> > >
> > > As shown in the table, the model’s accuracy on the current task gradually declines as the replay scale increases on ImageNet-R, demonstrating the negative impact on the model’s adaptability.
> > >
> > > |Buffer Size|High similarity|Medium similarity|Low similarity|
> > > |-------------------|---------------|----------------|--------------|
> > > |0|7.721|9.192|11.725|
> > > |200|5.673|6.639|6.908|
> > > |600|6.687|6.367|7.150|
> > > |800|6.795|6.707|7.320|
> > > |1200|7.148|6.964|7.903|
> > > |1600|6.917|7.105|7.580|
> > > |2000|7.381|6.995|7.751|
> > > |Average|6.903|7.138|8.048|
> > >
> > > As shown in the table, introducing replay initially improves performance. However, as the number of replays increases, the improvement in memory performance diminishes. Due to time constraints, the partial results are provided, but every effort has been made to include the informative findings.
> > >
> > > **Finally, thank you again for your constructive feedback and valuable comments. We look forward to your response and are happy to answer any questions or comments.**

---

### Official Review · Reviewer_EQuC · 2025-10-29

**Soundness:** 3
**Presentation:** 3
**Contribution:** 2
**Rating:** 6
**Confidence:** 3

**Summary:**

The authors theoretically analyze adaptation, memory, and generalization errors for rehearsal based continual learning within a linear Gaussian regression framework, covering both underparameterized and overparameterized regimes. They also validate their theoretical insights on several benchmarks.

**Strengths:**

1. The theoretical and experimental analyses are comprehensive. In particular, the paper studies rehearsal based continual learning in both the underparameterized and the overparameterized regimes, and evaluates on commonly used continual learning benchmarks to verify the theoretical results.
2. The paper offers several interesting insights, for example that more rehearsal does not always improve memorability, and that rehearsal enhances adaptability under underparameterization but can be provably detrimental under overparameterization.

**Weaknesses:**

1. The theoretical results rely heavily on the linear model and the Gaussian assumption. We do not deny the value of these assumptions: the linear Gaussian regime yields many elegant results with practical guidance. Nevertheless, adding a clear discussion of their limitations would make the paper more complete.
2. The theoretical novelty is limited, and parts of the results overlap with prior work (e.g., Banayeeanzade2024, Zheng2024). Nevertheless, the authors provide a relatively comprehensive discussion and accompanying experiments.

**Questions:**

1. If the Gaussian assumption is relaxed, do the theoretical results still hold? If so, could you add a simple synthetic data experiment to verify this?
2. The insights in lines 282--289 and 335--341 are derived for the two task case. Do these insights extend to $T>2$?
3. In the theoretical analysis, each update assumes using the entire buffer, whereas in standard training one typically draws random mini batches from the buffer. How is rehearsal implemented in your experiments? Do you sample mini batches from the buffer at random, or do you train with the full buffer at each step? We expect that drawing random mini batches from the buffer may lead to different phenomena and thus deserves discussion.

---

> ### Author Response · Authors · 2025-11-28
> **Response to Reviewer EQuC  ( Part I  )**
>
> **We sincerely appreciate the reviewers’ positive and constructive feedback. Below are our responses to each point.**
>
> **Q1: The theoretical results rely on the linear model and the Gaussian assumption. Adding a clear discussion of their limitations would make the paper more complete.**
> **A1**：Thank you for your suggestions. We sincerely appreciate your recognition of the theoretical value of analysis under the linear Gaussian model, which is important for refining this research. As you pointed out, the linear Gaussian model, thanks to its mathematical tractability, allows for clearer and practical theoretical results. **In existing continual learning theory,** commonly used assumptions include the linear Gaussian assumption [1][2][3], the fixed design assumption [4][5], and the fourth-moment assumption [6]. **The linear Gaussian assumption** leverages properties such as orthogonal projection to separate factors affecting performance and quantify results. However, it often struggles to capture the nonlinear nature of deep networks and the structural distribution of noise. **The fixed design assumption** treats the input data matrix as non-random; analyzing the geometric structure of data across tasks can reveal interference between tasks but neglects the inherent randomness in real data distributions. **The fourth-moment assumption** accommodates broader data distributions and enables analysis of noise and data perturbations, but it often fails to capture structural noise and does not yield explicit closed-form expressions. Therefore, each assumption has distinct strengths and limitations. Selecting the appropriate theoretical framework for research context is therefore essential to effectively address diverse research topics under continual learning. Thank you again for your valuable comments, and we have added this analysis into the revised version.
>
> [1]Evron et al. "How catastrophic can catastrophic forgetting be in linear regression?". COLT 2022.
> [2]Goldfarb et al. Analysis of catastrophic forgetting for random orthogonal transformation tasks in the overparameterized regime. AISTATS 2023.
> [3]Li et al. Theory on Mixture-of-Experts in Continual Learning. ICLR 2025.
> [4]Li et al. Fixed design analysis of regularization-based continual learning. CoLLAs 2023.
> [5]Zhao et al. A Statistical Theory of Regularization-Based Continual Learning. ICML 2024.
> [6]Ding et al. Understanding Forgetting in Continual Learning with Linear Regression. ICML 2024.
>
>
> **Q2: The theoretical novelty is limited, and parts of the results overlap with prior work. Nevertheless, the authors provide a relatively comprehensive discussion and accompanying experiments.**
> **A2**： Thank you for pointing out this. We understand concerns regarding potential risks and appreciate the opportunity to clarify this point. **Firstly,** we are appreciative that you recognize the comprehensiveness of our theoretical and experimental analysis and the interest of these insights. **Secondly,** theoretical research in continual learning has expanded rapidly, with studies examining the impact of model parameters, the impact of sampling strategies, and the effect of task similarity, and we provide a detailed comparison of these works in Appendix E. **Finally,** based on this review, we summarize the key differences between this work and prior studies as follows.
>
> **Compared to [Banayeeanzade et al., 2024],** they studied the impact of model dimensions in multi-task and replay learners and found that larger models are less prone to forgetting, their focus was on the positive effects at the model dimension in continual learning.
> **Compared to [Zheng et al., 2024],** they examined the effect of different sampling strategies on continual learning and found that larger models with reservoir sampling help reduce forgetting, their analysis focused on different sampling strategies.
>
> Finally, please allow us clarify on the contributions and significance of this work. **This theoretical study examines the impact of replay size on performance and uncovers several interesting findings: larger rehearsal sizes do not always improve memory performance, and rehearsal mechanisms can hinder a model’s adaptability to the current task, which are not discussed in prior research.** We support these insights through theoretical analysis, numerical simulations, and experiments with deep neural networks on real-world datasets. Thank you again for your valuable comments, and the remaining analysis is provided in Part II.

---

> > ### Author Response · Authors · 2025-11-28
> > **Response to Reviewer EQuC ( Part II )**
> >
> > **Q3: If the Gaussian assumption is relaxed, do the theoretical results still hold. If so, could you add a simple synthetic data experiment to verify this.**
> > **A3**：Thank you for your valuable comments. In theoretical analysis, the Gaussian distribution is widely used in fields such as statistical learning, signal processing, and continual learning, due to its strong mathematical tractability . This theoretical study also leverages the properties of the Gaussian distribution to derive clear closed-form expressions across multiple dimensions, and future research will further explore analyses beyond the Gaussian distribution. Furthermore, according to your suggestion, to improve the scalability of our theoretical framework, we relax the model assumptions to account for tasks with varying noise levels and sample sizes. The results are as follows.
> >
> > Under the relaxed assumption, the sample size and noise satisfying $n\_1\ne n\_2 \ne ...\ne n\_T$ and $\sigma\_1 \ne \sigma\_2 \ne ... \ne \sigma\_T$, and we can calculate the expected $\ell\_2$-norm, which is given by
> > $$
> > \begin{aligned}
> > \mathbb{E}[L\_{i}(\boldsymbol{w}\_{t})] =& \mathbb{E}||\boldsymbol{w}\_{t} - \boldsymbol{w}\_{i}^{\*}||^{2} \\\\
> > =& \mathbb{E}||\boldsymbol{w}\_t^\* +   (\boldsymbol{X}\_{t}\boldsymbol{X}\_{t}^{\top}+\sum\_{i=1}^{t-1}\boldsymbol{Z}\_{i}\boldsymbol{Z}\_{i}^{\top})^{-1} (\boldsymbol{X}\_{t}\boldsymbol \varepsilon \_t+\sum\_{i=1}^{t-1}\boldsymbol{Z}\_{i}\boldsymbol \varepsilon \_i)    - \boldsymbol{w}\_{i}^{\*}||^{2} \\\\
> > =& \mathbb{E} ||\boldsymbol{w}\_{t}^{\*} - \boldsymbol{w}\_{i}^{\*}||^{2} + \mathbb{E} || (\boldsymbol{X}\_{t}\boldsymbol{X}\_{t}^{\top}+\sum\_{i=1}^{t-1}\boldsymbol{Z}\_{i}\boldsymbol{Z}\_{i}^{\top})^{-1} (\boldsymbol{X}\_{t}\boldsymbol \varepsilon \_t+\sum\_{i=1}^{t-1}\boldsymbol{Z}\_{i}\boldsymbol \varepsilon \_i)||^{2} \\\\
> > =& \mathbb{E} ||\boldsymbol{w}\_{t}^{\*} - \boldsymbol{w}\_{i}^{\*}||^{2} +   \frac{p\sigma^{2}\_t}{n + s - p - 1}\\\\
> > \end{aligned}
> > $$
> > Then, the adaptation error can be calculated as
> > $$
> > \begin{aligned}
> > &\mathbb{E}[A(\boldsymbol{w}\_{t})]= \mathbb{E}||\boldsymbol{w}\_{t} - \boldsymbol{w}\_{t}^{\*}||^{2} \\\\
> > =&\mathbb{E}||\boldsymbol{w}\_t^\* +   (\boldsymbol{X}\_{t}\boldsymbol{X}\_{t}^{\top}+\sum\_{i=1}^{t-1}\boldsymbol{Z}\_{i}\boldsymbol{Z}\_{i}^{\top})^{-1} (\boldsymbol{X}\_{t}\boldsymbol \varepsilon \_t+\sum\_{i=1}^{t-1}\boldsymbol{Z}\_{i}\boldsymbol \varepsilon \_i)    - \boldsymbol{w}\_{t}^{\*}||^{2} \\\\
> > =& ||\boldsymbol{w}\_t^\* -\boldsymbol{w}\_{t}^{\*}||^{2}   +  \mathbb{E}||  (\boldsymbol{X}\_{t}\boldsymbol{X}\_{t}^{\top}+\sum\_{i=1}^{t-1}\boldsymbol{Z}\_{i}\boldsymbol{Z}\_{i}^{\top})^{-1} (\boldsymbol{X}\_{t}\boldsymbol \varepsilon \_t+\sum\_{i=1}^{t-1}\boldsymbol{Z}\_{i}\boldsymbol \varepsilon \_i) ||^{2} \\\\
> > =&  \frac{p\sigma^{2}\_t}{n + s - p - 1}\\\\
> > \end{aligned}
> > $$
> > And we can calculate the memory error
> > $$
> > \begin{aligned}
> > & \mathbb{E}[\mathcal{M}(\boldsymbol{w}\_t)] \\\\
> > =& \mathbb{E}\left[\frac{1}{t-1} \sum\_{k=1}^{t-1} \left( ||\boldsymbol{w}\_{t} - \boldsymbol{w}\_{k}^{\*}||^{2} - ||\boldsymbol{w}\_{k} - \boldsymbol{w}\_{k}^{\*}||^{2} \right) \right] \\\\
> > =& \frac{1}{t-1} \sum\_{k=1}^{t-1} \left[
> >     ||\boldsymbol{w}\_{t}^{\*} - \boldsymbol{w}\_{k}^{\*}||^{2}
> >     + \frac{p\sigma^{2}\_t}{n\_t + s - p - 1}
> >     - ||\boldsymbol{w}\_{k}^{\*} - \boldsymbol{w}\_{k}^{\*}||^{2}
> >     - \frac{p\sigma^{2}\_k}{n\_k + s - p - 1}
> >     \right] \\\\
> > =& \frac{1}{t-1} \sum\_{k=1}^{t-1} ||\boldsymbol{w}\_{t}^{\*} - \boldsymbol{w}\_{k}^{\*}||^{2} - \frac{1}{t-1} \sum\_{k=1}^{t-1}\frac{p\sigma^{2}\_k}{n\_k + s - p - 1} +\frac{p\sigma^{2}\_t}{n\_t + s - p - 1}\\\\
> > \end{aligned}
> > $$
> > Finally, we have the generalization error
> > $$
> > \begin{aligned}
> >     \mathbb{E}[\mathcal{G}\left(\boldsymbol{w}\_t\right)]
> >     &= \mathbb{E}\left[\frac{1}{t}\sum\_{k=1}^{t}||\boldsymbol{w}\_{t} - \boldsymbol{w}\_{k}^{\*}||^{2} \right] \\\\
> >     &= \frac{1}{t} \sum\_{k=1}^{t} \mathbb{E}||\boldsymbol{w}\_t^\* +   (\boldsymbol{X}\_{t}\boldsymbol{X}\_{t}^{\top}+\sum\_{i=1}^{t-1}\boldsymbol{Z}\_{i}\boldsymbol{Z}\_{i}^{\top})^{-1} (\boldsymbol{X}\_{t}\boldsymbol \varepsilon \_t+\sum\_{i=1}^{t-1}\boldsymbol{Z}\_{i}\boldsymbol \varepsilon \_i)    - \boldsymbol{w}\_{k}^{\*}||^{2}  \\\\
> >     &= \frac{1}{t}\sum\_{k=1}^{t} \left[ ||\boldsymbol{w}\_{t}^{\*} - \boldsymbol{w}\_{k}^{\*}||^{2} + \frac{p\sigma^{2}\_t}{n\_t + s - p - 1}\right]
> > \end{aligned}
> > $$
> >
> > After relaxing assumptions, adaptation error under underparameterization varies with sample size and noise level. Nevertheless, adaptation error still decreases gradually as replay increases, consistent with the previous analysis. We also validated this analysis experimentally using synthetic datasets, and the remaining analysis is provided in Part III.

---

> > > ### Author Response · Authors · 2025-11-28
> > > **Response to Reviewer EQuC ( Part III )**
> > >
> > > Based on the theoretical analysis above, the results for the adaptation error are as follows.
> > >
> > >
> > > |Num of rehearsal|p=200|p=400|p=600|p=800|p=1000|
> > > |-----------------|-----|-----|-----|-----|------|
> > > |0|0.572|1.334|2.402|4.005|6.678|
> > > |200|0.500|1.144|2.002|3.203|5.006|
> > > |600|0.400|0.889|1.501|2.287|3.336|
> > > |800|0.364|0.800|1.334|2.001|2.859|
> > > |1200|0.308|0.667|1.091|1.601|2.223|
> > > |1600|0.267|0.572|0.923|1.334|1.819|
> > > |1800|0.250|0.534|0.857|1.231|1.667|
> > > |2000|0.235|0.500|0.800|1.143|1.539|
> > >
> > > As shown in the table, under relaxed assumptions, adaptation error in the underparameterized regime decreases with increasing replay size, consistent with the theoretical analysis. The results of memory errors are as follows.
> > >
> > > |Num of rehearsal|p=200|p=400|p=600|p=800|p=1000|
> > > |---------------|-----|-----|-----|-----|------|
> > > |0|0.004|0.006|0.003|-0.055|-0.599|
> > > |200|0.003|0.005|0.008|0.003|-0.069|
> > > |600|0.001|0.001|0.003|0.007|0.011|
> > > |800|0.001|-0.001|0.000|0.003|0.008|
> > > |1200|0.000|-0.002|-0.004|-0.004|-0.002|
> > > |1600|0.000|-0.003|-0.006|-0.008|-0.008|
> > > |1800|0.000|-0.004|-0.006|-0.009|-0.010|
> > > |2000|0.000|-0.004|-0.007|-0.010|-0.012|
> > >
> > > As shown in the table, memory error no longer remains zero at high task similarity, but instead shows a slight continued decrease, which is due to the incomplete cancellation of the noise term. The results for generalization error are as follows.
> > >
> > > |Num of rehearsal|p=200|p=400|p=600|p=800|p=1000|
> > > |---------------|-----|-----|-----|-----|------|
> > > |0|0.54727|1.339878|2.407835|4.010439|6.683229|
> > > |200|0.505745|1.149107|2.007101|3.208636|5.01169|
> > > |600|0.405633|0.894816|1.506371|2.292781|3.341546|
> > > |800|0.369234|0.805833|1.339507|2.006683|2.864618|
> > > |1200|0.313243|0.672377|1.096838|1.606233|2.22889|
> > > |1600|0.272188|0.577065|0.928865|1.339322|1.824441|
> > > |1800|0.255511|0.538944|0.862882|1.236675|1.672794|
> > > |2000|0.240796|0.505589|0.805699|1.148698|1.544486|
> > >
> > > As shown in the table, under the relaxed assumption, the generalization error gradually decreases with increasing replay size. Thank you again for your valuable comments, which helped us explore different scenarios and further refine this work, and we have added the extended analysis to the revised version.
> > >
> > >
> > >
> > >
> > > **Q4: The insights in lines 282-289 and 335-341 are derived for the two task case. Can these insights extend to T>2.**
> > > **A4**：Thank you for pointing out this. In lines 282–289 and 335–341, we analyzed the trend of error variation using the expression for $T=2$ as a special case. **This is because** although Theorem 2-3 provides expressions for memory and generalization error for any number of tasks, we analyze the results as $T=2$ for ease of analysis and intuitive expression when examining how they vary with influencing factors. **Moreover,** based on Theorem 2-3, the expressions for memory and generalization errors at any given time can be easily calculated. **Additionally,** we report the numerical simulation curve variations for $T=8$ in Figure 2-3, which align with the analysis for the simplified case. Thank you again for your valuable comments.
> > >
> > >
> > > **Q5:  Do you sample mini batches from the buffer at random, or do you train with the full buffer at each step. We expect that drawing random mini batches from the buffer may lead to different phenomena and thus deserves discussion.**
> > > **A5**：Thank you for your suggestion. In theoretical analysis, the model is updated once per task to compute the analytical solution. **Under the replay mechanism,** all replayed and current task data are used together to update the model, yielding the expression for model parameters. **In practical deep neural network analysis,** data is typically processed in mini-batches via stochastic gradient descent. With replay, both replayed and current task data are combined in batch training to update the model. **In our experiments,** we use a dynamic buffer, which is a common approach in continual learning that calculates the required sample count per class and adjusts buffer contents dynamically by adding or removing samples. **According to your suggestion,** we also conducted experiments where mini-batches are sampled from the buffer. **Specifically,** a fixed number of samples per category are maintained, and portions are sampled for replay during each training iteration. The experimental results are as follows.
> > >
> > > |Tasks|T1|T2|T3|T4|T5|
> > > |-----|---|---|---|---|---|
> > > |0%|0.123|0.368|0.253|0.090|0.155|
> > > |5%|0.122|0.391|0.291|0.116|0.187|
> > > |10%|0.127|0.463|0.424|0.182|0.279|
> > > |50%|0.128|0.569|0.824|0.707|0.754|
> > >
> > > As shown in the table, the adaptation error gradually increases with the number of tasks across different rehearsal sizes, impacting the model’s performance on the current task, which  is consistent with previous theoretical analysis.
> > >
> > > **Finally, thank you again for your constructive feedback and valuable comments. We look forward to your response and are happy to answer any questions or comments.**

---

### Official Review · Reviewer_c7J2 · 2025-10-31

**Soundness:** 3
**Presentation:** 3
**Contribution:** 3
**Rating:** 6
**Confidence:** 4

**Summary:**

The paper studies rehearsal-based continual learning (CL) through a linear–Gaussian regression lens and provides closed-form expressions for adaptation, memory, and generalization errors under both underparameterized and overparameterized regimes. The core technical results (Theorems 1–3) quantify how rehearsal size s, parameter dimension p, sample size n, inter-task similarity, and noise σ affect these three errors.

Key takeaways include:

(i) more rehearsal is not always better that is there can be a decreasing floor in memory error and a degradation of adaptation/generalization when slightly overparameterized;

(ii) rehearsal helps adaptation when n+s>p+1 (underparameterized) but can hurt when p>n+s+1 (overparameterized) and

(iii) generalization can worsen with larger s in slight overparameterization but improve in heavy underparameterization.

The authors back the analysis with simulations and DNN experiments on MNIST, CIFAR-10/100, and Tiny-ImageNet. The empirical trends broadly match theory, and ablations cover buffer sizes and simple sampling strategies (Random/Reservoir/Herding).

**Strengths:**

- Clear, unified decomposition of CL performance into adaptation/memory/generalization with closed-form expectations. It was asy to reason about levers like s,p,n,σ (Sec. 3–4; Eqs. 2–4, 5–14).

- Main insights include: The rehearsal is helping underparameterization but harming in slight overparameterization. The existence of a decreasing floor for memory error is nice. An analysis of non-monotonic generalization vs. s given by Theorems 1–3 and the discussion under Fig. 2–3.

- The authors present simulations plus DNN studies across four datasets. The trends (e.g., adaptation worsening with larger buffers; memory non-monotonicity) qualitatively match the linear analysis. (Figs. 4–5; Tables 1–4).

- Positioning vs. recent theory on CL and rehearsal (e.g., comparisons to overparameterized analyses and multi-task links) seems valid. It shows awareness of the literature and clarifies scope (Related Work).

**Weaknesses:**

- Modeling assumptions might narrow external validity. Gaussian i.i.d. features, equal per-task n/σ, and the reliance on min-norm solutions limit transfer to realistic non-Gaussian, structured vision features or varying task sizes/noise. I would sugest that the authors consider relaxing Assumption 2 and see if they can can add/report at least partial results beyond equal n,σ (Assumptions 1–2).

- Empirical baselines are somewhat thin. The DNN section primarily varies buffer size/strategy and architecture depth but does not compare against strong non-rehearsal and hybrid CL baselines (e.g., EWC, LwF/Distillation, DER/ER-ACE, prompt-based CL). Without these, it is harder to assess practical implications of “harmful rehearsal” when practitioners could switch methods.

- Sampling strategy analysis seems not full. Table 1 covers adaptation for Random/Reservoir/Herding, but memory and generalization under these strategies are not reported. Also no class-balanced or distance-aware modern coresets are examined. As reported, it kind of weakens the claims about rehearsal “harms” in practice (might be strategy-dependent).

- Clarity on inter-task similarity questionable. The main text often appeals to “task similarity” but operationalizes it variously (parameter distances in theory; class overlap in DNNs). A single main-text definition plus a mapping between theory-side || w_k^∗ − w_j^∗ ||, ||w_k^∗ − w_j^∗||  and dataset-side similarity might be helpful here. (Secs. 3–5, Fig. 4c–f).

- Figures lack uncertainty. Many curves lack confidence intervals. Some axes are log while others are linear without explicit callouts, making visual comparisons harder. Simple recommendation would be to add error bars and consistent scaling. (Figs. 2–5).

- Reproducibility details light. The paper notes multiple runs and averages, but hyperparameters, optimizers, learning-rate schedules, buffer management, and training protocols for each dataset/architecture are mainly in the appendix. A dedicated reproducibility checklist and code link would help. (Sec. 5; Appendix H).

**Questions:**

- Scope beyond equal n, σ. Can the theorems extend to unequal per-task sample sizes and noise? If so, what changes in Eqs. (5–14) qualitatively (e.g., replace n with n_t)? A brief corollary or remark would boost applicability.

- From min-norm to practical SGD: Appendix A argues GD/SGD converges to the min-norm interpolant under zero-loss. In noisy DNN training (non-interpolating), which part of the theory do the authors expect to persist, and what breaks? Do the authors maybe have any experiments with early stopping / weight decay to probe this?

- Sampling strategies: Could the authors show/report memory/generalization errors for Random/Reservoir/Herding (Table 1 only shows adaptation) and include a stronger, recent coreset selection (e.g., gradient-based or diversity-aware) to test whether “harmful rehearsal” is mitigated by better buffers?

- Baselines: It might make sense to add 1–2 strong non-rehearsal CL baselines (e.g., EWC, LwF/distillation, adapter/prompt-based) to the DNN section to contextualize how the outlined rehearsal insights translate into method choice?

- Where is the turning point? The theory predicts non-monotonicity vs. s in slight overparameterization. Can the authors provide a simple practitioner rule (in terms of p,n,s,σ or proxy measures) to detect when increasing s starts to harm? A small diagnostic might add more value to the paper.

**Details Of Ethics Concerns:**

/

---

> ### Author Response · Authors · 2025-11-28
> **Response to Reviewer c7J2  ( Part I )**
>
> **We would like to express our sincere gratitude to the reviewers for their detailed feedback. Below are our responses to each point.**
>
> **Q1: Modeling assumptions might narrow external validity. I would suggest that the authors consider relaxing Assumption 2 and see if they can can add/report at least partial results beyond equal n and σ.**
> **A1**：Thank you for your suggestion. According to your suggestion, we have extended existing theoretical analysis by relaxing Assumption 2 to accommodate scenarios with varying sample sizes and noise variances across different tasks. The partial theoretical results are provided as follows.
> Under the relaxed Assumption 2, the sample size and noise satisfying $n\_1\ne n\_2 \ne ...\ne n\_T$ and $\sigma\_1 \ne \sigma\_2 \ne ... \ne \sigma\_T$, and the optimization objective under the underparameterized regime is given by: $\widehat{\boldsymbol{w}}\_{t}^{(\text {Reh})} := \underset{\boldsymbol{w}}{\arg \min } \left||\boldsymbol{X}\_{t}^{\top} \boldsymbol{w}-\boldsymbol{y}\_{t}\right||^{2}  + \sum\_{i=1}^{t-1} \left||\boldsymbol{Z}\_{i}^{\top} \boldsymbol{w} -\boldsymbol{g}\_{i}\right||^{2}$, and we have
> $$
> \begin{aligned}
> \boldsymbol{w}\_t
> &=(\boldsymbol{X}\_{t}\boldsymbol{X}\_{t}^{\top}+\sum\_{i=1}^{t-1}\boldsymbol{Z}\_{i}\boldsymbol{Z}\_{i}^{\top})^{-1}(\boldsymbol{X}\_{t}\boldsymbol{y}\_{t}+ \sum\_{i=1}^{t-1}\boldsymbol{Z}\_{i}\boldsymbol{g}\_{i}) \\\\
> &=(\boldsymbol{X}\_{t}\boldsymbol{X}\_{t}^{\top}+\sum\_{i=1}^{t-1}\boldsymbol{Z}\_{i}\boldsymbol{Z}\_{i}^{\top})^{-1} [\boldsymbol{X}\_{t}(\boldsymbol{X}\_{t}^{\top}\boldsymbol{w}^\*+\boldsymbol \varepsilon \_t)+\sum\_{i=1}^{t-1}\boldsymbol{Z}\_{i}(\boldsymbol{Z}\_{i}^{\top}\boldsymbol{w}^\*+\boldsymbol \varepsilon \_i)] \\\\
> &= \boldsymbol{w}\_t^\* + (\boldsymbol{X}\_{t}\boldsymbol{X}\_{t}^{\top}+\sum\_{i=1}^{t-1}\boldsymbol{Z}\_{i}\boldsymbol{Z}\_{i}^{\top})^{-1} (\boldsymbol{X}\_{t}\boldsymbol \varepsilon \_t+\sum\_{i=1}^{t-1}\boldsymbol{Z}\_{i}\boldsymbol \varepsilon \_i)
> \end{aligned}
> $$
> Then we calculate the expected $\ell\_2$-norm, which is given by
> $$
> \begin{aligned}
> \mathbb{E}[L\_{i}(\boldsymbol{w}\_{t})] =& \mathbb{E}||\boldsymbol{w}\_{t} - \boldsymbol{w}\_{i}^{\*}||^{2} \\\\
> =& \mathbb{E}||\boldsymbol{w}\_t^\* +   (\boldsymbol{X}\_{t}\boldsymbol{X}\_{t}^{\top}+\sum\_{i=1}^{t-1}\boldsymbol{Z}\_{i}\boldsymbol{Z}\_{i}^{\top})^{-1} (\boldsymbol{X}\_{t}\boldsymbol \varepsilon \_t+\sum\_{i=1}^{t-1}\boldsymbol{Z}\_{i}\boldsymbol \varepsilon \_i)    - \boldsymbol{w}\_{i}^{\*}||^{2} \\\\
> =& \mathbb{E} ||\boldsymbol{w}\_{t}^{\*} - \boldsymbol{w}\_{i}^{\*}||^{2} + \mathbb{E} || (\boldsymbol{X}\_{t}\boldsymbol{X}\_{t}^{\top}+\sum\_{i=1}^{t-1}\boldsymbol{Z}\_{i}\boldsymbol{Z}\_{i}^{\top})^{-1} (\boldsymbol{X}\_{t}\boldsymbol \varepsilon \_t+\sum\_{i=1}^{t-1}\boldsymbol{Z}\_{i}\boldsymbol \varepsilon \_i)||^{2} \\\\
> =& \mathbb{E} ||\boldsymbol{w}\_{t}^{\*} - \boldsymbol{w}\_{i}^{\*}||^{2} +   \frac{p\sigma^{2}\_t}{n + s - p - 1}\\\\
> \end{aligned}
> $$
> Then, the adaptation error can be calculated as
> $$
> \begin{aligned}
> &\mathbb{E}[A(\boldsymbol{w}\_{t})]= \mathbb{E}||\boldsymbol{w}\_{t} - \boldsymbol{w}\_{t}^{\*}||^{2} \\\\
> =&\mathbb{E}||\boldsymbol{w}\_t^\* +   (\boldsymbol{X}\_{t}\boldsymbol{X}\_{t}^{\top}+\sum\_{i=1}^{t-1}\boldsymbol{Z}\_{i}\boldsymbol{Z}\_{i}^{\top})^{-1} (\boldsymbol{X}\_{t}\boldsymbol \varepsilon \_t+\sum\_{i=1}^{t-1}\boldsymbol{Z}\_{i}\boldsymbol \varepsilon \_i)    - \boldsymbol{w}\_{t}^{\*}||^{2} \\\\
> =& ||\boldsymbol{w}\_t^\* -\boldsymbol{w}\_{t}^{\*}||^{2}   +  \mathbb{E}||  (\boldsymbol{X}\_{t}\boldsymbol{X}\_{t}^{\top}+\sum\_{i=1}^{t-1}\boldsymbol{Z}\_{i}\boldsymbol{Z}\_{i}^{\top})^{-1} (\boldsymbol{X}\_{t}\boldsymbol \varepsilon \_t+\sum\_{i=1}^{t-1}\boldsymbol{Z}\_{i}\boldsymbol \varepsilon \_i) ||^{2} \\\\
> =&  \frac{p\sigma^{2}\_t}{n + s - p - 1}\\\\
> \end{aligned}
> $$
> And we can calculate the memory error
> $$
> \begin{aligned}
> & \mathbb{E}[\mathcal{M}(\boldsymbol{w}\_t)] \\\\
> =& \mathbb{E}\left[\frac{1}{t-1} \sum\_{k=1}^{t-1} \left( ||\boldsymbol{w}\_{t} - \boldsymbol{w}\_{k}^{\*}||^{2} - ||\boldsymbol{w}\_{k} - \boldsymbol{w}\_{k}^{\*}||^{2} \right) \right] \\\\
> =& \frac{1}{t-1} \sum\_{k=1}^{t-1} \left[
>     ||\boldsymbol{w}\_{t}^{\*} - \boldsymbol{w}\_{k}^{\*}||^{2}
>     + \frac{p\sigma^{2}\_t}{n\_t + s - p - 1}
>     - ||\boldsymbol{w}\_{k}^{\*} - \boldsymbol{w}\_{k}^{\*}||^{2}
>     - \frac{p\sigma^{2}\_k}{n\_k + s - p - 1}
>     \right] \\\\
> =& \frac{1}{t-1} \sum\_{k=1}^{t-1} ||\boldsymbol{w}\_{t}^{\*} - \boldsymbol{w}\_{k}^{\*}||^{2} - \frac{1}{t-1} \sum\_{k=1}^{t-1}\frac{p\sigma^{2}\_k}{n\_k + s - p - 1} +\frac{p\sigma^{2}\_t}{n\_t + s - p - 1}\\\\
> \end{aligned}
> $$
>
> After relaxing assumptions, adaptation errors in the underparameterized regime are no longer constant but vary across tasks due to differences in sample size and noise variance. However, they still decrease as the replay scale increases, consistent with previous analyses, and the remaining analysis will be provided in Part II.

---

> > ### Author Response · Authors · 2025-11-28
> > **Response to Reviewer c7J2 ( Part II )**
> >
> > Then, we have the generalization error
> > $$
> > \begin{aligned}
> >     \mathbb{E}[\mathcal{G}\left(\boldsymbol{w}\_t\right)]
> >     &= \mathbb{E}\left[\frac{1}{t}\sum\_{k=1}^{t}||\boldsymbol{w}\_{t} - \boldsymbol{w}\_{k}^{\*}||^{2} \right] \\\\
> >     &= \frac{1}{t} \sum\_{k=1}^{t} \mathbb{E}||\boldsymbol{w}\_t^\* +   (\boldsymbol{X}\_{t}\boldsymbol{X}\_{t}^{\top}+\sum\_{i=1}^{t-1}\boldsymbol{Z}\_{i}\boldsymbol{Z}\_{i}^{\top})^{-1} (\boldsymbol{X}\_{t}\boldsymbol \varepsilon \_t+\sum\_{i=1}^{t-1}\boldsymbol{Z}\_{i}\boldsymbol \varepsilon \_i)    - \boldsymbol{w}\_{k}^{\*}||^{2}  \\\\
> >     &= \frac{1}{t}\sum\_{k=1}^{t} \left[ ||\boldsymbol{w}\_{t}^{\*} - \boldsymbol{w}\_{k}^{\*}||^{2} + \frac{p\sigma^{2}\_t}{n\_t + s - p - 1}\right]
> > \end{aligned}
> > $$
> >
> > As demonstrated by theoretical results, for memory error, additional noise terms are introduced, which benefit from the rehearsal size and further reduce memory error. And for generalization error, it exhibits the same trend with respect to rehearsal scale as in the previous analysis. Thank you again for your valuable comments, and we have added the expanded theoretical analysis into the revised version.
> >
> >
> > **Q2: Empirical baselines are somewhat thin.**
> > **A2**： Thank you for your suggestion. We have expanded the experiments to include classical continual learning methods such as EWC[1], LwF[2], iCaRL[3], and DER[4], as well as recent approaches like Foster[5], MEMO[6], and CSReL[7]. The experimental setup follows [8] to ensure fair comparisons. Since EWC and LwF do not natively use replay, we added replay-enabled modules to assess their performance under rehearsal scenarios. The experimental results on CIFAR100 are as follows.
> >
> > |Methods|2000|4000|6000|8000|10000|
> > |---|---|---|---|---|---|
> > |EWC|86.20|84.60|83.17|81.91|80.11|
> > |LwF|73.35|73.86|72.92|73.90|74.76|
> > |iCaRL|84.59|82.98|81.59|80.89|80.19|
> > |DER|73.87|73.57|72.40|72.86|72.40|
> > |FOSTER|79.48|84.71|84.35|83.95|83.10|
> > |MEMO|68.10|66.78|66.01|65.56|65.16|
> > |CSReL|73.29|68.20|68.13|67.96|60.59|
> >
> > As shown in the table, as the replay increases, the average accuracy on the current task decreases for most algorithms, and CSReL exhibits a more pronounced decline, while DER and MEMO decrease more slowly. This may be due to CSReL's use of an additional network to constrain model loss, which can introduce extra interference to current task. We then analyze how forgetting rates vary with rehearsal size. The experimental results are as follows.
> >
> > |Methods|Forg. 1000| Diff. 1000|Forg. 2000 |Diff. 2000 |Forg. 3000|Diff. 3000|Forg. 4000|Diff. 4000|Forg. 5000|Diff. 5000|Forg. 6000|Diff. 6000|
> > |--|--|--|--|--|--|--|--|--|--|--|--|--|
> > |EWC|61.744|\\|49.513|12.232|42.356|7.157|37.633|4.722|33.322|4.311|30.689|2.633|
> > |LwF|40.256|\\|34.567|5.689|31.533|3.033|30.622|0.911|28.422|2.200|26.367|2.056|
> > |iCaRL|59.744|\\|46.356|13.389|40.844|5.511|35.811|5.033|31.856|3.956|31.013|0.843|
> > |DER|23.467|\\|16.656|6.811|12.522|4.133|10.322|2.200|8.489|1.833|7.456|1.033|
> > |FOSTER|18.589|\\|27.156|-8.567|30.267|-3.111|31.456|-1.189|31.090|0.366|30.811|0.279|
> > |MEMO|25.344|\\|19.111|6.233|14.956|4.156|12.700|2.256|10.200|2.500|8.744|1.456|
> > |CSReL|56.733|\\|44.689|12.044|31.411|13.278|27.822|3.589|27.833|-0.011|27.439|0.394|
> >
> >
> > As shown in the table, the forgetting rate decreases for most algorithms as the replay size increases, but the rate of decrease gradually slows as the replay scale grows larger. When the number of replays reaches around 6000, the gain in forgetting resistance is only about 0.3% for some algorithms. Thank you again for your valuable comments.
> >
> > [1]Kirkpatrick et al. Overcoming catastrophic forgetting in neural networks. PNAS 2017.
> > [2]Li et al. Learning without forgetting. TPAMI 2017.
> > [3]Rebuffi et al. icarl: Incremental classifier and representation learning. CVPR 2017.
> > [4]Yan et al. Der: Dynamically expandable representation for class incremental learning. CVPR 2021.
> > [5]Wang et al. Foster: Feature boosting and compression for class-incremental learning. ECCV 2022.
> > [6]Zhou et al. A Model or 603 Exemplars: Towards Memory-Efficient Class-Incremental Learning. ICLR 2023.
> > [7]Tong et al. Coreset selection via reducible loss in continual learning. ICLR 2025.
> > [8]Zhou et al. Class-incremental learning: A survey. TPAMI 2024.
> >
> > **Q3: Sampling strategy analysis seems not full.**
> > **A3**：Thank you for your suggestions. According to your suggestion, we have reported the memory and generalization error results under random, reservoir and herd sampling strategies. Additionally, we incorporate gradient-based coreset sampling strategies [9] and the recent diversity-based reducible loss sampling strategy [10]. The gradient-based strategy (GSS) selects diverse samples by measuring the cosine similarity between a sample’s gradient and those in the reservoir, while the reducible loss strategy (CSReL) selects representative samples based on loss variation, and the experimental results under different sampling strategies are presented in Part III.

---

> > > ### Author Response · Authors · 2025-11-28
> > > **Response to Reviewer c7J2 ( Part III )**
> > >
> > > The experimental results for the adaptation error are as follows.
> > >
> > > |Methods|0|5%|10%|50%|
> > > |--|--|--|--|--|
> > > |Random|0.196|0.212|0.257|0.382|
> > > |Reservoir|0.201|0.222|0.289|0.613|
> > > |Herding|0.196|0.203|0.207|0.218|
> > > |GSS|0.193|0.201|0.215|0.483|
> > > |CSReL|0.198|0.203|0.205|0.213|
> > >
> > > As shown in the table, increasing the replay size leads to higher adaptation error on the current task. This effect is more pronounced for random, reservoir, and gradient-based sampling strategies. The experimental results for the memory errors are as follows.
> > >
> > > |Methods|0|5%|10%|50%|
> > > |--|--|--|--|--|
> > > |Random|16.945|3.602|2.131|0.767|
> > > |Reservoir|17.415|3.314|1.549|0.386|
> > > |Herding|17.255|4.816|3.624|3.389|
> > > |GSS|17.748|5.085|4.035|3.810|
> > > |CSReL|18.049|11.287|11.080|8.245|
> > >
> > > As shown in the table, increasing the buffer size reduces memory error to some extent, with more performance improvements in the initial stages. As replay increases, the performance gains gradually diminish. Under the Herding sampling, replaying 50% of the samples reduces the error by only 0.235 compared to replaying 10% of the samples.
> > >
> > > |Methods|0|5%|10%|50%|
> > > |--|--|--|--|--|
> > > |Random|11.819|2.820|1.849|0.954|
> > > |Reservoir|12.121|2.609|1.422|0.799|
> > > |Herding|12.014|3.049|2.838|2.762|
> > > |GSS|9.942|3.083|2.542|2.700|
> > > |CSReL|10.081|6.467|6.468|4.851|
> > >
> > > As shown in the table, the model's generalization error decreases to varying degrees as the replay size increases. This effect is more pronounced for Herding sampling, Random sampling, and Reservoir sampling strategies. Thank you again for your valuable comments.
> > >
> > > [9]Aljundi et al. Gradient based sample selection for online continual learning. NeurIPS 2019.
> > > [10]Tong et al. Coreset selection via reducible loss in continual learning. ICLR 2025.
> > >
> > >
> > >
> > > **Q4: Clarity on inter-task similarity questionable.**
> > > **A4**：Thank you for your suggestion. In theoretical analysis, differences in parameters directly reflect functional differences, with the distance between optimal parameters naturally reflecting task similarity. **In deep neural networks,** true parameters are inaccessible, making it impossible to directly quantify task similarity. **However,** when the class overlap between two tasks is high, the model tends to learn shared discriminative boundaries, and their optimal solutions are typically close. **Previous work** has addressed inter-task similarity in various ways: [11][12] adjust it based on classes overlap, [13] use distribution connections via pixel permutations, and [14] leverage semantic connections across hyperclasses. **Therefore,** beyond classes, we also analyze performance under distribution and semantic similarity metrics. The results are presented in Appendix H, with detailed experimental results as follows.
> > >
> > > |Similarity Metrics| |2000|4000|6000|8000|
> > > |---|---|---|---|---|---|
> > > |Class Similarity|Low|1.059|0.608|0.525|0.546|
> > > | |Medium|0.922|0.540|0.512|0.520|
> > > | |High|0.788|0.369|0.335|0.321|
> > > |Distribution Similarity|Low|-0.051|-0.040|-0.033|-0.021|
> > > | |Medium|-0.070|-0.037|-0.050|-0.046|
> > > | |High|-0.053|-0.052|-0.048|-0.048|
> > > |Semantic Similarity|Low|0.388|0.315|0.305|0.316|
> > > | |Medium|0.476|0.269|0.239|0.248|
> > > | |High|-0.052|-0.043|-0.039|-0.027|
> > >
> > > As shown in the table, the memory performance generally improves with higher task similarity. Negative memory errors occur under high distributional and semantic similarity, indicating positive knowledge transfer when tasks are more similar. Thank you again for your valuable comments, and we have added this definition in the revised version.
> > >
> > > [11]Ramasesh et al. Anatomy of Catastrophic Forgetting: Hidden Representations and Task Semantics. ICLR 2021.
> > > [12]Lin et al. Theory on forgetting and generalization of continual learning. ICML 2023.
> > > [13]Goldfarb et al. The Joint Effect of Task Similarity and Overparameterization on Catastrophic Forgetting-An Analytical Model. ICLR 2024.
> > > [14]Ramesh et al. “Model Zoo: A Growing Brain That Learns Continually.” ICLR 2021.
> > >
> > > **Q5: Figures lack uncertainty.**
> > > **A5**：Thank you for your suggestion. In Figures 2–3, logarithmic axes are used to capture variations in model error under different parameterization regimes. The horizontal axis illustrates the transition from underparameterization to overparameterization, while the vertical axis depicts the differential impacts of factors such as noise levels. It is worth noting that logarithmic scaling provides a wider coordinate range while preserving the original trends. Similar configurations have been used in [15][16][17] to reveal subtle variations in model error. According to your suggestion, we have annotated the error bars and axes in the revised version. Thank you again for your valuable comments.
> > >
> > > [15]Evron et al. "How catastrophic can catastrophic forgetting be in linear regression? " COLT 2022.
> > > [16]Li et al. Fixed design analysis of regularization-based continual learning. CoLLAs 2023.
> > > [17]Zhao et al. A Statistical Theory of Regularization-Based Continual Learning. ICML 2024.

---

> > > > ### Author Response · Authors · 2025-11-28
> > > > **Response to Reviewer c7J2 ( Part IV )**
> > > >
> > > > **Q6: Reproducibility details light.**
> > > > **A6**：Thank you for your suggestions. We have provided the experimental code in the supplementary materials, along with a dedicated reproducibility checklist. Additionally, an anonymous code link is available: https://anonymous.4open.science/r/URCL-8BEC. Thank you again for your valuable comments.
> > > >
> > > > **Q7: Scope beyond equal n, σ. Can the theorems extend to unequal per-task sample sizes and noise.**
> > > > **A7**： Thank you for your suggestion. We have extended the theoretical analysis to cases with varying sample sizes $n$ and noise levels $σ$ across tasks. The detailed analysis is provided in A1, and we have included this extended analysis as a remark in the revised version. Thank you again for your valuable comments.
> > > >
> > > >
> > > >
> > > > **Q8: The Appendix A argues GD/SGD converges to the min-norm interpolant under zero-loss. Do the authors maybe have any experiments with early stopping / weight decay to probe this.**
> > > > **A8**：Thank you for your suggestion. In Appendix A, we show that gradient descent gradient descent converges to the minimum norm solution under zero loss in continual learning scenarios. According to your suggestion, we conducted early stopping experiments to validate this analysis. Specifically, in the continual learning setup with two tasks, we measured the norm difference between the solution obtained after training the previous task and the minimum norm solution. The experimental results are as follows.
> > > >
> > > > |Epoch|Train Loss|Solution Norm|Difference|Relative Error|
> > > > |---|---|---|---|---|
> > > > |50|4.2465|7.3235|2.4340|29.53%|
> > > > |100|0.9057|7.8090|1.3898|16.86%|
> > > > |250|0.0706|8.1575|0.5051|6.13%|
> > > > |300|0.0390|8.1872|0.3873|4.70%|
> > > > |350|0.0226|8.2052|0.3008|3.65%|
> > > > |500|0.0051|8.2293|0.1465|1.78%|
> > > > |800|0.0003|8.2409|0.0369|0.45%|
> > > > |850|0.0002|8.2416|0.0294|0.36%|
> > > > |900|0.0001|8.2421|0.0235|0.28%|
> > > > |950|0.0001|8.2424|0.0187|0.23%|
> > > > |992（ES）|0.0001|8.2427|0.0155|0.0019%|
> > > > |1000|0.0001|8.2427|0.0155|0.0019%|
> > > >
> > > > As shown by the experimental results, although the norm of the gradient descent solution does not exactly match the minimum-norm solution in continual learning scenario, the gap decreases progressively with more training epochs. The gradient-updated solution continues to converge toward the small-norm trend, and future work will extend this analysis to more complex scenarios. Thank you again for your valuable comments.
> > > >
> > > >
> > > >
> > > > **Q9: Could the authors show memory/generalization errors for Random/Reservoir/Herding and include a stronger, recent coreset selection.**
> > > > **A9**：Thank you for your suggestion. We have presented the memory and generalization error for Random, Reservoir, and Herding sampling in A3. Moreover, we introduced gradient-based coreset and diversity-based reducible loss sampling strategies for analysis. Detailed analysis can be found in A3. Thank you again for your valuable comments.
> > > >
> > > >
> > > >
> > > > **Q10: It might make sense to add 1–2 strong non-rehearsal CL baselines.**
> > > > **A10**：Thank you for your suggestion. We have added strong continual learning baselines in A2, including classical non-replay methods (EWC[1], LwF[2]), replay-based methods (iCaRL[3], CSReL[7]), and hybrid methods (DER[4], Foster[5], MEMO[6]). Detailed experimental analysis can be found in A2. Thank you again for your valuable comments.
> > > >
> > > >
> > > >
> > > > **Q11: The theory predicts non-monotonicity vs. s in slight overparameterization. Can the authors provide a  practitioner rule to detect when increasing s starts to harm.**
> > > > **A11**： Thank you for your suggestions. We demonstrate that theoretical predictions exhibit a non-monotonic relationship with replay size under overparameterization in this work. According to your suggestion, we further analyzed metrics such as $p$, $n$ and $\sigma$ to identify the inflection point of model error with respect to $s$. This analysis has been added to the revised version to explain the theoretical results. Thank you again for your valuable comments.
> > > >
> > > >
> > > > **Finally, thank you again for your constructive feedback and valuable comments. We look forward to your response and are happy to answer any questions or comments.**

---

### Official Review · Reviewer_au2v · 2025-11-01

**Soundness:** 3
**Presentation:** 3
**Contribution:** 2
**Rating:** 2
**Confidence:** 4

**Summary:**

This paper provides a theoretical investigation of rehearsal-based continual learning, focusing on how model capacity influences adaptation, memory, and generalization performance. Using a linear Gaussian regression framework, the authors derive closed-form expressions for different error components under both underparameterized and overparameterized settings. Theoretical findings show that underparameterized models benefit from rehearsal, while overparameterized ones may experience degradation. Simulations and neural network experiments on MNIST, CIFAR-10/100, and Tiny-ImageNet confirm that the analytical trends hold in practice, suggesting the theoretical results capture key behaviors of real-world continual learning systems.

**Strengths:**

The theoretical analysis of continual learning with rehearsal is an important and meaningful topic. The paper is clearly presented, combining theoretical results derived from a linear model with empirical validation on deep neural networks.

**Weaknesses:**

A major limitation of this paper is the lack of novelty in its theoretical results. Much of the analysis appears to replicate existing work rather than introduce new insights. In particular, compared with the recent work [Deng et al., 2025], the setup (including the linear regression framework, Gaussian feature assumptions, rehearsal mechanism, and even the mathematical expressions in the theorems) appears nearly identical.

*Reference:*
Junze Deng, Qinhang Wu, Peizhong Ju, Sen Lin, Yingbin Liang, and Ness Shroff. *Unlocking the Power of Rehearsal in Continual Learning: A Theoretical Perspective*, 2025.

**Questions:**

1. As mentioned in the weaknesses, the comparison with related literature is not sufficiently careful. The citation of [Deng et al., 2025] in the last paragraph of Section 2 is too brief ("While some of these studies also use the linear Gaussian model, they focus on different aspects of continual learning"). A detailed comparison is needed to clarify which results are novel and which have already been established.
2. The font size in Figures 2 and 3 is too small to read comfortably. Please enlarge the text for clarity.
3. In the last paragraph of Section 3, the authors discuss striking a balance among adaptability, memorability, and generalizability. However, the paper does not provide any quantitative metric to measure or verify this balance. How can one determine whether such a balance is achieved in practice?

---

> ### Author Response · Authors · 2025-11-28
> **Response to Reviewer au2v**
>
> **We sincerely appreciate the reviewers’ constructive feedback and valuable comments. Below are our responses to each point.**
>
> **Q1: The lack of novelty in its theoretical results. Compared with the recent work [Deng et al., 2025], the setup appears nearly identical.**
> **A1**： We sincerely appreciate the reviewers’ valuable comments. However, we must clarify the distinctions and contributions of this work relative to the study.
> **Firstly,** the Linear Gaussian framework has been widely used to analyze meta-learning [1], transfer learning [2], Mixture-of-Experts models [3], and continual learning [4] due to its analytical clarity.  **Secondly,** recent work [Deng et al., 2025] studied replay strategies in continual learning, showing that sequential replay outperforms concurrent replay when tasks are dissimilar. The Appendix E provides a more detailed comparison of their distinctions and connections. **Then,** we must clarify the contributions and significance of our work.
>
> **Different Research Objectives:** This work investigates the role and limitations of replay size, focusing on whether the replay adversely affects continual learning performance. This work uncovers several intriguing theoretical insights.
>
> **Interesting Theoretical Finding:** This work reveals that increasing the rehearsal scale does not always improve performance, contrary to traditional assumptions. In fact, the rehearsal mechanism can sometimes impair a model’s adaptability to the current task. Even when forgetting is mitigated, there exists a lower bound on memory error reduction. Furthermore, the proposed proposition shows that robust adaptability, effective memory retention, and strong performance on the initial task are all essential for achieving good generalization.
>
> **Detailed Experimental Validation:** We confirmed these theoretical findings through numerical simulations and extended them to deep neural networks. By examining different network architectures, diverse sampling strategies, and longer task sequences, experiments on real-world datasets further validated and enriched these theoretical insights.
>
> **Finally,** we do not deny the contributions made by [Deng et al., 2025] to the field of replay-based continuous learning. **However,** unlike their work, this study focuses on a different aspect-specifically, the potential detrimental effects of **replay scale** on continual learning-and has led to theoretical findings that are supported by experimental validation. We fully understand concerns regarding potential risks and appreciate the opportunity to clarify this point. Thank you again for your valuable comments.
>
> [1]Ju et al. Theoretical characterization of the generalization performance of overfitted meta-learning. ICLR 2023.
> [2]Ju et al. Theoretical Analysis on the Generalization Power of Overfitted Transfer Learning.
> [3]Li et al. Theory on Mixture-of-Experts in Continual Learning. ICLR 2025.
> [4]Lin et al. Theory on forgetting and generalization of continual learning. ICML 2023.
>
>
> **Q2: The comparison with related literature is not sufficiently careful.**
> **A2**：Thank you for pointing out this. We provide a detailed comparison of the distinctions and connections with related work in Appendix E. **Specifically,** [Deng et al., 2025] focused on how replay strategies affect continual learning and showed that sequential replay outperforms concurrent replay when tasks are dissimilar. **In contrast,** we emphasize the negative impact of replay scale relative to the model's current task learning. The detailed analysis can be found in A1. Thank you again for your valuable comments.
>
>
>
> **Q3: The font size in Figures 2 and 3 is too small to read comfortably.**
> **A3**：Thank you for pointing out this. We have adjusted the font size of the Figures 2-3 in the revised version to enhance visual clarity. Thank you again for your valuable feedback.
>
>
>
> **Q4: The paper does not provide any quantitative metric to measure or verify balance.**
> **A4**： Thanks for your valuable comments. Considering that ideal continual learning should balance adaptability, memory, and generalization, we introduce adaptation error on the current task to examine the impact and limitations of rehearsal. Specifically, the analysis reveals that increasing replay scale may negatively impact learning on the current task and that memory error exhibits a lower bound, and future research will explore further foster this balance. Finally, in Section 4, we establish connections among the three performance, showing that strong generalization requires better adaptation, memory, and initial-task performance-revealing the intrinsic interdependencies within model capabilities.  Thank you again for your valuable comments.
>
>
> **Finally, thank you again for your constructive feedback and valuable comments. We look forward to your response and are happy to answer any questions or comments.**

---

### Meta-Review · Area_Chair_BhRQ · 2025-12-19

**Summary:**

This paper provides a theoretical analysis of continual learning to understand how rehearsal size affects adaptation, generalization, and memory performance. The proposed analytical framework relies on several assumptions, including a linear Gaussian model, independent and identically distributed features, equal per task noise variance $\sigma$, and minimum norm solutions. The analysis covers both the underparameterized regime $(n+s > p+1)$ and the overparameterized regime $(p > n+s+1)$. Under these assumptions, the authors derive closed form expressions for key metrics such as adaptation and generalization errors, explicitly characterizing their dependence on rehearsal size, parameter dimension, sample size, and task similarity. The main theoretical takeaway is that, in the overparameterized regime, increasing the rehearsal size beyond a certain point can degrade both adaptation and generalization performance. Experiments on deep neural networks are provided to empirically validate this theoretical finding regarding the impact of rehearsal.

**Reviewer Concerns:**

All reviewers agree that analyzing the effect of rehearsal in continual learning from a theoretical perspective is interesting and important. The paper is generally well written, with clear motivation and takeaways. However, several concerns were raised. Reviewer au2v’s main concern is the limited technical novelty, noting that the analysis closely follows recent work such as Deng et al. 2025 under nearly identical assumptions and setups. Reviewer c7J2 questioned the restrictive nature of the assumptions and their practical relevance, and also raised concerns about the experimental evaluation, including the lack of comprehensive baselines and the need for clarification on several experimental details. Reviewer EQuC similarly expressed concerns about both the realism of the assumptions and the degree of technical novelty given the overlap with prior work. Reviewer juEZ questioned the simplicity of the theoretical setup and requested evaluations on more complex datasets.

Based on my reading, the authors’ responses adequately addressed several experimental concerns, particularly those raised by Reviewer c7J2 and Reviewer EQuC, by adding multiple additional experiments and baseline comparisons. I agree that starting from simplified linear models is a reasonable approach for gaining insight into complex continual learning phenomena, and the added experiments significantly improve the empirical section. While the authors did not expand to additional datasets, I believe the dataset coverage is reasonable, and the added baseline experiments are valuable. However, after carefully comparing this paper with Deng et al. 2025, I find that the theoretical setups are indeed quite similar, especially in the overparameterized regime, where the derivations share a very similar spirit. As a result, the technical contribution appears limited. While the high level takeaway regarding the impact of rehearsal in continual learning is important, it would be more compelling to extend the analysis to broader and more realistic settings. For example, recent work has begun to analyze multi layer MLPs or CNNs, or to study the behavior of concrete optimization algorithms such as SGD or the algorithms used in the experiments rather than assuming direct access to minimum norm solutions.

**Reviewer Scores:**

Overall, I consider this to be a borderline paper with borderline scores $(2, 4, 6, 6)$. Despite the authors’ substantial effort in improving the experimental evaluation, which I believe successfully addresses several reviewer concerns, the issues of limited technical novelty and restrictive assumptions raised by Reviewers au2v, EQuC, c7J2, and juEZ remain largely unresolved. As a result, these concerns are unlikely to significantly change reviewer scores. For these reasons, I lean toward rejecting the paper at this time.

---

### Decision · Program_Chairs · 2026-01-26

Reject